# Fine particle pH for Beijing winter haze as inferred from different thermodynamic equilibrium models

Shaojie Song[1,*], Meng Gao[1,*], Weiqi Xu[2], Jingyuan Shao[3,4], Guoliang Shi[5], Shuxiao Wang[6], Yuxuan Wang[7,8], Yele Sun[2,9], Michael B. McElroy[1,10]

[1]School of Engineering and Applied Sciences, Harvard University, Cambridge, Massachusetts 02138, USA
[2]State Key Laboratory of Atmospheric Boundary Physics and Atmospheric Chemistry, Institute of Atmospheric Physics, Chinese Academy of Sciences, Beijing 100029, China
[3]Laboratory for Climate and Ocean-Atmosphere Studies, Department of Atmospheric and Oceanic Sciences, School of Physics, Peking University, Beijing 100871, China
[4]Department of Atmospheric Sciences, University of Washington, Seattle, Washington 98195, USA
[5]State Environmental Protection Key Laboratory of Urban Ambient Air Particulate Matter Pollution Prevention and Control, College of Environmental Science and Engineering, Nankai University, Tianjin 300071, China
[6]State Key Joint Laboratory of Environmental Simulation and Pollution Control, School of Environment, Tsinghua University, Beijing 100084, China
[7]Department of Earth and Atmospheric Sciences, University of Houston, Houston, Texas 77004, USA
[8]Department of Earth System Science, Tsinghua University, Beijing 100084, China
[9]University of Chinese Academy of Sciences, Beijing 100049, China
[10]Department of Earth and Planetary Sciences, Harvard University, Cambridge, Massachusetts 02138, USA
[*]These authors contributed equally to this work.

*Correspondence to*: Shaojie Song (songs@seas.harvard.edu), Michael B. McElroy (mbm@seas.harvard.edu), Yele Sun (sunyele@mail.iap.ac.cn)

**Abstract.** pH is an important property of aerosol particles but is difficult to measure directly. Several studies have estimated the pH values for fine particles in North China winter haze using thermodynamic models (i.e., E-AIM and ISORROPIA) and ambient measurements. The reported pH values differ widely, ranging from close to 0 (highly acidic) to as high as 7 (neutral). In order to understand the reason for this discrepancy, we calculated pH values using these models with different assumptions with regard to model inputs and particle phase states. We find that the large discrepancy is due primarily to differences in the model assumptions adopted in previous studies. Calculations using only aerosol phase composition as inputs (i.e., reverse mode) are sensitive to the measurement errors of ionic species and inferred pH values exhibit a bimodal distribution with peaks between −2 and 2 and between 7 and 10, depending on whether anions or cations are in excess. Calculations using total (gas plus aerosol phase) measurements as inputs (i.e., forward mode) are affected much less by these measurement errors. In future studies, the reverse mode should be avoided whereas the forward mode should be used. Forward mode calculations in this and previous studies collectively indicate a moderately acidic condition (pH from about 4 to about 5) for fine particles in North China winter haze, indicating further that ammonia plays an important role in determining this property. The particle phase state assumed, either stable (solid plus liquid) or metastable (only liquid), does not significantly impact pH predictions. The unrealistic pH values of about 7 in a few previous studies (using the standard ISORROPIA model and stable state assumption) resulted from coding errors in the model, which have been identified and fixed in this study.

## 1 Introduction

Aerosols in the atmosphere are reported to be associated with respiratory and cardiovascular diseases, and affect climate and ecosystems via aerosol-radiation-cloud interactions (Ramanathan et al., 2001; Lim et al., 2012; Ma et al., 2016). Liquid water is a ubiquitous component of aerosols (Nguyen et al., 2016). The hydrogen ion activity expressed on a logarithmic scale, pH, is an essential property describing the acidity of aqueous aerosols and has been suggested to influence particle formation, toxicity, and nutrient delivery. pH plays a role in the formation of sulfate and secondary organic aerosols (Jang et al., 2002; Xu et al., 2015a; Cheng et al., 2016), and also changes the gas-particle partitioning of semi-volatile species (Keene et al., 2004; Guo et al., 2016; Weber et al., 2016). It affects the solubility of trace metals and thus aerosol toxicity (Ghio et al., 2012; Fang et al., 2017). Low pH may enhance iron mobility in dust and impact ocean productivity (Meskhidze et al., 2003).

In spite of its significance, ambient particle pH is still poorly constrained. The direct filter sampling approach is challenged by the nature of hydrogen ion in that its concentration in a solution does not scale in proportion to the level of dilution, and is also subject to sampling errors (Hennigan et al., 2015). A few studies have determined the pH of laboratory generated particles using colorimetry/spectrometry and Raman microspectroscopy (Li and Jang, 2012; Rindelaub et al., 2016; Craig et al., 2017), but such techniques have not been applied to ambient particles partly due to their much more complex chemical and physical properties. Therefore, an indirect (or proxy) method—thermodynamic equilibrium modeling—has been widely used to estimate particle pH for many regions of the world (Yao et al., 2007; Zhang et al., 2007; Bougiatioti et al., 2016; Guo et al., 2017a; Murphy et al., 2017; Parworth et al., 2017). A number of thermodynamic models have been developed, subject to the principle of minimizing the Gibbs energy of the multi-phase aerosol system, leading to a computationally intensive optimization problem. Thus these models usually incorporate a variety of simplifications and assumptions in their calculations (Fountoukis and Nenes, 2007).

Over the past few years, several studies have estimated values of fine particle pH during North China winter haze events using the E-AIM (Friese and Ebel, 2010) and ISORROPIA (Fountoukis and Nenes, 2007) thermodynamic equilibrium models, as summarized in Table 1. The inferred pH values varied significantly, ranging from close to 0 (highly acidic) to about 7 (neutral). The primary goal of this study is to critically examine the reason for such a large discrepancy. In order to address this problem, we calculate particle pH using the ISORROPIA and E-AIM models under different assumptions (e.g., open vs. closed systems and stable vs. metastable states). The measured data on gas and particle compositions and meteorological parameters collected in Beijing winter serve as model inputs. We have identified and fixed important coding errors in ISORROPIA which are involved in pH calculations when a closed system and the stable state are assumed (details in the Supplement, Sect. S1). We compare pH values obtained from these different thermodynamic calculations in Sects. 3.1 and 3.2, as well as in Sect. 3.3 with results from previous winter haze studies. The assumptions and limitations of thermodynamic models are discussed in Sect. 3.4.

## 2 Methods

### 2.1 Field measurements

During 2014 winter (from 17 November to 12 December), measurements of air pollutants were conducted at an urban site in Beijing (Institute of Atmospheric Physics, Chinese Academy of Sciences, 39°58'N 116°22'E, 49 m ASL). The $PM_{2.5}$ and $PM_1$ chemical compositions and several semi-volatile gases were measured with high time resolution, as described below. The ambient temperature and relative humidity (RH) were recorded by a Rotronic HC2-S3 probe. Concentrations of carbon monoxide were also measured (Model 48i, Thermo Fisher Scientific Inc., USA).

The concentrations of six water-soluble inorganic ions (i.e., $SO_4^{2-}$, $NO_3^-$, $Cl^-$, $NH_4^+$, $Na^+$, and $K^+$) in $PM_{2.5}$ and three semi-volatile gases (i.e., $NH_3$, $HNO_3$, and $HCl$) were measured with a time resolution of 30 min using a Gas and Aerosol Collector Ion Chromatography (GAC-IC) system. The instrument was modified based on the Steam Jet Aerosol Collector (Khlystov et al., 1995) in order to better apply to the heavily polluted conditions in China. Ambient air was drawn in at a flow rate of 16.7 L min$^{-1}$ through a $PM_{2.5}$ cyclone inlet. Trace gases were absorbed in a wet annular denuder and then the water-soluble ions in the aerosols were extracted with an improved aerosol collector. The samples of aqueous solution were quantified by two ion chromatography analyzers. Details on the GAC-IC performance (including the detection limits for each species) were described in a previous publication (Dong et al., 2012). The measurement uncertainties arise from several inaccuracies such as internal calibration, pressure and flow control, and collection efficiencies. The intercomparison experiments with filter sampling and other online methods (e.g., Monitor for AeRosols and Gases in Ambient Air, MARGA; Metrohm, Switzerland) reveal that the overall relative uncertainties of the GAC-IC system remain within ± 20% for major species (Dong et al., 2012; Young et al., 2016).

An Aerodyne high-resolution time-of-flight aerosol mass spectrometer (referred to as the AMS) was used to measure size-resolved non-refractory submicron aerosol (NR-$PM_1$) species (DeCarlo et al., 2006). The detailed operations and calibrations of the AMS have been described elsewhere (Xu et al., 2015b; Sun et al., 2016). Briefly, aerosol particles were drawn into the sampling chamber at a flow rate of 10 L min$^{-1}$, of which ~ 0.1 L min$^{-1}$ was isokinetically sampled into the AMS after being dried with a silica gel dryer. Concentrations were obtained for organics, sulfate, nitrate, ammonium, and chloride. The AMS was calibrated for ionization efficiency using pure $NH_4NO_3$ particles following standard protocols (Jayne et al., 2000). A constant collection efficiency of 0.5 was chosen because (1) particles were dried before being analyzed, (2) the mass fraction of $NH_4NO_3$ was smaller than 0.4, and (3) the particle acidity was not high enough to affect collection efficiency substantially (Sun et al., 2016). The default relative ionization efficiencies, except for ammonium that was determined from pure $NH_4NO_3$, were applied to all of the species for mass quantifications. The overall uncertainties for each species were estimated following Bahreini et al. (2009) with details provided in the Supplement, Sect. S2.

**2.2 pH prediction by thermodynamic models**

In this study, pH is defined in Eq. (1) as the negative logarithm with base 10 of the hydrogen ion activity on a molality basis, which is recommended by IUPAC (*goldbook.iupac.org/html/P/P04524.html*) and is also consistent with previous studies (Guo et al., 2016; Battaglia et al., 2017; Pye et al., 2018).

$pH = -\log_{10}(\gamma_{H^+} m_{H^+}) = -\log_{10} m_{H^+} - \log_{10} \gamma_{H^+}$                                    (1)

where $m_{H^+}$ and $\gamma_{H^+}$ indicate the molality (mol kg$^{-1}$ water) and the molality-based activity coefficient (a factor accounting for deviations from ideal behavior) of hydrogen ions, respectively. Here, particle pH is predicted using the latest E-AIM (version IV; *www.aim.env.uea.ac.uk*) and ISORROPIA (version II; *isorropia.eas.gatech.edu*) thermodynamic equilibrium models. E-AIM is usually considered as a benchmark model (Zaveri et al., 2008; Seinfeld and Pandis, 2016), while ISORROPIA employs

a number of simplifications to make it computationally efficient for application in large-scale atmospheric models (Fountoukis and Nenes, 2007; Pye et al., 2009). E-AIM uses the Pitzer, Simonson, and Clegg equations to calculate activity coefficients for water and ions (Pitzer and Simonson, 1986; Clegg et al., 1992; Wexler and Clegg, 2002). With ISORROPIA, $\gamma_{H^+}$ and $\gamma_{OH^-}$ are assumed equal to unity, whereas the activity coefficients for the other ionic pairs (e.g., H$^+$–Cl$^-$) are calculated (Fountoukis and Nenes, 2007). For both models, the equilibrium state is calculated at a given temperature and RH. E-AIM solves for the

equilibrium of an NH$_4^+$–H$^+$–Na$^+$–SO$_4^{2-}$–NO$_3^-$–Cl$^-$–H$_2$O inorganic aerosol and its precursor gases (HNO$_3$, NH$_3$, and HCl), and can also include certain organic compounds. E-AIM assumes that the inorganic ions and organic solutes do not influence each other in the aqueous solution, when estimating their activity coefficients (Clegg et al., 2001). ISORROPIA treats only inorganic aerosols but includes more crustal species (i.e., Ca, K, and Mg) when compared to E-AIM. The two models can solve for either forward (or closed, in which the total (gas + aerosol) concentration of each species is fixed) or reverse (or open, in which the

concentration of each species in the aerosol phase is fixed) condition. The model outputs include concentrations for each species in the solid, liquid, and gas phases (Fountoukis and Nenes, 2007).

It is well known that atmospheric aerosol particles can exist in two states of thermodynamic equilibrium, stable and metastable, depending on their chemical composition and RH history (Rood et al., 1989). Particles in the stable state may be solid, solid

plus liquid, or liquid as ambient RH increases (liquid phase appears when ambient RH reaches the deliquescence RH). If the ambient RH over a completely liquid aerosol decreases below the deliquescence RH, the aerosol may not crystalize immediately but may constitute a supersaturated aqueous solution (i.e., in the metastable state). The ambient RH varies widely over the North China Plain (NCP) in winter. Synoptic weather patterns are dominated by the Siberian high-pressure system (Jia et al., 2015), under which the northerly winds bring dry and clean air into this region with ambient RH often dropping to

as low as about 20% (when aerosol particles are most likely solid). When the northerly winds slacken, often occurring during the NCP winter haze events, the atmospheric conditions are characterized by stagnant inversion, weak southerly winds, and rapid accumulation of both air pollutants and water vapor, and the ambient RH often reaches 80–90% (when aerosol particles are most likely liquid) (Zheng et al., 2015; Gao et al., 2016; Sun et al., 2016; Wang et al., 2016; Tie et al., 2017; Yin et al.,

2017). So far, there has been no observational evidence to suggest whether Beijing winter haze fine particles are in a metastable or stable state. It is also unlikely to figure out particle phase states from theoretical calculations because of the very large variability of ambient RH and the difficulty in estimating the efflorescence RH for multicomponent salts (Seinfeld and Pandis, 2016). Thus, a practical approach is to predict pH for both stable and metastable states, which can provide an estimate of its uncertainty due to the phase state assumption. In this study, pH values are calculated in both stable and metastable states using ISORROPIA, whereas E-AIM (version IV) can only address the stable state condition.

We have identified and fixed coding errors in the standard ISORROPIA model, which may significantly affect forward stable mode calculations of pH. In this study, the ISORROPIA model with these errors fixed is denoted as the revised ISORROPIA model. Details concerning the revision of ISORROPIA are provided in the Supplement, Sect. S1. Briefly, in several subregimes of the solution domain, the standard ISORROPIA model fails to consider the partitioning of $NH_3$ between the gas and aqueous phases, and therefore the predicted particle pH is very often around 7 (neutral). Importantly, we find that Beijing winter haze conditions (ammonia-rich) happen to belong to the subregimes where coding errors exist. For example, as shown in Fig. 1, the standard ISORROPIA predicts a nearly constant pH of 7.6 for most cases when $\frac{\text{total } [NH_x]}{17} > \frac{\text{total } [H_2SO_4]}{49}$, where total $[NH_x]$ and $[H_2SO_4]$ are the concentrations (gas plus aerosol phase) of the corresponding species. The revised ISORROPIA model predicts pH lower than 7 (acidic), which varies as a function of $[NH_x]$ and $[H_2SO_4]$. Interestingly, these coding errors have little effect on the predicted gas phase $NH_3$ concentrations (Wang et al., 2016; Guo et al., 2017b), and thus cannot be identified by simply comparing the measured and predicted $NH_3$ phase partitioning (Supplement, Sect. S1). A few previous studies have been affected by these ISORROPIA coding errors (Wang et al., 2016; He et al., 2017).

## 2.3 Ion balance and equivalent ratio

The ion balance and equivalent ratio are calculated using the charge equivalent measured ion concentrations and Eqs. (2–5):

$$[\text{cations}] = \frac{[NH_4^+]}{18} + \frac{[Na^+]}{23} + \frac{[K^+]}{39} + \frac{[Ca^{2+}]}{20} + \frac{[Mg^{2+}]}{12} \tag{2}$$

$$[\text{anions}] = \frac{[SO_4^{2-}]}{48} + \frac{[NO_3^-]}{62} + \frac{[Cl^-]}{35.5} \tag{3}$$

$$\text{ion balance} = [\text{cations}] - [\text{anions}] \tag{4}$$

$$\text{equivalent ratio} = [\text{cations}]/[\text{anions}] \tag{5}$$

where $[NH_4^+]$, $[Na^+]$, $[K^+]$, $[Ca^{2+}]$, $[Mg^{2+}]$, $[SO_4^{2-}]$, $[NO_3^-]$, and $[Cl^-]$ are the mass concentrations ($\mu g\ m^{-3}$) of these ions in the atmosphere. [cations] and [anions] denote the sum of charge equivalent total molar concentrations ($\mu mol\ m^{-3}$) of cations and anions, respectively. Although they are straightforward to calculate, a few recent studies (Hennigan et al., 2015; Guo et al., 2016; Murphy et al., 2017) have demonstrated that the ion balance and equivalent ratio calculated from ambient particle measurements should not be used to predict the acidity of particles, especially under ammonia-rich conditions, for several reasons summarized as follows. (1) This would require all ions other than $H^+$ and $OH^-$ to be measured with both very high

accuracy and precision, conditions unlikely to be achieved in practice. For example, the filter sampling of semi-volatile species ($NH_4^+$, $NO_3^-$, and $Cl^-$) is subject to both positive and negative biases (Pathak et al., 2004; Wei et al., 2015). Organic acid salts which may contribute significantly to charge balance are usually ignored. In ammonia-rich environments, $H^+$ commonly accounts for only a tiny fraction of the total concentrations of ions and hence its concentration falls within the range of the accumulated analytical uncertainties for measured ions. (2) The dissociation states of many potentially important ionic species (e.g., $HSO_4^-$ and organic acids) are not considered. (3) The activity coefficients of ionic species are unknown.

## 2.4 Overview of measurements and model calculations

In the measurement period exhibited here, there were five pollution episodes characterized by high particulate matter (PM) concentrations and RH (Fig. S4). The chemical measurements, along with ambient temperature and RH, are converted to hourly averages and used as inputs to E-AIM and ISORROPIA. This study mainly uses $PM_{2.5}$ data because water-soluble ions are measured and more chemical species ($Na^+$ and $K^+$) are available. The pH values of $PM_1$ are also estimated for comparison. $SO_4^{2-}$, $NO_3^-$, $Cl^-$, and $NH_4^+$ are identified as the major inorganic ions and their concentrations are positively correlated with RH. With respect to measurements of semi-volatile gases, the mixing ratios of $NH_3$ (18 ± 9 ppb, median ± median absolute deviation) are high whereas $HNO_3$ (0.08 ± 0.04 ppb) and HCl (0.25 ± 0.07 ppb) are observed to be low, consistent with Liu et al. (2017a). With Eqs. (6–8), we can calculate the total (gas $NH_3$ + particle $NH_4^+$) $NH_x$ concentrations, the $NH_x$ concentrations required for the overall (gas + particle phases) charge balance, and the excess $NH_x$ concentrations.

$$\text{Total NH}_x = 17 \times \left( \frac{[NH_4^+]}{18} + \frac{[NH_3]}{22.4} \right) \tag{6}$$

$$\text{Required NH}_x = 17 \times \left( \frac{[SO_4^{2-}]}{48} + \frac{[NO_3^-]}{62} + \frac{[Cl^-]}{35.5} + \frac{[HNO_3]}{22.4} + \frac{[HCl]}{22.4} - \frac{[Na^+]}{23} - \frac{[K^+]}{39} - \frac{[Ca^{2+}]}{20} - \frac{[Mg^{2+}]}{12} \right) \tag{7}$$

$$\text{Excess NH}_x = \text{Total NH}_x - \text{Required NH}_x \tag{8}$$

where $[NH_4^+]$, $[Na^+]$, $[K^+]$, $[Ca^{2+}]$, $[Mg^{2+}]$, $[SO_4^{2-}]$, $[NO_3^-]$, and $[Cl^-]$ are the measured concentrations ($\mu g\ m^{-3}$) of these ions, and $[HNO_3]$, $[HCl]$, and $[NH_3]$ are the mixing ratios (ppb) of these gases. If Excess $NH_x > 0$ (Total $NH_x >$ Required $NH_x$), the system is considered to be $NH_x$-rich. If Excess $NH_x < 0$ (Total $NH_x <$ Required $NH_x$), the system is considered to be $NH_x$-poor. The field measurements in this and previous studies (Wang et al., 2016; Liu et al., 2017a) collectively indicate that the Beijing winter haze fine particles are nearly always in an $NH_x$-rich region, as shown in Fig. 2.

The inputs for the forward mode calculations involve the measured total (gas plus aerosol) concentrations of $NH_3$, $H_2SO_4$, HCl, $HNO_3$, $Na^+$, and $K^+$. For the reverse mode calculations, the inputs involve the measured aerosol phase concentrations of $NH_4^+$, $SO_4^{2-}$, $NO_3^-$, $Cl^-$, $Na^+$, and $K^+$. Note that for E-AIM, the measured concentrations of $K^+$ are accounted for as equivalent $Na^+$. Note also that gaseous HCl concentrations are taken as zero in the forward mode calculations since a large proportion of HCl data is unavailable. But this treatment only has a very small effect owing to the low concentrations of gaseous HCl. Model calculations are limited to hourly samples meeting the following criteria: (1) $SO_4^{2-}$, $NO_3^-$, $Cl^-$, $NH_4^+$, and $NH_3$ (only for the

forward mode) are available, and (2) RH > 20%. The number of eligible samples for ISORROPIA calculations is about three hundred, whereas this number for E-AIM is only about one hundred since version IV additionally requires RH > 60%. Moreover, for the ISORROPIA forward mode pH calculations, we adopt a Monte Carlo approach to account for the measurement uncertainties of model inputs, including concentrations of ions and gases (uncertainties described in Sect. 2.1),

values of temperature (maximum–minimum range of 2 °C), and values of relative humidity (maximum–minimum range of 10%). All of these variables are assumed to follow a uniform distribution and their values are selected randomly and calculated 5000 times for each hourly sample.

## 3 Results and discussion

### 3.1 Reverse mode calculations

Figure 3 presents the relationship between ion balance and predicted pH values for $PM_{2.5}$ from four model calculations (i.e., ISORROPIA forward metastable, ISORROPIA reverse metastable, E-AIM forward, and E-AIM reverse), as well as a comparison of measured and predicted gas phase $NH_3$ mixing ratios. As shown in Fig. 3a, a good correlation ($r = 0.98$, $n = 106$) is found between the measured cation and anion concentrations and the average cation-to-anion equivalent ratio ($0.99 \pm 0.18$) is close to unity (similar to previous North China winter haze studies, see Table 1). Figure 3b shows that the reverse

mode pH values (both E-AIM and ISORROPIA) are highly sensitive to whether the ion balance is positive ($n = 61$) or negative ($n = 45$). The samples with negative ion balance (cations < anions) usually project pH values below 2 (highly acidic), whereas those with positive ion balance (cations > anions) are identified with pH values above 7.4 (neutral or basic). These features have been demonstrated also by Hennigan et al. (2015) and Murphy et al. (2017) using different observational datasets. Since the inputs to reverse mode calculations include only aerosol phase measurements of ions, the predicted pH values depend

largely on the ion balance (Hennigan et al., 2015). On the other hand, the pH values calculated using the forward mode (both E-AIM and ISORROPIA) range from 3.5 to 5.3 and are not as sensitive to the ion balance. This is because the forward mode calculations account for additional constraints imposed by the partitioning of semi-volatile species. The small difference in pH values between the forward mode E-AIM and ISORROPIA calculations is discussed in Sect. 3.2.1. The agreement between the measured and predicted gas phase concentrations of semi-volatile species serves usually as verification of the accuracy of

thermodynamic calculations. Figures 3c–d compare measured mixing ratios of $NH_3$ with model outputs. Good agreement is found for the forward mode, but the reverse mode calculations predict either implausibly high (> 1 ppm) or implausibly low (< 1 ppb) values of $NH_3$, when compared with measurements. The equilibrium partial pressures of $NH_3$ for the reverse mode are computed based on the predicted pH values with fixed aerosol $NH_4^+$ concentrations, and hence the extremely large biases of $NH_3$ reflect a significant deviation of pH with respect to the real values. Similar behavior is found for gas phase $HNO_3$ and

HCl (Fig. S5).

The above results suggest that the reverse mode calculations (only using aerosol quantity as model inputs) are strongly affected by the ion balance and hence the ionic measurement errors are very likely to lead to unreliable estimates of particle pH (Hennigan et al., 2015; Murphy et al., 2017). Furthermore, an equivalent ratio of near unity may not indicate that fine particles of winter haze have a pH of around 7 or close to 7 (Cheng et al., 2016; Wang et al., 2016; Ma et al., 2017). The forward mode calculations (using gas + aerosol quantity as model inputs) are affected much less by the measurement errors and should be used to predict the pH for winter haze particles.

## 3.2 Forward mode calculations

### 3.2.1 E-AIM vs. ISORROPIA

As shown in Fig. 3b, the pH values predicted by the forward mode E-AIM and ISORROPIA calculations differ slightly. Their pH difference, $\Delta$pH (ISORROPIA − E-AIM), can be expressed in Eq. (9):

$$\Delta\text{pH} = \text{pH}_\text{I} - \text{pH}_\text{E} = -\Delta \log_{10} m_{\text{H}^+} + \log_{10} \gamma_{\text{H}^+} \tag{9}$$

where $\text{pH}_\text{I}$ and $\text{pH}_\text{E}$ represent the pH predicted by ISORROPIA and E-AIM, respectively. $\Delta$pH can be considered to consist of two parts: difference in their estimated $\text{H}^+$ concentrations, denoted as $-\Delta \log_{10} m_{\text{H}^+}$, and difference in $\text{H}^+$ activity coefficients, which equals to the estimated $\log_{10} \gamma_{\text{H}^+}$ by E-AIM (since $\gamma_{\text{H}^+}$ is assumed to be unity in ISORROPIA). The forward mode model calculations using our field measurements in Beijing suggest, when RH varies from 70% to 90%, that $\Delta$pH is greater than zero, and negatively correlated with RH (Fig. 4a). $\Delta$pH would approach approximately zero, if this relationship were maintained as RH approaches 100%. Similarly, Liu et al. (2017a) found that pH values in ISORROPIA were on average 0.3 unit higher than E-AIM under winter haze conditions. Figure 4b indicates that $\Delta$pH is related to the differences in both $\text{H}^+$ concentrations and activity coefficients.

Since E-AIM (version IV) cannot be used when RH is below 60%, we conduct calculations using E-AIM (version II), in order to examine whether this relationship between $\Delta$pH and RH still holds at relatively low RH values (Fig. S6 in the Supplement). We find, under typical Beijing winter haze conditions ($\text{NH}_x$-rich), that pH predicted by ISORROPIA is systematically higher than E-AIM, and $\Delta$pH is negatively related with RH, when RH varies from 30% to 90%. The exact factors contributing to $\Delta$pH remain unclear, because these two thermodynamic models differ in many ways (e.g., their methods in calculating the activity coefficients for $\text{H}^+$ and the other ionic species and in estimating aerosol water contents). Note that $\Delta$pH is less than one unit for the cases tested in this study, which is much smaller than the pH discrepancy reported in previous winter haze studies (up to 8 pH units). Note also that the above analysis is based on the data sets collected in Beijing winter and may not apply to other conditions.

### 3.2.2 S curves of semi-volatile species

The S curves of ammonia, nitric acid, and hydrochloric acid describe the relationship between particle pH and their equilibrium fractions in the aqueous phase ($\varepsilon(NH_4^+) = [NH_4^+]/([NH_4^+] + [NH_3])$, $\varepsilon(NO_3^-) = [NO_3^-]/([NO_3^-] + [HNO_3])$, and $\varepsilon(Cl^-) = [Cl^-]/([Cl^-] + [HCl])$) at a given temperature and aerosol water content (AWC), assuming ideal solutions (water activity and all activity coefficients equal to unity). The S curves have been shown as useful tools to qualitatively and conceptually estimate particle pH (Guo et al., 2017a), and are calculated in this study using Henry's law constants and acid–base dissociation constants for each semi-volatile species (details in the Supplement, Sect. S3). We choose very humid conditions (RH > 75%) in the field measurements when particles are most likely in a completely aqueous phase, under which the average temperature and AWC (predicted by ISORROPIA) are 278 K and 144 µg m$^{-3}$, respectively. As shown in Fig. 5, the calculated $\varepsilon(NH_4^+)$ increases with pH whereas $\varepsilon(NO_3^-)$ and $\varepsilon(Cl^-)$ decrease with pH. The field measurements suggest that about a half of the total ammonia resides in the condensed phase ($\varepsilon(NH_4^+) = 54\% \pm 12\%$), and that almost all of the total nitric acid and hydrochloric acid are in the condensed phase ($\varepsilon(NO_3^-) = 99.6\% \pm 0.1\%$ and $\varepsilon(Cl^-) = 98.1\% \pm 0.7\%$). Thus, the ammonia S curve and the measured $\varepsilon(NH_4^+)$ suggest that the particle pH should be around 4 and is unlikely to exceed 5.5 when $\varepsilon(NH_4^+) < 1\%$ or below 1.5 when $\varepsilon(NH_4^+) > 99\%$. The S curves for nitric acid and hydrochloric acid and the measured $\varepsilon(NO_3^-)$ and $\varepsilon(Cl^-)$ also suggest that pH should be greater than 2, as $\varepsilon(NO_3^-)$ and $\varepsilon(Cl^-)$ become close to unity and are consequently insensitive to pH. Note that the assumption of ideal solutions is applied in the above analysis of the S curves. Thermodynamic equilibrium models can calculate the values of activity coefficients (and thus consider the non-ideality of solutions) and are therefore able to provide more quantitative results for particle pH compared to the S curves. As shown in Fig. 5, the forward mode ISORROPIA and E-AIM calculations predict similar average pH values (4.1 and 4.6 respectively), compared to the average number of 3.6 inferred from the S curve of ammonia. Note that calculations using the standard ISORROPIA model with the stable state assumption obtain an unrealistic average pH value of 7.7 due to its coding errors.

### 3.2.3 Driving factors for particle pH

It has been suggested that ambient RH plays an important role in the evolution of winter haze events (Sun et al., 2013; Wang et al., 2014; Tie et al., 2017) and the phase state of aerosols (Liu et al., 2017b). Thus, we present the pH and AWC values for PM$_{2.5}$ predicted by the ISORROPIA forward mode calculations (in both metastable and stable states) as a function of RH (Figs. 6a–b). Note that the revised ISORROPIA model is used for the stable state calculations.

Several previous studies have indicated that the values of AWC predicted by ISORROPIA are in reasonable agreement with those based on measurements of aerosol light scattering coefficients and hygroscopic growth factors (Bian et al., 2014; Guo et al., 2015; Tan et al., 2017; Wu et al., 2018). The predicted AWC increases with RH, and is greater for the metastable state (a completely aqueous solution). The absolute difference of AWC between the two states is minor at either high (> 70%) or low (< 40%) RH but is large at intermediate RH. Most inorganic species deliquesce at RH below 70% and, at a higher RH, particles

are liquid for both states. At a very low RH, particles are solid in the stable state (thus the AWC is zero and no prediction of pH is given), but can absorb a small amount of water if they are in the metastable state.

As shown in Fig. 6a and Fig. S7 in the Supplement, the pH values for these two phase states are very similar (ranging from 4 to 5) and of a moderately acidic nature for a wide range of RH. The results are consistent with the qualitative understanding of particle pH inferred from the S curves. Thermodynamic model calculations with either stable or metastable state assumption can provide reasonable estimates of aerosol water pH, at least for the winter haze conditions considered in this study. The pH values predicted by the AMS $PM_1$ measurements and forward mode ISORROPIA calculations (Fig. S8 in the Supplement) are about 0.2 unit lower than those of $PM_{2.5}$, due partly to lack of crustal ions ($Na^+$ and $K^+$) for the AMS $PM_1$ measurements.

By analyzing the sensitivity of pH to ammonia concentrations, recent studies have emphasized the important role of ammonia in determining winter haze particle pH (Guo et al., 2017b; Liu et al., 2017a). It was suggested, under ammonia-rich conditions, that a 10-fold increase in gas phase $NH_3$ concentrations roughly corresponds to one unit increase in pH (i.e., a 10-fold decrease in $H^+$ activity) (Guo et al., 2017b). This is obvious, since the equilibrium of dissolution and dissociation of ammonia in water can be expressed as: $NH_{3(g)} + H^+_{(aq)} \leftrightarrow NH^+_{4(aq)}$. These sensitivity tests have also indicated that atmospheric relevant ammonia concentrations are not high enough to achieve a fully neutralized condition (pH of around 7) for aerosol particles (Guo et al., 2017b; Liu et al., 2017a). The sensitivity tests conducted in this study are consistent with these previous studies (Fig. S9 in the Supplement).

The ambient RH has a minor effect on the predicted pH values. An insignificant ($p = 0.14$) increasing trend is calculated from the ISORROPIA metastable analysis in Fig. 6a: pH $= 0.01 \times$ RH (%) $+ 3.9$. A more detailed sensitivity analysis suggests that pH from ISORROPIA is insensitive to RH and the variability of pH is less than 0.3 unit within a RH range of 30%–90% (Fig. S9 in the Supplement). On the other hand, the pH predicted by E-AIM increases by 0.8 unit when RH increases from 30% to 90%, reflecting the systematic difference in pH between ISORROPIA and E-AIM discussed in Sect. 3.2.1.

Our calculations have shown that Beijing winter haze particles are moderately acidic with pH values ranging from 4 to 5 for a wide range of RH. As shown in Fig. 6c, our field measurements indicate that the concentrations of CO and total $NH_x$ (the sum of gas phase $NH_3$ and aerosol phase $NH_4^+$) exhibit similar increasing trends with ambient RH. CO is usually considered as an inactive chemical species during rapid haze formation and its enhancement with increased RH may be taken to reflect the accumulation of primary pollutants in the shallower boundary layer (Tie et al., 2017). Thus, the total $NH_x$ is accumulated as a primary pollutant and undergoes a considerable gas-to-particle conversion (Wang et al., 2015; Wei et al., 2015). The $NH_3$ to $NH_4^+$ ratio decreases from about 3 at a RH of 30% to about 1 at a RH of 80%. However, relatively high levels of $NH_3$ remain in the gas phase throughout the range of RH considered here. Note that the measured gas phase $HNO_3$ and $HCl$ mixing ratios were very low. Based on the above evidence, we suggest that the amount of total $NH_x$ is rich enough so as to balance most of

the $HNO_3$, $H_2SO_4$, and $HCl$ formed in gas and particle phases (Fig. 6d). Comparable $NH_3$ levels have been measured at multiple sites over the NCP in winter (Wang et al., 2016; Xu et al., 2016; Zhao et al., 2016; Liu et al., 2017a), indicating that ammonia-rich conditions are common. Thus, we suggest that ammonia may strongly affect particle pH over the NCP from clean to hazy conditions (low to high RHs). The sources of ammonia include agriculture, fossil fuel use, and green space, and their contributions vary in different environments (Pan et al., 2016; Sun et al., 2017; Teng et al., 2017; Zhang et al., 2017). We note however that the similar behavior of CO and total $NH_x$ does not imply necessarily similar emission sources.

### 3.3 Summary of existing studies

In the analysis so far, we have examined the impacts of different thermodynamic models (E-AIM vs. ISORROPIA), model configurations (forward vs. reverse), and phase states (stable vs. metastable) on the estimation of fine particle pH using the data sets collected during 2014 winter in Beijing. Here we summarize and compare the existing fine particle pH studies of North China winter haze, highlighting the importance of using an appropriate thermodynamic modeling approach. Figure 7 presents predicted pH from these studies. Their experimental details are summarized in Table 1.

The average reverse mode pH reported by previous studies ranged from –1 to 6.2 (Cheng et al., 2011; He et al., 2012; Cheng et al., 2016; Tian et al., 2018). Our calculations show that the reverse mode pH has a bimodal distribution with peaks between –2 and 2 (highly acidic) and between 7 and 10 (basic), and is very sensitive to errors in ionic measurements (Sect. 3.1). In fact, this implies, for an observational study, that the average pH calculated from many individual samples can be any value between –2 and 10 for E-AIM (between –2 and 8 for ISORROPIA), depending on the number of samples with negative and positive ion balances. The mean pH values in our calculations are 5.4 and 4.2 for E-AIM and ISORROPIA, respectively, which happen to show a weakly acidic condition.

The forward mode pH is affected much less by measurement errors and exhibits thus a narrow unimodal distribution according to our calculations. The average pH values are 4.6 (95% confidence interval 4.0 to 5.1) and 4.0 (95% confidence interval 3.6 to 4.4) for ISORROPIA and E-AIM, respectively. It is essential to note that the revised ISORROPIA when running in the stable state yields an almost identical distribution as compared to ISORROPIA in the metastable state. The studies using the standard ISORROPIA model with the stable state assumption have predicted unrealistic pH values of around 7 and should be re-evaluated (Wang et al., 2016; He et al., 2017). Previous ISORROPIA calculations using the metastable state assumption obtained average pH values from 4.1 to 5.4 (Cheng et al., 2016; Guo et al., 2017b; He et al., 2017; Liu et al., 2017a; Shi et al., 2017; Tan et al., 2018), agreeing reasonably with our results (an average pH value of 4.6). The $Ca^{2+}$ and $Mg^{2+}$ concentrations were not measured in this study and a sensitivity test suggests that including these crustal cations in calculations would increase predicted pH values by about 0.1 unit (Fig. S10). Among these studies, the highest pH value of 5.4 was obtained in Beijing in 2013 January (Cheng et al., 2016) and may be related to two factors: the contribution of organics to AWC was considered

which might increase the pH values for about 0.1 unit, and the $NH_3$ concentrations estimated from an empirical relationship with $NO_x$ might be biased high (He et al., 2014; Pan et al., 2016).

The above comparison suggests that the large discrepancy in pH values (from about 0 to about 7) reported in previous studies of North China winter haze may be attributed primarily to differences in the applications of thermodynamic models. We suggest the use of the forward mode rather than the reverse mode in future studies, and the use of the revised ISORROPIA model when the stable state is assumed for particle phase. The appropriate applications of thermodynamic modeling indicate a moderately acidic condition (pH from about 4 to about 5) for fine particles in North China winter haze.

### 3.4 Assumptions and limitations of thermodynamic modeling

It is important to acknowledge that most thermodynamic equilibrium models, including E-AIM and ISORROPIA, incorporate a few basic assumptions. First, gas and particle phases are assumed to be equilibrated. This seems reasonable given that we use hourly measurement data and that the equilibration timescale for semi-volatile species between gas and submicron particles is estimated to be 15–30 min (Fountoukis et al., 2009). Second, the aerosol curvature effect on equilibrium partial pressures of semi-volatile species (also known as the Kelvin effect) is ignored. This should have a negligible impact on bulk properties as the effect is important only for particles with sizes smaller than 0.1 µm, a fraction that does not contribute significantly to the mass of $PM_{2.5}$ particles (Nenes et al., 1998). Third, aerosols are assumed to be internally mixed and are treated as bulk properties, meaning that all the particles have the same size and chemical composition (Nenes et al., 1998; Box and Box, 2015). Note that only the measured inorganic components are included in the above calculations, whereas organic compounds, which account for about 30–50% of total fine particle mass during winter haze events (Huang et al., 2014; Zhang et al., 2015; Tan et al., 2018), are not considered. Organics may affect particle pH in several ways: (1) by increasing the absorption of aerosol water; (2) by participating in the charge balance and modifying the activity coefficients of inorganic ions in the aqueous phase; and (3) by changing the aerosol phase state (liquid-liquid phase separation). Next, we will discuss these aspects.

(1) The contribution of organics to aerosol water is parameterized usually based on the hygroscopicity parameter $\kappa_{org}$ (Guo et al., 2015; Cheng et al., 2016):

$$W_{org} = OM \, \frac{\rho_w}{\rho_{org}} \frac{\kappa_{org}}{(100\%/RH - 1)} \tag{6}$$

where $W_{org}$ is the aerosol water associated with organics, OM is the mass concentration of organics, and $\rho_w$ ($1.0 \times 10^3$ kg m$^{-3}$) and $\rho_{org}$ ($1.4 \times 10^3$ kg m$^{-3}$) are the densities of water and organics, respectively. The average $\kappa_{org}$ of 0.06 was obtained from an earlier cloud condensation nuclei study in Beijing (Gunthe et al., 2011). Black carbon may also absorb water with an average $\kappa$ of 0.04 from Peng et al. (2017). Using the OM and BC concentrations measured by the AMS and a $PM_{2.5}/PM_1$ conversion factor of 1.6 (Zhao et al., 2017), we find that the aerosol water associated with these species is only about $14 \pm 3\%$ (median $\pm$ median absolute deviation) of that associated with inorganic salts, and thus has a very minor impact on the predicted pH values

of fine particles (Fig. S11). The particle pH values increase by $0.05 \pm 0.01$, consistent with Liu et al. (2017a). Even with a $\kappa_{org}$ of 0.2 (a potential upper limit) (Zhao et al., 2015), the change of particle pH ($0.13 \pm 0.03$) is still small.

(2) It has been suggested that organic compounds (e.g., amines and organic acid salts) may affect particle pH, especially under weakly acidic conditions (Hennigan et al., 2015). Due to lack of detailed measurements of these species, a thorough evaluation of their effect is difficult. Here, we conduct a sensitivity test using the E-AIM model including oxalate ($C_2O_4^{2-}$), which is the most abundant organic acid salt in $PM_{2.5}$ in winter Beijing (Huang et al., 2005; Wang et al., 2017) and is also one of the strongest organic acids (acid dissociation constants $pK_{a1} = 1.27$ and $pK_{a2} = 4.27$). Strong positive correlations are measured between oxalate and sulfate and their ratios are about 1–2% (Wang et al., 2017). Considering also the relative concentration levels of oxalate and other organic acid salts (Table S8), the oxalate concentration is set at 20% of sulfate in this test, which may represent the upper limit for the concentration of total organic acids. The pH values predicted by the forward mode E-AIM decrease only by $0.07 \pm 0.03$ when oxalate is included (Fig. S12), indicating that particle pH is not strongly affected by organic acids if the system is equilibrated. Note that E-AIM assumes that the organics in the aqueous solution do not affect the activity coefficients of inorganic ions (Clegg et al., 2001). Using the AIOMFAC model (*web.meteo.mcgill.ca/aiomfac*), Pye et al. (2018) have recently showed that the interaction of inorganic ions with water-soluble organic compounds resulted in a 0.1 unit increase in pH for aerosols in the southeast United States.

(3) As described earlier, during severe winter haze events at very high RH, the aerosol phase state should be liquid. However, we suggest that the particles very likely undergo liquid-liquid phase separation between the inorganic and organic components. This phase separation is believed to depend primarily on the oxygen-to-carbon (O/C) atomic ratio (a parameter used to roughly describe the oxidation state) of organic aerosols and occurs almost always when the O/C ratio < 0.5 (Guo et al., 2016; Freedman, 2017). The average O/C ratio calculated based on the AMS $PM_1$ measurements and updated calibrations by Canagaratna et al. (2015) is $0.4 \pm 0.1$ (Fig. S13), similar to values reported previously in winter Beijing (Hu et al., 2016; Sun et al., 2016). The effect of phase separation on pH values of these two liquid phases remains unclear. It has been suggested in a recent laboratory study that the pH of the organic-rich fraction under phase separation is about 0.4 unit higher than that for a fully mixed aqueous phase (Dallemagne et al., 2016).

The above discussion suggests that the assumptions and limitations implicit in the thermodynamic models may not lead to large biases in prediction of the bulk pH of fine particles in North China winter haze, which is supported by the reasonable agreement between the measured and predicted gas-particle partitioning of semi-volatile species such as ammonia.

## 4 Conclusions

This study suggests that the significant discrepancy of fine particle pH, ranging from about 0 (highly acidic) to about 7 (neutral), calculated in previous studies of North China winter haze is due primarily to differences in the ways in which the E-AIM and ISORROPIA thermodynamic equilibrium models have been applied. The reverse mode calculations (only using aerosol phase composition as inputs) lead to erroneous results of pH since they are strongly affected by ionic measurement errors (especially under ammonia-rich conditions), and therefore should be avoided in future winter haze studies. The forward mode calculations (using the total (gas plus aerosol phase) compositions as inputs) account for additional constraints imposed by the partitioning of semi-volatile species and are affected much less by the measurement errors, and therefore, should be used in future studies. The forward mode calculations in this and previous studies collectively indicate, during North China winter haze events, that aerosol particles are moderately acidic with pH values ranging from about 4 to about 5. The assumed particle phase state (stable or metastable) does not significantly affect the pH calculations of ISORROPIA after coding errors in its standard model being fixed. A few previous studies, in which the standard ISORROPIA model was used and the stable state was assumed, predicted unrealistic pH values of around 7, and should be re-evaluated. In agreement with previous studies, we confirm that ammonia plays an important role in determining particle pH under winter haze conditions in northern China.

## Data availability

The Windows stand-alone executable of the ISORROPIA model is available at *isorropia.eas.gatech.edu*. The web-based E-AIM model is available at *www.aim.env.uea.ac.uk*. The measurement data on gas and particle compositions and meteorology are available upon request to the authors. The revision for the forward stable state in the source codes of ISORROPIA made in this study is available at *wiki.seas.harvard.edu/geos-chem/index.php/ISORROPIA_II*.

## Competing interests

The authors declare that they have no conflict of interest.

## Acknowledgments

This study was supported by the Harvard Global Institute and the National Natural Science Foundation of China (91744207, 21625701). We thank Athanasios Nenes for helpful discussions and for providing the source codes of ISORROPIA, and Becky Alexander, Michael Battaglia Jr., Simon Clegg, Hongyu Guo, Daniel Jacob, Mingxu Liu, Pengfei Liu, Mario Molina, Rachel Silvern, Gehui Wang, Lin Zhang, and Guangjie Zheng for helpful discussions.

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

**Figures**

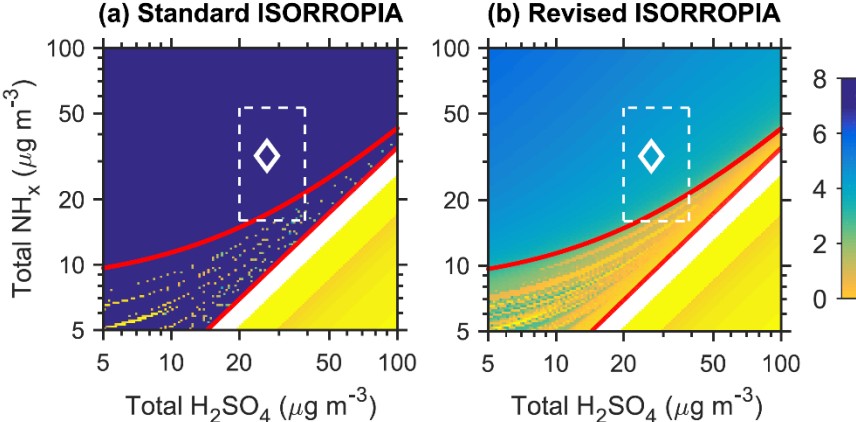

**Figure 1. Sensitivity of particle pH to the total (gas + aerosol) NH$_x$ and H$_2$SO$_4$ concentrations predicted by the standard and revised ISORROPIA model.** Model calculations are conducted in the forward mode with the stable state assumption. The red curves indicate the NH$_x$-rich (above the curve) and NH$_x$-poor (below the curve) regions. The red straight lines are used to distinguish the different subregimes in the ISORROPIA solution domain (G1 above the line, and I3 and J3 below the line, details in the supplement, Sect. S1). The input data (total Na = 0 µg m$^{-3}$, total HNO$_3$ = 26 µg m$^{-3}$, total HCl = 1.7 µg m$^{-3}$, RH = 56%, and T = 274.1 K) of an NH$_3$–Na–H$_2$SO$_4$–HNO$_3$–HCl–H$_2$O aerosol system reflect the average PM$_1$ measurements for Beijing winter haze pollution episodes reported in Wang et al. (2016). The white boxes define the observed concentration ranges for the Beijing winter haze pollution episodes and diamonds represent the average Beijing haze conditions (total NH$_x$ = 32 µg m$^{-3}$, total H$_2$SO$_4$ = 26 µg m$^{-3}$) reported in Wang et al. (2016). The noises in pH between the two red lines are due very likely to the instability of numerical solver currently used in the ISORROPIA model (more information can be found at *wiki.seas.harvard.edu/geos-chem/index.php/ISORROPIA_II*).

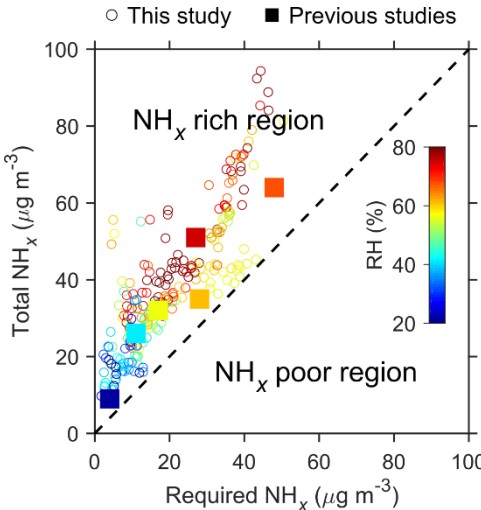

**Figure 2. Relationship between required NH$_x$ and total NH$_x$ concentrations under Beijing winter haze conditions.** The circles indicate hourly measurements in this study and the filled squares indicate measurement results from previous studies (Wang et al., 2016; Liu et al., 2017a) in winter Beijing. The dash line indicates a 1:1 relationship and defines the NH$_x$-rich region (above) and NH$_x$-poor region (below).

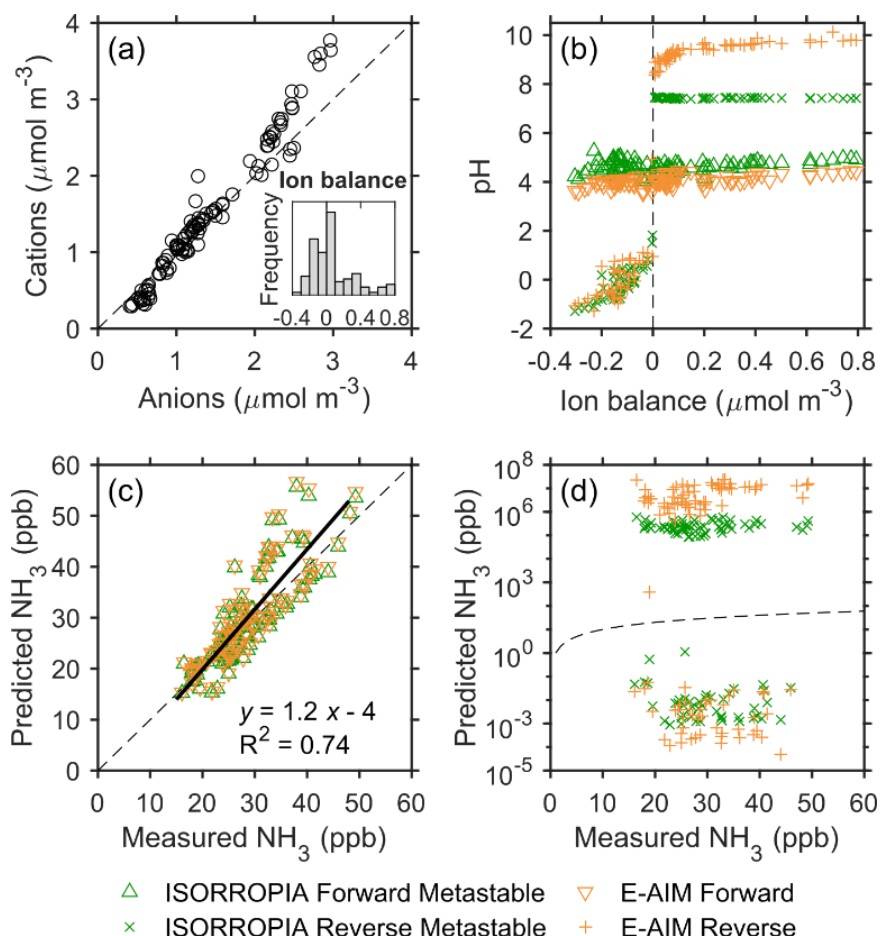

**Figure 3. Relationship between ion balance and predicted particle pH in forward mode and reverse mode calculations, and a comparison of measured and predicted gas phase NH₃ mixing ratios.** (a) Cation-to-anion equivalent ratios in PM₂.₅ during the field measurements. The inserted figure displays the frequency distribution of ion balance values. The dash line indicates a 1:1 relationship. (b) Predicted pH vs. ion balance. The dash line indicates ion balance equal to zero. (c–d) Comparisons of predicted and measured gas phase NH₃ mixing ratios. The dash lines indicate a 1:1 relationship. The solid line in (c) represents the linear correlation between predicted and measured NH₃ levels (the ISORROPIA forward metastable and E-AIM forward calculations have essentially the same results). The eligible number of samples ($n = 106$) is limited by the requirement in E-AIM that RH > 60%.

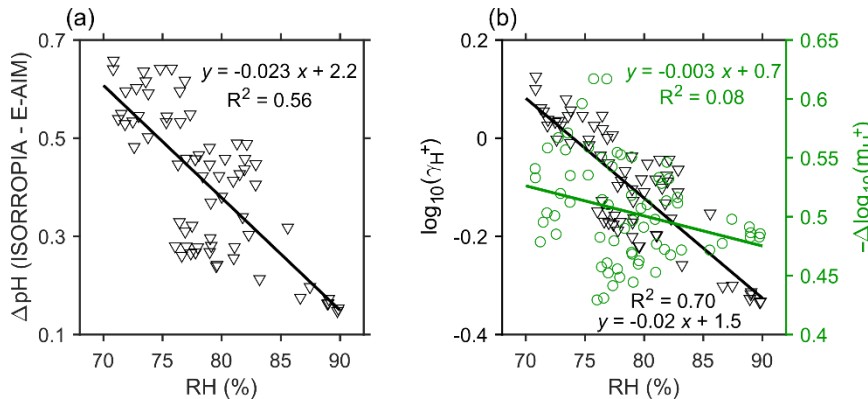

**Figure 4. pH difference between forward mode E-AIM and ISORROPIA calculations and its relationship with RH.** (a) $\Delta$pH vs. RH. (b) The relationship between RH and hydrogen ion activity coefficient from E-AIM (left) and difference in predicted H$^+$ concentrations (right). The solid lines indicate linear regressions. For a more appropriate comparison, we choose the samples ($n = 68$) that are in a completely aqueous phase predicted by E-AIM, and K$^+$ is accounted for as equivalent Na$^+$ in ISORROPIA. The ISORROPIA calculations are carried out assuming a metastable state, but the pH results are very similar for the stable state when the revised ISORROPIA model is applied.

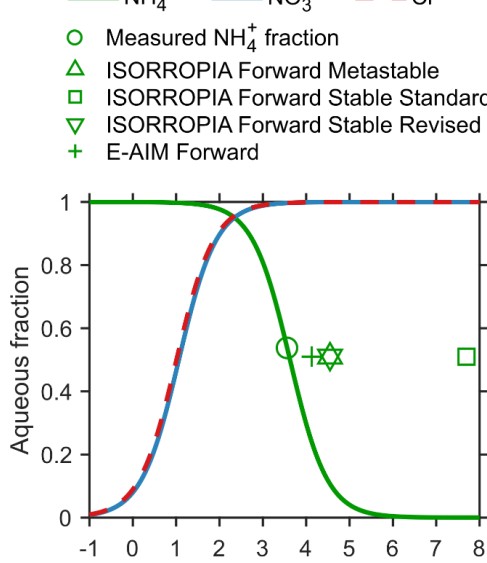

**Figure 5. Equilibrium fraction of total ammonia, nitric acid, and hydrochloric acid in the aqueous phase as a function of particle pH.** The average temperature (278 K) and aerosol water content ($144\ \mu g\ m^{-3}$) during severe haze conditions (RH > 75%) are used to calculate these S curves. The circle on top of the ammonia curve indicates the measured average aqueous fraction, which is calculated with the gas phase $NH_3$ and $PM_{2.5}$ $NH_4^+$ concentrations. The corresponding results from different model calculations are also shown as scatter plots: the $x$ axis is the calculated average pH value and the $y$ axis is the calculated average $NH_4^+$ aqueous fraction.

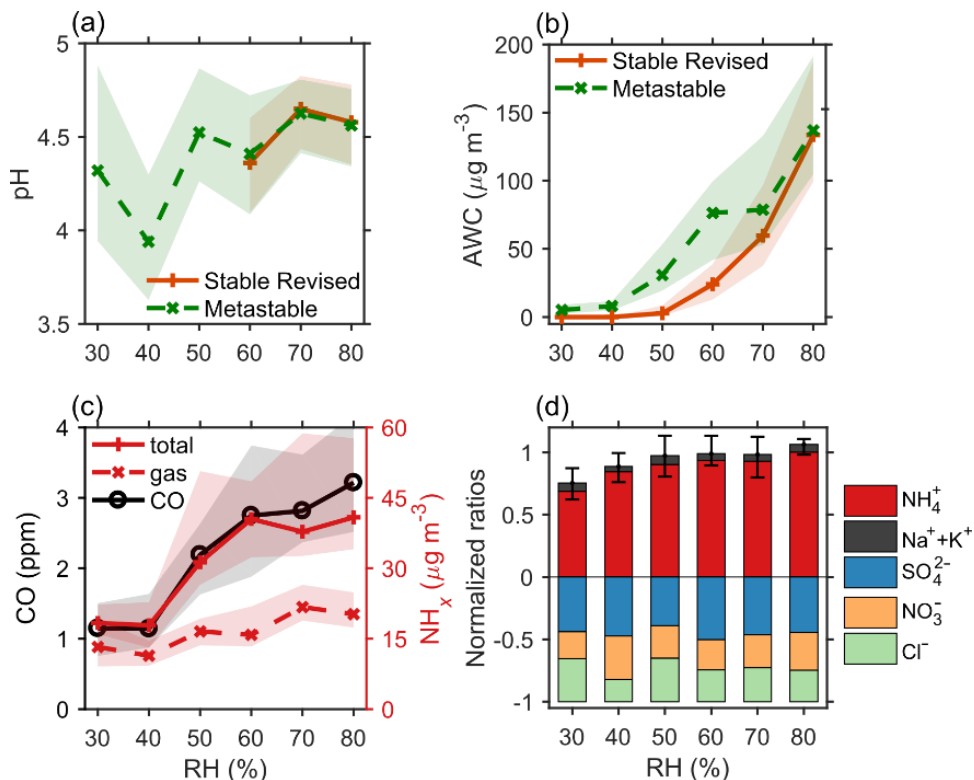

**Figure 6. Variations of several chemical and physical parameters as a function of RH.** (a–b) PM$_{2.5}$ pH and AWC predicted from forward mode ISORROPIA calculations in both stable and metastable states. (c) Measured concentrations of CO, total (gas + aerosol) NH$_x$, and gas phase NH$_3$. For NH$_3$, 1 µg m$^{-3}$ ≈ 1.3 ppb at standard temperature and pressure. (d) Equivalent ratios of different ions normalized by the levels of total anions. The data are grouped in RH bins (10% increment). The shaded regions in (a–c) and error bars in (d) indicate the 25$^{th}$ and 75$^{th}$ percentiles. The measurement uncertainties of ions and gases are considered in pH and AWC calculations using a Monte Carlo approach.

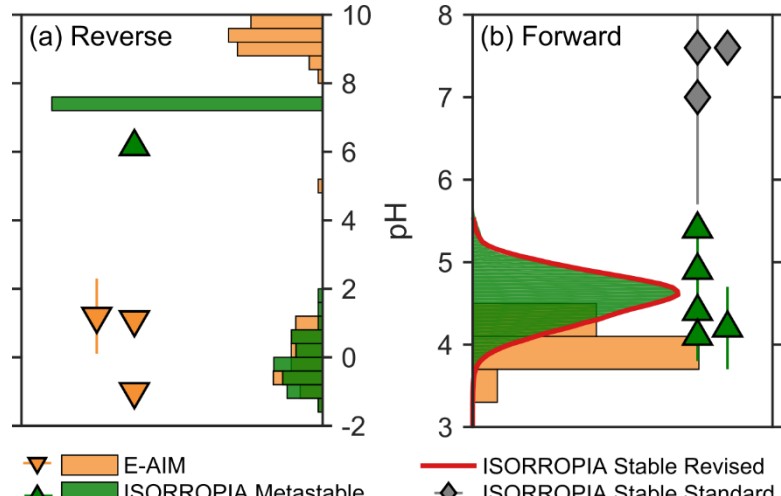

**Figure 7. pH predictions during North China winter haze events from this and previous studies.** Note differences in pH scales for the reverse mode (a) and forward mode (b) calculations. The frequency distributions and symbols reflect results from this and previous studies, respectively. Here we include only our samples during winter haze events (RH > 60%), with a Monte Carlo approach used in the ISORROPIA forward mode calculations to better account for the ionic and gas measurement uncertainties.

# Tables

**Table 1. Previously reported cation-to-anion equivalent ratios and particle pH values during winter haze periods in North China.**

| City | Year | Time Resolution | Size | Model | Equivalent Ratio | pH | Note | Reference |
|------|------|-----------------|------|-------|------------------|-----|------|-----------|
| *Forward (closed)* | | | | | | | | |
| Beijing | 2015 | 1 h | $PM_1$ | ISORROPIA Stable[c] | 1.09±0.11 | 7.6±0.0 | $PM_{2.5}$=114±44 µg m$^{-3}$; RH=56±14% | Wang et al. (2016) |
| Beijing[a] | 2014/2015 | 12 h | $PM_{2.5}$ | ISORROPIA Stable | 1.16 | 7.6±0.1 | $PM_{2.5}$>75 µg m$^{-3}$; RH=62±12% | He et al. (2017) |
| Xi'an | 2012 | 1 h | $PM_{2.5}$ | ISORROPIA Stable[c] | 1.06±0.06 | 7.0±1.3 | $PM_{2.5}$=250±120 µg m$^{-3}$; RH=68±14% | Wang et al. (2016) |
| Beijing | 2013 | 2 h | $PM_{2.5}$ | ISORROPIA Metastable | 1.08 | 5.4 | Average of January[e] | Cheng et al. (2016) |
| Beijing | 2015/2016 | 1 h | $PM_{2.5}$ | ISORROPIA Metastable | 0.99 | 4.2±0.5 | RH=68±16% | Liu et al. (2017a) |
| Beijing[a] | 2014/2015 | 12 h | $PM_{2.5}$ | ISORROPIA Metastable | 1.16 | 4.4±0.6 | $PM_{2.5}$>75 µg/m$^{-3}$; RH=62±12% | He et al. (2017) |
| Beijing | 2014 | 1 d | $PM_{2.5}$ | ISORROPIA Metastable | 1.2 | 4.1 | $PM_{2.5}$>150 µg m$^{-3}$ | Tan et al. (2018) |
| Tianjin | 2014/2015 | 1 h | $PM_{2.5}$ | ISORROPIA Metastable | 1.13 | 4.9±0.4 | RH=72±10% | Shi et al. (2017) |
| *Reverse (open)* | | | | | | | | |
| Beijing | 2013 | 2 h | $PM_{2.5}$ | ISORROPIA Metastable | 1.08 | 6.2 | Average of January[e] | Cheng et al. (2016) |
| Beijing | 2013 | 1 d | $PM_{2.1}$ | E-AIM | 1.16 | 1.1 | $PM_{2.5}$>150 µg m$^{-3}$; RH>60% | Tian et al. (2018) |
| Beijing[b] | 2005/2006 | 7 d | $PM_{2.5}$ | E-AIM[d] | 1.09 | 1.2±1.1 | RH=63±15% | He et al. (2012) |
| Jinan | 2006/2007 | 1 d | $PM_1$ | E-AIM[d] | Not Available | −1 | $PM_{1.8}$=193 µg m$^{-3}$ | Cheng et al. (2011) |

All measurements were made in the urban region except [a]suburban and [b]both urban and rural. Thermodynamic models were ISORROPIA (version II) or E-AIM (version IV) except for [d]E-AIM version II. [c]Personal communication with G. Wang. [e]Personal communication with G. Zheng.