# Peer review of "Fine particle pH for Beijing winter haze as inferred from different thermodynamic equilibrium models"

_Atmospheric Chemistry and Physics, 2018_

## Short Comment (SC1) · 8 Feb 2018

There is a mistake in this ACPD discussion paper about the unit of ionic strength. The ionic strength is calculated on a molality basis and its unit should be mol kg-1 water rather than M (mol L-1, on a molarity basis). All the values of ionic strength in Section 3.2.1, Figure 2, and Figure S6 have the unit of mol kg-1 water.

Thank Prof. Armistead Russell and Ms. Xing Peng at Georgia Tech very much for drawing my attention to this.

Shaojie Song songs@seas.harvard.edu

---

## Referee Comment (RC1) · Anonymous Referee #1 · 10 Feb 2018

This manuscript needs minor reversion before acceptance for publication. (1) As the authors stated, some papers reported the aerosol pH in Beijing, I suggest it should be declared clearly what are the correct results in the abstract and conclusion, also to let the readers know, which method is correct to estimate a reasonable aerosol pH. (2) The "forward stable" module was modified by the authors, although I did not read it in detail, the results seem more reasonable than previous runs. If possible, please contact GIT group to confirm it. (3) Some recent studies declared the aerosol pH could be close to 7 due to the high ammonia level, please use the sensitivity test to show if it is possible. The implications of aerosol pH should be very important for atmospheric reactions. (4) Please have some discussion on the aerosol water content effects on pH

values especially when RH was high.
* * *

---

## Short Comment (SC2) · 13 Feb 2018

**Comment on Song et al. (by A.Nenes, H.Guo, A.Russell and R.Weber)**

We would like to commend Song et al. for their extensive analysis that goes deep into the model code and data. The importance of understanding aerosol pH is key to understanding of aerosol growth and impacts, as has been demonstrated in a growing body of literature. This literature, however, also exposes knowledge gaps. Following are some comments and thoughts about the analysis that in our opinion require attention, especially on the impact and importance of the $H^+$ activity coefficient. Addressing these points, may require considerable rewriting and refocusing of the paper, but we

feel it will eventually substantially enhance the contribution.

**Algorithm changes to ISORROPIA-II routines.**

We would like to thank the authors for their very detailed explanation of the pH calculation issue, and the resolution provided. This clearly shows the value of having open source codes so that they are continuously used and tested by the community. The alternative approach in the standard code was used in the routines identified, because loss in precision in calculating the SQRT function (at low concentrations of aerosol precursors and when solid precipitates formed, e.g., $NH_4Cl$), made partitioning calculations at times inaccurate and noisy. Although the alternative approach captured partitioning, pH was clearly not, so adopting the standard calculation approach used in the subcases with higher RH values is appropriate; however, provision still needs to be shown to avoid loss of precision (e.g., Taylor expansion approximations or renormalization instead of SQRT). We will address this in the upcoming version of ISORROPIA-II.

**Application of thermodynamic models when interpreting data.**

We were very pleased to see that the analysis of Beijing data carried out here fully supports our prior work on how to use observational data to constrain pH, namely: $i)$ avoiding usage of molar ratios and ion balances as pH proxies (Guo et al., 2015; Hennigan et al., 2015; Guo et al., 2016; Weber et al., 2016; Guo et al., 2017b), and, $ii)$ the large pH errors that can result when aerosol-only concentrations from observations are used in open-system thermodynamic calculations (i.e., "reverse mode" calculations that are not subject to a global constraint of mass balance (Pilinis et al., 2000; Hennigan et al., 2015)). It should also be noted that the secondary effect of water-soluble organics on aerosol pH is also consistent with the recent work of Pye et al. (2017).

One conclusion that the authors come to is that the pH calculations are not sensitive to the assumption of metastable and stable state. As presented, this can be misinterpreted by the reader that partitioning evaluations are not valuable for constraining aerosol pH. Partitioning calculations can sufficiently constrain pH, but only when

predictions of aerosol water and semivolatile partitioning (of $NH_3$-$NH_4^+$, $HNO_3$-$NO_3^-$, and HCl-$Cl^-$ if possible) are reproduced by observations (as shown in e.g., Guo et al. (2015) and other studies) – and a sufficient fraction of the partitioned aerosol species is associated with the aqueous phase. When aerosol water measurements are lacking or too uncertain, then showing that when **aqueous phase semivolatile partitioning by itself** (i.e., provided by the metastable solution) reproduces aerosol observations, aerosol pH is sufficiently constrained. **The pH values calculated for the metastable solution, for cases where partitioning is consistent with observations, provide the most plausible estimates of acidity.** pH values for the stable solution, especially when the liquid water content becomes very small (hence aqueous-phase partitioning a secondary contribution to the total partitioning), are subject to considerably more uncertainty – even if the pH corresponding to the metastable and stable solutions agree.

**Activity coefficient discussion**

The authors extensively comment on the usage of $\gamma_{H+} = 1$ in some of the calculations behind ISORROPIA-II. In fact, the assumption that $\gamma_{H+} = 1$ is thought to be a major source of pH discrepancy between ISORROPIA and E-AIM (it's even stated in the abstract). The data presented does not really support this for the following reasons:

1. $\gamma_{H+}$ varies by ±0.2 units over the ionic strengths considered (0-20), Figure 2b, while pH differences between the models are typically larger than 0.2 units.

2. The correlation of pH discrepancy between ISORROPIA-II (as calculated with the formula of Guo et al. (2015)) and E-AIM with $\gamma_{H+}$ *does not* indicate a causal relationship.

3. If $\gamma_{H+} = 1$ was indeed the reason for the discrepancy, then at an ionic strength of ~20, when $\gamma_{H+}$ ~1, the pH discrepancy between ISORROPIA and E-AIM should be zero (Figure 2b). This is not the case at all.

Considering points 1, 2, 3 together, one can actually conclude that about 0.2 pH units discrepancy between ISORROPIA-II and E-AIM may arise from the assumption of $\gamma_{H+} = 1$ for the RH (ionic strength) range considered, while the rest of the discrepancy may be related to the predicted concentration of $H^+$. This may even suggest that $\gamma_{H+} = 1$ is not a leading cause of discrepancy. In support of this, we find it very interesting that when one compares the $\gamma_{H+}$ values from E-AIM (Figure 2c) and from AIOMFAC (Figure S6), $\log(\gamma_{H+})$ differs by about 0.6 units at an ionic strength of 20 M (E-AIM gives 0.1 and AIOMFAC gives -0.5; note the -0.6 difference in $\log(\gamma_{H+})$ means $+0.6$ pH compared to E-AIM), which seems to be consistent with the 0.6 higher pH comparing ISORROPIA to E-AIM at the same ionic strength. Could it just be then that the calculation of $\gamma_{H+}$ by E-AIM is more uncertain than implied? The Beijing haze polluted period has an ionic strength close to 40 M, which brings $\gamma_{H+}$ close to 1 according to Figure S6. Assuming $\gamma_{H+} = 1$ to diagnose pH from ISORROPIA (single point) translates to an uncertainty of less than 0.5 pH units over a large range of RH or ionic strength (Figure S6).

There is a lack of discussion on the effects of the other ionic species as sources of discrepancy between E-AIM and ISORROPIA-II, which is surprising, given that E-AIM uses the single ion activity approach, while ISORROPIA uses mean activity coefficients of ion pairs. Predictions from the two types of activity coefficient models do show important differences (e.g., Kim et al., 1993). The configuration used in ISORROPIA-II (Kusik-Meisner binary activity coefficients with the Bromley mixing rule for multicomponent aerosol) has been shown to provide stable solutions for ionic strengths that far exceed 30, the limit where Pitzer coefficients have been shown to work well (Kim et al., 1993). The latter point of course is quite relevant for the discussion raised by the authors concerning the applicability of the activity coefficient models used by ISORROPIA-II and E-AIM when applied to the high ionic strengths corresponding to RH below 70%.

Given the above, unless a thorough analysis of how all the activity coefficients,

[Figure]

water uptake and equilibrium constants contribute to the pH differences between ISORROPIA-II and E-AIM, one cannot really state how much uncertainty in pH arises from the assumption of $\gamma_{H+} = 1$, though it appears to be bounded and much less than the difference in the predicted pH's between the two models. Perhaps it would be better to just plot the predicted particle phase fractions ($\varepsilon(NH_4^+)$ and $\varepsilon(NO_3^-)$ as a function of pH) by each model and compare them against the data (following the approach of Guo et al., 2017a). Then one will have a better sense of the pH uncertainty (given by the range between models) for a given value of (observed) $\varepsilon$.

***Specific (but important) comments***

- The authors note early in the manuscript that the discrepancy in calculated pH when assuming $\gamma_{H+} = 1$ can be multiple units. This is not supported by the supplementary figure ($\gamma_{H+}$ from AIOMFAC), where for an ionic strength range of 0-100, log $\gamma_{H+}$ (hence the contribution of assuming $\gamma_{H+} = 1$ to pH discrepancy) varies within 1 unit. This has always been, by the way, our view – so it is nice to see this confirmed by the analysis presented!

- Also noted throughout is that pH is overestimated when assuming $\gamma_{H+} = 1$. This is not always true as well;as noted by the supplementary figure ($\gamma_{H+}$ from AIOMFAC), pH can be underestimated or overestimated by assuming $\gamma_{H+} = 1$, but not more than half a unit. In fact, the average pH estimated by ISORROPIA-II is *actually lower* than that reported for E-AIM (4.2 vs 5.4, for ISORROPIA-II vs E-AIM respectively) inconsistent with pH trends stated above.

- In ISORROPIA-II, the non-ideal interactions of H+ with all the ions in solution (especially NO3, Cl, HSO4, SO4) is explicitly considered by the Kusik-Meisner and Bromely formulations. $\gamma_{H+} = 1$ is only invoked when the singe ion activity is required. This is not sufficiently noted in the text.

- The authors understandably treat NVC (i.e. Ca, Mg, K, Na) as equivalent sodium,

because E-AIM cannot explicitly treat Ca, Mg and K. The impact of this assumption can lead to important differences in the predicted thermodynamic state, owing to the strong nonideality of divalent ions and different water uptake characteristics of sodium salts vs. their other counterparts (e.g., Fountoukis et al., 2009).

- What constitutes a "large/important" and "small/minor" different in pH depends on the context in which the pH is used. Constraining "absolute" pH for ambient aerosol to within less than 0.5 units may prove to be extremely challenging (e.g., the difference in log $\gamma_{H+}$ between E-AIM and AIOMFAC, the effects of organics on activity and water uptake and so on); so it may most likely be necessary to use a consistent pH calculation method and thermodynamic model when comparing aerosol acidities between models and/or observations.

- The authors caution about the predictions of ISORROPIA-II in metastable mode (for RH below the mutual deliquescence point) owing to the large ionic strengths of the solutions. Although we agree the ionic strengths are high, literature supports that the activity coefficient models used in ISORROPIA-II are stable for ionic strengths above 30, a situation that is also not the case for the Pitzer method (Kim et al., 1993).

In closing, we very much appreciate the analysis and it demonstrates an increasing sophistication of which the community is both understanding and discussing the thermodynamics of aerosols and the important topic of aerosol acidity. We also hope that the comments provided here add insight that will considerably strengthen the paper, and provide ideas for future work on the important topic of aerosol acidity.

**References**

Fountoukis, C., Nenes, A., Sullivan, A., Weber, R., VanReken, T., Fischer, M., Matias, E., Moya, M. Farmer, D., and Cohen, R.: Thermodynamic characterization of Mexico City Aerosol during MILAGRO 2006, Atmos. Chem. Phys., 9, 2141-2156, 2009.

Guo, H., et al.: Fine-particle water and pH in the southeastern United States, Atmos. Chem. Phys., 15, 5211-5228, doi: 10.5194/acp-15-5211-2015, 2015.

Guo, H., et al.: Fine particle pH and the partitioning of nitric acid during winter in the northeastern United States, Journal of Geophysical Research: Atmospheres, 121, 10355-10376, doi: 10.1002/2016jd025311, 2016.

Guo, H., Liu, J., Froyd, K. D., Roberts, J. M., Veres, P. R., Hayes, P. L., Jimenez, J. L., Nenes, A., and Weber, R. J.: Fine particle pH and gas–particle phase partitioning of inorganic species in Pasadena, California, during the 2010 CalNex campaign, Atmos. Chem. Phys., 17, 5703-5719, doi: 10.5194/acp-17-5703-2017, 2017a.

Guo, H., Nenes, A., and Weber, R. J.: The underappreciated role of nonvolatile cations on aerosol ammonium-sulfate molar ratios, Atmospheric Chemistry and Physics Discussions, 1-19, doi: 10.5194/acp-2017-737, 2017b.

Hennigan, C. J., Izumi, J., Sullivan, A. P., Weber, R. J., and Nenes, A.: A critical evaluation of proxy methods used to estimate the acidity of atmospheric particles, Atmos. Chem. Phys., 15, 2775-2790, doi: 10.5194/acp-15-2775-2015, 2015.

Kim, Y.P., Seinfeld, J.H. and Saxena, P: Atmospheric gas-aerosol equilibrium. 2. Analysis of common approximations and activity-coefficient calculation methods, Aer.Sci.Tech., 19, 182-198, 1993

Pilinis, C., Capaldo, K.P., Nenes, A., Pandis, S.N.: MADM - A New Multicomponent Aerosol Dynamics Model, Aerosol Sci. Tech., 32(5), 482-502, 2000

Pye, H. O. T., et al.: On the implications of aerosol liquid water and phase separation for organic aerosol mass, Atmos. Chem. Phys., 17, 343-369, doi: 10.5194/acp-17-343-2017, 2017.

Weber, R. J., Guo, H., Russell, A. G., and Nenes, A.: High aerosol acidity despite declining atmospheric sulfate concentrations over the past 15 years, Nature Geoscience, 9, 282-285, doi: 10.1038/ngeo2665, 2016.

---

## Short Comment (SC3) · 14 Feb 2018

In our comment, the reference to Pye et al. (2017) should refer to the following publication:

Coupling of organic and inorganic aerosol systems and the effect on gas–particle partitioning in the southeastern US, Atmos. Chem. Phys., 18, 357-370, https://doi.org/10.5194/acp-18-357-2018, 2018.
* * *

---

## Referee Comment (RC2) · Anonymous Referee #2 · 12 Mar 2018

This paper provides insight into the acidity of aerosols in Beijing. Table 1 provides a nice summary of previously published values which range from very acidic ( -1) to basic (7.6). The paper uses ISORROPIA and E-AIM to estimate pH and provides discussion on how organic compounds may modify pH. The paper is well written, fairly thorough, and detailed. The Monte Carlo approach provides additional confidence in the results. A number of (mostly clarifying) comments are listed below. One role for organic compounds in modifying pH was missing from the discussion. Specifically, on page 11, the authors list three ways in which organics can modify pH: (1) adding aerosol water, (2) participating in charge balance (e.g. dissociation of organic acids), and (3) by changing the aerosol phase state. The third area could use clarification (see detailed

comments), but a fourth way (that seemed to be missing) is through modification of the chemical environment and therefore by modifying the activity coefficients of the inorganic species. This could be scoped out using the AIOMFAC model. The authors should also be more forceful in their statements regarding what is a reasonable pH calculation and what is likely erroneous (see specific comment 1).

Specific comments:

1. Page 1, line 29: The authors indicate reverse mode calculations "exhibit a bimodal distribution with peaks between -2 and 2 and between 7 and 10." This reads as if these peaks are plausible values. Consider adding "depending on whether cations or anions were in excess" to highlight that the bimodal values are artifacts.

2. Page 1, line 34-35 "The phase state assumed, which can be either stable (solid plus liquid) or metastable (only liquid), does not significantly impact pH predictions of ISORROPIA." Presumably this is true only at high RH? Figure 4a does not provide "stable" pH estimates below 60% RH and Figure 4b indicates the metastable and stable aerosol water differs (and is nonzero) between 40 and 70%.

3. Page 3, line 27-30. The collection efficiency of the AMS is known to be a function of the ammonium to sulfate ratio (e.g. Middlebrook et al., 2012 https://www.tandfonline.com/doi/pdf/10.1080/02786826.2011.620041). Was this factored in?

4. Page 4, line 12: What effects of organic compounds does E-AIM consider? Dissociation of acids? Does it treat the effects of organics on inorganic activity coefficients?

5. Page 5, equations: Add "charge equivalent" before "measured ion concentrations" to indicate that sulfate, Ca, Mg have been multiplied by 2.

6. Section 3.2.1 and Figure 2: Do E-AIM and ISORROPIA predict different H+ concentrations? To what degree? How much of the difference between ISORROPIA and E-AIM is due to including gamma_H+ different than 1 in reporting pH vs the activity

coefficient of H+ actually modifying the thermodynamics? In other words, if you plotted E-AIM and ISORROPIA and set the activity coefficient to 1 in both for plotting purposes only, what would the difference be?

7. Figure 3: Could ISORROPIA or E-AIM predictions be overlaid on the plot? What measurement technique is the measured NH4+ fraction from? Is it different than the AMS value?

8. Page 10, before section 3.4: Emphasize and clearly state what your best estimate of aerosol pH is

9. Page 11, line 6-8: See above comment about a missing organic modification to pH

10. Page 11, line 17: What fraction of the total aerosol water is due to organic compounds?

11. Page 12, near line 7. What is your hypothesis regarding liquid-liquid phase separation and the effect on pH? Isn't your default configuration essentially liquid-liquid phase separation into and organic-rich and inorganic-rich phase? This ties in with the fourth possible way organics affect pH (via activity coefficients if organic compounds coexist in the inorganic phase).

12. In the supporting information, can you provide the exact ISORROPIA file names and line numbers and what the content was modified

---

## Author Comment (AC3) · 12 Apr 2018

Correct reference has been applied.
* * *

---

## Author Response (AR1)

**Response to Anonymous Referee #1**

Song et al.

*Comments are in black and responses are in* *blue*.

This manuscript needs minor reversion before acceptance for publication.

Our responses to specific comments are provided below.

(1) As the authors stated, some papers reported the aerosol pH in Beijing, I suggest it should be declared clearly what are the correct results in the abstract and conclusion, also to let the readers know, which method is correct to estimate a reasonable aerosol pH.

The abstract and conclusion have been revised to highlight the appropriate applications of thermodynamic modeling and the reasonable range of aerosol water pH inferred from such method.

Revisions made in the manuscript:

> **"Abstract.** *pH is an important property of aerosol particles but is difficult to measure directly. Several studies have estimated the pH values for fine particles in North China winter haze using thermodynamic models (i.e., E-AIM and ISORROPIA) and ambient measurements. The reported pH values differ widely, ranging from close to 0 (highly acidic) to as high as 7 (neutral). In order to understand the reason for this discrepancy, we calculated pH values using these models with different assumptions with regard to model inputs and particle phase states. We find that the large discrepancy is due primarily to differences in the model assumptions adopted in previous studies. Calculations using only aerosol phase composition as inputs (i.e., reverse mode) are sensitive to the measurement errors of ionic species and inferred pH values exhibit a bimodal distribution with peaks between −2 and 2 and between 7 and 10, depending on whether anions or cations are in excess. Calculations using total (gas plus aerosol phase) measurements as inputs (i.e., forward mode) are affected much less by these measurement errors. In future studies, the reverse mode should be avoided whereas the forward mode should be used. Forward mode calculations in this and previous studies collectively indicate a moderately acidic condition (pH from about 4 to about 5) for fine particles in North China winter haze, indicating further that ammonia plays an important role in determining this property. The particle phase state assumed, either stable (solid plus liquid) or metastable (only liquid), does not significantly impact pH predictions. The unrealistic pH values of about 7 in a few previous studies (using the standard ISORROPIA model and stable state assumption) resulted from coding errors in the model, which have been identified and fixed in this study."*

> **"Conclusions.** *This study suggests that the significant discrepancy of fine particle pH, ranging from about 0 (highly acidic) to about 7 (neutral), calculated in previous studies of North China winter haze is due primarily to differences in the ways in which the E-AIM and ISORROPIA thermodynamic equilibrium models have been applied. The reverse mode calculations (only using aerosol phase composition as inputs) lead to erroneous results of pH since they are strongly affected by ionic measurement errors (especially under ammonia-rich conditions), and therefore should be avoided in future winter haze studies. The forward mode calculations (using the total (gas plus aerosol phase) compositions as inputs) account for additional constraints imposed by the partitioning of semi-volatile species and are affected much less by the measurement errors, and therefore, should be used in future studies. The forward mode calculations in this and previous studies collectively indicate, during North China winter haze events, that aerosol particles are moderately acidic with pH values ranging from about 4 to about 5. The assumed particle phase state (stable or metastable) does not significantly affect the pH calculations of ISORROPIA after coding errors in its standard model being fixed. A few previous studies, in which the standard ISORROPIA model was used and the stable state was assumed, predicted unrealistic pH values of around 7, and should be re-evaluated. In agreement with previous studies, we confirm that ammonia plays an important role in determining particle pH under winter haze conditions in northern China."*

(2) The "forward stable" module was modified by the authors, although I did not read it in detail, the results seem more reasonable than previous runs. If possible, please contact GIT group to confirm it.

Our modification of the ISORROPIA source code for the "forward stable" mode has been confirmed by its developers.

(3) Some recent studies declared the aerosol pH could be close to 7 due to the high ammonia level, please use the sensitivity test to show if it is possible. The implications of aerosol pH should be very important for atmospheric reactions.

The sensitivity of pH to ammonia concentration levels has been examined in a few previous winter haze studies (e.g., Liu et al., 2017 and Guo et al., 2017). Thus, in this study we cited their results in Sect. 3.2.3 in the revised manuscript and also provided a similar sensitivity test. Our conclusions are the same as those from these previous studies.

Revisions made in the manuscript:

> *"By analyzing the sensitivity of pH to ammonia concentrations, recent studies have emphasized the important role of ammonia in determining winter haze particle pH (Guo et al., 2017b; Liu et al., 2017a). It was suggested, under ammonia-rich conditions, that a 10-fold increase in gas phase $NH_3$ concentrations roughly corresponds to one unit increase in pH (i.e., a 10-fold decrease in $H^+$ activity) (Guo et al., 2017b). This is obvious, since the equilibrium of dissolution and dissociation of ammonia in water can be expressed as: $NH_{3(g)} + H^+_{(aq)} \leftrightarrow NH^+_{4(aq)}$. These sensitivity tests have also indicated that atmospheric relevant ammonia concentrations are not high enough to achieve a fully neutralized condition (pH of around 7) for aerosol particles (Guo et al., 2017b; Liu et al., 2017a). The sensitivity tests conducted in this study are consistent with these previous studies (Fig. S9 in the Supplement)."*

Liu, M., Song, Y., Zhou, T., Xu, Z., Yan, C., Zheng, M., Wu, Z., Hu, M., Wu, Y., and Zhu, T.: Fine particle pH during severe haze episodes in northern China, Geophys. Res. Lett., 44, 5213-5221, doi:10.1002/2017GL073210, 2017a.

Guo, H., Weber, R. J., and Nenes, A.: High levels of ammonia do not raise fine particle pH sufficiently to yield nitrogen oxide-dominated sulfate production, Sci. Rep., 7, 12109, doi:10.1038/s41598-017-11704-0, 2017b.

(4) Please have some discussion on the aerosol water content effects on pH values especially when RH was high.

In the revised manuscript, we have added more discussion in Sect. 3.2.3 on the effects of ammonia concentrations and RH on the pH values predicted by ISORROPIA and E-AIM. We think it does not make much sense to directly compare aerosol water content and pH values, because the aerosol water content predicted by the thermodynamic equilibrium models depends on the amount of chemical species and RH values in the input. If the input RH and excess ammonia remain constant, increasing the amount of chemical species would increase aerosol water content but would not change predicted pH values significantly. On the other hand, if the excess ammonia and the amount of chemical species in the input remain constant, increasing RH values would affect predicted pH values, but this effect is different for E-AIM and ISORROPIA. Thus we choose to evaluate the effect of changing RH on the predicted pH values.

Below is a new figure in the supplement of the revised manuscript. It shows the sensitivity of pH to $NH_x$ concentrations and RH. The model inputs include total $H_2SO_4$ = 30 µg m$^{-3}$, total $HNO_3$ = 51 µg m$^{-3}$, temperature = 278 K, and varying RH (from 30% to 90%) and total $NH_x$ concentrations (from 25 to 100 µg m$^{-3}$). The inputs are calculated from our field measurements during haze episodes (RH > 60%) as the average temperature and average concentrations of total $H_2SO_4$, and $HNO_3$. $Na^+$ and $K^+$ are accounted for as equivalent $NH_4^+$, and $Cl^-$ as equivalent $NO_3^-$. The required $NH_x$ concentrations calculated for the input total $H_2SO_4$ and $HNO_3$ concentrations are about 24 µg m$^{-3}$. The curves in panels (c and d) show the average pH in each bin for $NH_x$ concentrations or RH.

Panel (d) of this figure shows the relationship of RH and pH. We find that the pH from ISORROPIA is insensitive to RH and the variability of pH is less than 0.3 unit. On the other hand, pH from E-AIM increases by 0.8 unit with increasing RH from 30% to 90%. The reason for this difference in pH between ISORROPIA and E-AIM remains unclear, and is discussed briefly in Sect. 3.2.1.

[Figure]

**Response to Anonymous Referee #2**

Song et al.

*Comments are in black and responses are in blue.*

This paper provides insight into the acidity of aerosols in Beijing. Table 1 provides a nice summary of previously published values which range from very acidic (-1) to basic (7.6). The paper uses ISORROPIA and E-AIM to estimate pH and provides discussion on how organic compounds may modify pH. The paper is well written, fairly thorough, and detailed. The Monte Carlo approach provides additional confidence in the results. A number of (mostly clarifying) comments are listed below.

Thanks for these positive comments for our manuscript. Our responses to specific comments are provided below.

One role for organic compounds in modifying pH was missing from the discussion. Specifically, on page 11, the authors list three ways in which organics can modify pH: (1) adding aerosol water, (2) participating in charge balance (e.g. dissociation of organic acids), and (3) by changing the aerosol phase state. The third area could use clarification (see detailed comments), but a fourth way (that seemed to be missing) is through modification of the chemical environment and therefore by modifying the activity coefficients of the inorganic species. This could be scoped out using the AIOMFAC model.

It is a very good comment. E-AIM assumes that the inorganic ions and organic solutes do not influence each other. Their molality-based activity coefficients are thus equal to those calculated for the systems water + inorganic ions only, and water + organic solutes only (Clegg et al., 2001). In the revised manuscript, we add the fourth way in Sect. 3.4 when discussing the effect of organics on particle pH calculation. Specifically, we modified this sentence to "*Organics may affect particle pH in several ways: (1) by increasing the absorption of aerosol water; (2) by participating in the charge balance and modifying the activity coefficients of inorganic ions in the aqueous phase; and (3) by changing the aerosol phase state (liquid-liquid phase separation)*", and added "*Note that E-AIM assumes that the organics in the aqueous solution do not affect the activity coefficients of inorganic ions (Clegg et al., 2001). Using the AIOMFAC model (web.meteo.mcgill.ca/aiomfac), Pye et al. (2018) recently showed that the interaction of inorganic ions with water-soluble organic compounds resulted in a 0.1 unit increase in pH for aerosols in the southeast United States*".

S. L. Clegg, J. H. Seinfeld, and P. Brimblecombe (2001) Thermodynamic modelling of aqueous aerosols containing electrolytes and dissolved organic compounds. J. Aerosol Sci. 32, 713-738.

Pye, H. O. T., Zuend, A., Fry, J. L., Isaacman-VanWertz, G., Capps, S. L., Appel, K. W., Foroutan, H., Xu, L., Ng, N. L., and Goldstein, A. H.: Coupling of organic and inorganic aerosol systems and the effect on gas–particle partitioning in the southeastern US, Atmos. Chem. Phys., 18, 357-370, doi:10.5194/acp-18-357-2018, 2018.

The authors should also be more forceful in their statements regarding what is a reasonable pH calculation and what is likely erroneous (see specific comment 1).

The abstract and conclusion have been revised to highlight the appropriate applications of thermodynamic modeling and the reasonable range of aerosol water pH inferred from such method. Revisions made in the manuscript:

> "***Abstract.*** *pH is an important property of aerosol particles but is difficult to measure directly. Several studies have estimated the pH values for fine particles in North China winter haze using thermodynamic models (i.e., E-AIM and ISORROPIA) and ambient measurements. The reported pH values differ widely, ranging from close to 0 (highly acidic) to as high as 7 (neutral). In order to understand the reason for this discrepancy, we calculated pH values using these models with different assumptions with regard to model inputs and particle phase states. We find that the large discrepancy is due primarily to differences in the model assumptions adopted in previous studies. Calculations using only aerosol phase composition as inputs (i.e., reverse mode) are sensitive to the measurement errors of ionic species and inferred pH values exhibit a bimodal distribution with peaks between −2 and 2 and between 7 and 10, depending on whether anions or cations are in excess.*

*Calculations using total (gas plus aerosol phase) measurements as inputs (i.e., forward mode) are affected much less by these measurement errors. In future studies, the reverse mode should be avoided whereas the forward mode should be used. Forward mode calculations in this and previous studies collectively indicate a moderately acidic condition (pH from about 4 to about 5) for fine particles in North China winter haze, indicating further that ammonia plays an important role in determining this property. The particle phase state assumed, either stable (solid plus liquid) or metastable (only liquid), does not significantly impact pH predictions. The unrealistic pH values of about 7 in a few previous studies (using the standard ISORROPIA model and stable state assumption) resulted from coding errors in the model, which have been identified and fixed in this study."*

*"Conclusions. This study suggests that the significant discrepancy of fine particle pH, ranging from about 0 (highly acidic) to about 7 (neutral), calculated in previous studies of North China winter haze is due primarily to differences in the ways in which the E-AIM and ISORROPIA thermodynamic equilibrium models have been applied. The reverse mode calculations (only using aerosol phase composition as inputs) lead to erroneous results of pH since they are strongly affected by ionic measurement errors (especially under ammonia-rich conditions), and therefore should be avoided in future winter haze studies. The forward mode calculations (using the total (gas plus aerosol phase) compositions as inputs) account for additional constraints imposed by the partitioning of semi-volatile species and are affected much less by the measurement errors, and therefore, should be used in future studies. The forward mode calculations in this and previous studies collectively indicate, during North China winter haze events, that aerosol particles are moderately acidic with pH values ranging from about 4 to about 5. The assumed particle phase state (stable or metastable) does not significantly affect the pH calculations of ISORROPIA after coding errors in its standard model being fixed. A few previous studies, in which the standard ISORROPIA model was used and the stable state was assumed, predicted unrealistic pH values of around 7, and should be re-evaluated. In agreement with previous studies, we confirm that ammonia plays an important role in determining particle pH under winter haze conditions in northern China."*

Specific comments:

1. Page 1, line 29: The authors indicate reverse mode calculations "exhibit a bimodal distribution with peaks between -2 and 2 and between 7 and 10." This reads as if these peaks are plausible values. Consider adding "depending on whether cations or anions were in excess" to highlight that the bimodal values are artifacts.

*Revised accordingly.*

2. Page 1, line 34-35 "The phase state assumed, which can be either stable (solid plus liquid) or metastable (only liquid), does not significantly impact pH predictions of ISORROPIA." Presumably this is true only at high RH? Figure 4a does not provide "stable" pH estimates below 60% RH and Figure 4b indicates the metastable and stable aerosol water differs (and is nonzero) between 40 and 70%.

*This statement is true for a large RH range. In Figure 4b, the predicted metastable and stable aerosol water contents are zero for 30% and 40% RH bins (the x and y axis is not intersected at zero). For the RH bin at 50% (i.e., 45%-55%), the aerosol water content is 2.8 µg m$^{-3}$ (a small value but nonzero). We did not show in Figure 4a the pH estimates for this RH range because there were many cases with no liquid phase in the stable state (the mutual deliquescence RH is around 50%). The average pH value for this RH range is 4.3 for the stable state (calculated using the available cases), which is ~0.2 unit smaller compared to that for the metastable state. Thus we think phase state does not significantly (compared to the reported 3 to 4 units' difference in some previous studies) impact pH predictions of ISORROPIA. We added a supplementary figure in the revised manuscript in order to clarify and emphasize this point and please see the details in the responses to Nenes et al.*

3. Page 3, line 27-30. The collection efficiency of the AMS is known to be a function of the ammonium to sulfate ratio (e.g. Middlebrook et al., 2012 https://www.tandfonline.com/doi/pdf/10.1080/02786826.2011.620041). Was this factored in?

*The several factors described in Middlebrook et al. (2012) which may affect the collection efficiency (CE) of the AMS have been considered when the CE of 0.5 was chosen in our AMS analysis. One sentence has been added here: "A*

*constant collection efficiency of 0.5 was chosen because (1) particles were dried before being analyzed, (2) the mass fraction of $NH_4NO_3$ was smaller than 0.4, and (3) the particle acidity was not high enough to affect CE substantially*".

4. Page 4, line 12: What effects of organic compounds does E-AIM consider? Dissociation of acids? Does it treat the effects of organics on inorganic activity coefficients?

E-AIM considers the dissociation of organic acids and treats the produced organic anions by the Pitzer, Simonson and Clegg (PSC) equations. E-AIM assumes that the inorganic ions and organic solutes do not influence each other. Their molality-based activity coefficients are thus equal to those calculated for the systems water + inorganic ions only, and water + organic solutes only (Clegg et al., 2001).

S. L. Clegg, J. H. Seinfeld, and P. Brimblecombe (2001) Thermodynamic modelling of aqueous aerosols containing electrolytes and dissolved organic compounds. J. Aerosol Sci. 32, 713-738.

The above information has been added in the revised manuscript.

5. Page 5, equations: Add "charge equivalent" before "measured ion concentrations" to indicate that sulfate, Ca, Mg have been multiplied by 2.

Revised accordingly.

6. Section 3.2.1 and Figure 2: Do E-AIM and ISORROPIA predict different H+ concentrations? To what degree? How much of the difference between ISORROPIA and E-AIM is due to including gamma_H+ different than 1 in reporting pH vs the activity coefficient of H+ actually modifying the thermodynamics? In other words, if you plotted E-AIM and ISORROPIA and set the activity coefficient to 1 in both for plotting purposes only, what would the difference be?

We have provided a detailed response to this question in the response to Nenes et al. Please refer to that.

7. Figure 3: Could ISORROPIA or E-AIM predictions be overlaid on the plot? What measurement technique is the measured $NH_4^+$ fraction from? Is it different than the AMS value?

In the revised manuscript, we have added the average pH values from different model calculations to this figure. It is now noted in the caption of this figure that "*the measured average aqueous fraction ... is calculated with the gas phase $NH_3$ and $PM_{2.5}$ $NH_4^+$ concentrations.*" The AMS pH value is similar, although it is not shown.

8. Page 10, before section 3.4: Emphasize and clearly state what your best estimate of aerosol pH is.

One more sentence has been added: "*The appropriate applications of thermodynamic modeling indicate a moderately acidic condition (pH from about 4 to about 5) for fine particles in North China winter haze.*"

9. Page 11, line 6-8: See above comment about a missing organic modification to pH.

See our responses to the general comment about the impacts of organics on pH above.

10. Page 11, line 17: What fraction of the total aerosol water is due to organic compounds?

The data has been added in the revised manuscript: "*we find that the aerosol water associated with these species is only about 14 ± 3% (median ± median absolute deviation) of that associated with inorganic salts*".

11. Page 12, near line 7. What is your hypothesis regarding liquid-liquid phase separation and the effect on pH? Isn't your default configuration essentially liquid-liquid phase separation into and organic-rich and inorganic-rich phase? This ties in with the fourth possible way organics affect pH (via activity coefficients if organic compounds coexist in the inorganic phase).

The aerosol solution under liquid-liquid phase separation can be separated into two phases: organic-rich and inorganic-rich, which may have different pH values (although such effect remains unclear). The study we cited here (Dallemagne et al., 2016) showed that the organic-rich phase had a pH value higher by 0.4 unit compared to the single phase situation. This effect is not the same as the fourth possible way, which is the influence of water-soluble organics on the activity coefficients of inorganic ions in the aqueous phase (inorganic-rich).

12. In the supporting information, can you provide the exact ISORROPIA file names and line numbers and what the content was modified?

The standard ISORROPIA source code is password protected at
*http://isorropia.eas.gatech.edu/index.php?title=Code_Repository*, but there is a version of ISORROPIA-II implemented by Pye et al. (2009) in the GEOS-Chem chemical transport model and fully publicly accessible at: *http://acmg.seas.harvard.edu/geos/doc/man/.* The ISORROPIA-II code is under the directory *ISOROPIA/.* Thus we have published bug fixes for ISORROPIA-II stable mode and the exact line numbers and contents can be found at: *http://wiki.seas.harvard.edu/geos-chem/index.php/ISORROPIA_II*

Pye, H. O. T., H. Liao, S. Wu, L. J. Mickley, D. J.Jacob, D. K. Henze, and J. H. Seinfeld, Effect of changes in climate and emissions on future sulfate-nitrate-ammonium aerosol levels in the United States, J. Geophys. Res., 114, D01205, 2009.

The above information has been added in the revised manuscript and the supplement.

**Response to A. Nenes, H. Guo, A. Russell and R. Weber**

Song et al.

*Comments are in black and responses are in blue.*

We would like to commend Song et al. for their extensive analysis that goes deep into the model code and data. The importance of understanding aerosol pH is key to understanding of aerosol growth and impacts, as has been demonstrated in a growing body of literature. This literature, however, also exposes knowledge gaps. Following are some comments and thoughts about the analysis that in our opinion require attention, especially on the impact and importance of the $H^+$ activity coefficient. Addressing these points, may require considerable rewriting and refocusing of the paper, but we feel it will eventually substantially enhance the contribution.

We thank Nenes et al. for their comments, which are very useful for improving the quality of this manuscript. The comments on the $H^+$ activity coefficients are especially helpful. Our responses to the specific points raised by Nenes et al. are given below.

*Algorithm changes to ISORROPIA-II routines.*

We would like to thank the authors for their very detailed explanation of the pH calculation issue, and the resolution provided. This clearly shows the value of having open source codes so that they are continuously used and tested by the community. The alternative approach in the standard code was used in the routines identified, because loss in precision in calculating the SQRT function (at low concentrations of aerosol precursors and when solid precipitates formed, e.g., $NH_4Cl$), made partitioning calculations at times inaccurate and noisy. Although the alternative approach captured partitioning, pH was clearly not, so adopting the standard calculation approach used in the subcases with higher RH values is appropriate; however, provision still needs to be shown to avoid loss of precision (e.g., Taylor expansion approximations or renormalization instead of SQRT). We will address this in the upcoming version of ISORROPIA-II.

We thank Nenes for sharing us the source codes of ISORROPIA, which allowed us to examine and identify the coding errors in the model.

*Application of thermodynamic models when interpreting data.*

We were very pleased to see that the analysis of Beijing data carried out here fully supports our prior work on how to use observational data to constrain pH, namely: *i*) avoiding usage of molar ratios and ion balances as pH proxies (Guo et al., 2015; Hennigan et al., 2015; Guo et al., 2016; Weber et al., 2016; Guo et al., 2017b), and, *ii*) the large pH errors that can result when aerosol-only concentrations from observations are used in open-system thermodynamic calculations (i.e., "reverse mode" calculations that are not subject to a global constraint of mass balance (Pilinis et al., 2000; Hennigan et al., 2015)).

We agree with Nenes et al. on this.

It should also be noted that the secondary effect of water-soluble organics on aerosol pH is also consistent with the recent work of Pye et al. (2017). Note that the reference to Pye et al. (2017) should refer to the following publication: Coupling of organic and inorganic aerosol systems and the effect on gas–particle partitioning in the southeastern US, Atmos. Chem. Phys., 18, 357-370, https://doi.org/10.5194/acp-18-357-2018, 2018.

The effect of water-soluble organic compounds is discussed in more detail in the revised manuscript. Please see the responses to anonymous reviewer #2.

One conclusion that the authors come to is that the pH calculations are not sensitive to the assumption of metastable and stable state. As presented, this can be misinterpreted by the reader that partitioning evaluations are not valuable for constraining aerosol pH. Partitioning calculations can sufficiently constrain pH, but only when predictions of aerosol water and semivolatile partitioning (of $NH_3$-$NH_4^+$, $HNO_3$-$NO_3^-$, and $HCl$-$Cl^-$ if possible) are reproduced by

observations (as shown in e.g., Guo et al. (2015) and other studies) – and a sufficient fraction of the partitioned aerosol species is associated with the aqueous phase. When aerosol water measurements are lacking or too uncertain, then showing that when **aqueous phase semivolatile partitioning by itself** (i.e., provided by the metastable solution) reproduces aerosol observations, aerosol pH is sufficiently constrained. **The pH values calculated for the metastable solution, for cases where partitioning is consistent with observations, provide the most plausible estimates of acidity.** pH values for the stable solution, especially when the liquid water content becomes very small (hence aqueous-phase partitioning a secondary contribution to the total partitioning), are subject to considerably more uncertainty – even if the pH corresponding to the metastable and stable solutions agree.

We partially disagree with Nenes et al. on this.

We think partitioning evaluations are important for evaluating the rationality of pH calculations in thermodynamic modeling. A good model–observation comparison of semivolatile species partitioning is a necessary condition for a good estimate of pH, but not a sufficient condition. When using the standard source codes, although model calculations can well reproduce the partitioning of semivolatile species for both particle phase states, the predicted pH values can be significantly different, for example, on average 4.6±0.4 (metastable) and 7.0±1.3 (stable) during 2012 winter in Xi'an, China (Wang et al., 2018). On the other hand, when using the revised source codes of ISORROPIA, model calculations under both states can well reproduce the partitioning of semivolatile species and also predict very similar pH values.

It is important to note that the predicted partitioning of semivolatile species is almost identical for both particle phase states. Guo et al. (2017c) mentioned that the partitioning of aerosol inorganic concentrations (e.g., $NH_4^+$, $NO_3^-$) using the metastable mode agreed better with the observations, when compared to those using the stable mode. We believe that the model–observation comparisons in Figure S1 of Guo et al. (2017c) were conducted inappropriately for the stable mode because only aqueous phase concentrations were used. However, the total particle phase (aqueous + solid) concentrations should be used in order to be consistent with ambient observations. The same results were also given in a recent study by Wang et al. (2018). In fact, since the forward thermodynamic calculations take the measured total (gas + particle) concentrations as model inputs, good model–observation comparisons for gas phase concentration (e.g., $NH_3$(g)) definitely mean that the model can well reproduce the observed particle phase concentrations (e.g., $NH_4^+$(p)).

**We think thermodynamic model calculations with either stable or metastable state assumption can provide reasonable estimates of aerosol water pH, and the predicted pH values for the stable solution are NOT subject to "considerably more uncertainty" when the aerosol water content is small, at least for the winter haze conditions considered in our study.** In order to better describe our point of view, here we conduct some more model calculations using ISORROPIA. The inputs are the average temperature and the average concentrations of total $H_2SO_4$, $HNO_3$, and $NH_3$ from our field measurements during haze episodes, and varied RH values from low to high. Figure R1 shows the comparisons of the predicted pH, AWC, ionic strength, and partitioning of $NH_3$ for both stable and metastable solutions.

As shown in Figure R1a–b, the predicted pH values for the stable and metastable solutions are exactly the same when the RH is larger than about 80% (when the RH is larger than the deliquescence RH for all the salts and both of the solution includes an aqueous phase only). When the RH is between about 60% and about 80% (when both aqueous and solid phases are present for the stable solution), the predicted pH values for the stable solution are on average 0.02 ± 0.00 greater than those for the metastable solution. This difference in pH is small relative to the uncertainty resulting from other factors (e.g., measurements of gas and aerosol species and meteorological parameters). Note that in Figure R1c for some cases the AWC in the stable solution is more than one order of magnitude lower than that in the metastable solution, and that for the same cases in the stable solution the "aqueous-phase partitioning a secondary contribution to the total partitioning", as can be seen in Figure R1e–f.

We also would like to emphasize that there has been no observational evidence so far to suggest whether the Beijing winter haze fine particles are in a metastable or stable state. It is also unlikely to figure out particle phase states from theoretical calculations because of the very large variability of ambient RH (see Section 2.2 of our paper) and the difficulty in estimating the efflorescence RH for multicomponent salts (Seinfeld and Pandis, 2016 Chapter 10). Therefore, at the current stage, a practical approach is to predict the aerosol water content and pH for both stable and

metastable states, which can provide a way to estimate the uncertainty of these variables due to the assumption of different phase states.

[Figure]

Figure R1. Comparisons of the ISORROPIA-predicted pH (a–b), AWC (c), ionic strength (d), and partitioning of NH$_3$ (e–f) under assumptions of the metastable and stable phase states. The model inputs include total H$_2$SO$_4$ = 30 µg m$^{-3}$, total HNO$_3$ = 51 µg m$^{-3}$, total NH$_3$ = 47 µg m$^{-3}$, temperature = 278 K, and varied RH. The inputs are calculated from our field measurements during haze episodes (RH > 60%) as the average temperature and the average concentrations of total H$_2$SO$_4$, HNO$_3$, and NH$_3$. Na$^+$ and K$^+$ are accounted for as equivalent NH$_4^+$, and Cl$^-$ as equivalent NO$_3^-$.

*Activity coefficient discussion*

The authors extensively comment on the usage of $\gamma_{H+}$ = 1 in some of the calculations behind ISORROPIA-II. In fact, the assumption that $\gamma_{H+}$ = 1 is thought to be a major source of pH discrepancy between ISORROPIA and E-AIM (it's even stated in the abstract). The data presented does not really support this for the following reasons:

1. $\gamma_{H+}$ varies by ±0.2 units over the ionic strengths considered (0-20), Figure 2b, while pH differences between the models are typically larger than 0.2 units.

2. The correlation of pH discrepancy between ISORROPIA-II (as calculated with the formula of Guo et al. (2015)) and E-AIM with $\gamma_{H+}$ *does not* indicate a causal relationship.

3. If $\gamma_{H+}$ = 1 was indeed the reason for the discrepancy, then at an ionic strength of ~20, when $\gamma_{H+}$ ~1, the pH discrepancy between ISORROPIA and E-AIM should be zero (Figure 2b). This is not the case at all.

Considering points 1, 2, 3 together, one can actually conclude that about 0.2 pH units discrepancy between ISORROPIA-II and E-AIM may arise from the assumption of $\gamma_{H+}$ = 1 for the RH (ionic strength) range considered, while the rest of the discrepancy may be related to the predicted concentration of H$^+$. This may even suggest that $\gamma_{H+}$ = 1 is not a leading cause of discrepancy. In support of this, we find it very interesting that when one compares the $\gamma_{H+}$ values from E-AIM (Figure 2c) and from AIOMFAC (Figure S6), log($\gamma_{H+}$) differs by about 0.6 units at an ionic strength of 20 M (E-AIM gives 0.1 and AIOMFAC gives -0.5; note the -0.6 difference in log($\gamma_{H+}$) means +0.6 pH compared to E-AIM), which seems to be consistent with the 0.6 higher pH comparing ISORROPIA to E-AIM at the same ionic strength. Could it just be then that the calculation of $\gamma_{H+}$ by E-AIM is more uncertain than implied? The Beijing haze polluted period has an ionic strength close to 40 M, which brings $\gamma_{H+}$ close to 1 according to Figure S6.

Assuming $\gamma_{H+} = 1$ to diagnose pH from ISORROPIA (single point) translates to an uncertainty of less than 0.5 pH units over a large range of RH or ionic strength (Figure S6).

**We agree with Nenes et al. that our data analysis does not support the statement in the original manuscript that the assumption of $\gamma_{H+} = 1$ is the major source of pH discrepancy between E-AIM and ISORROPIA. The Section 3.2.1 has been extensively revised and this statement has been removed from the abstract and conclusion.** In the revised manuscript, we conduct additional model calculations using E-AIM version II (for $H^+$ - $NH_4^+$ - $SO_4^{2-}$ - $NO_3^-$ - $H_2O$ aerosol), because this version can be used assuming a metastable state and thus predict pH at low RH values. It was not insightful to introduce a third model, AIOMFAC, to help explain the differences in predicted pH between ISORROPIA and E-AIM. Figure R2 below the difference in pH predicted by ISORROPIA and E-AIM as well as several other parameters. What we find from this additional data analysis is that the difference in predicted pH is: (1) systematic and related to RH, (2) related to both $H^+$ concentrations and activity coefficients, and (3) smaller than one unit for the cases tested. We note in the revised manuscript that "*The exact factors contributing to $\Delta pH$ remain unclear, since these two thermodynamic models differ in many ways (e.g., their methods in calculating the activity coefficients for ionic species other than $H^+$)*", and that "*the above analysis is based on the data sets collected in Beijing winter and may not apply to other conditions*".

[Figure]

Figure R2. Comparison of predicted pH values and several other parameters from ISORROPIA and E-AIM (version II) under typical Beijing winter haze conditions ($NH_x$-rich). The curve in each panel (c–f) shows the average value for each bin of RH. The model inputs include total $H_2SO_4 = 30$ µg m$^{-3}$, total $HNO_3 = 51$ µg m$^{-3}$, temperature = 278 K, and varied RH (from 30% to 90%) and total $NH_x$ concentrations (from 25 to 100 µg m$^{-3}$). The inputs are calculated from our field measurements during haze episodes (RH > 60%) as the average temperature and average concentrations of total $H_2SO_4$ and $HNO_3$. $Na^+$ and $K^+$ are accounted for as equivalent $NH_4^+$, and $Cl^-$ as equivalent $NO_3^-$.

There is a lack of discussion on the effects of the other ionic species as sources of discrepancy between E-AIM and ISORROPIA-II, which is surprising, given that E-AIM uses the single ion activity approach, while ISORROPIA uses mean activity coefficients of ion pairs. Predictions from the two types of activity coefficient models do show important differences (e.g., Kim et al., 1993).

We agree that the different methods in calculating activity coefficients for ionic species may be a source of difference in predicted pH between E-AIM and ISORROPIA (Kim et al., 1993; Zhang et al., 2000).

The configuration used in ISORROPIA-II (Kusik-Meisner binary activity coefficients with the Bromley mixing rule for multicomponent aerosol) has been shown to provide stable solutions for ionic strengths that far exceed 30, the limit where Pitzer coefficients have been shown to work well (Kim et al., 1993). The latter point of course is quite relevant for the discussion raised by the authors concerning the applicability of the activity coefficient models used by ISORROPIA-II and E-AIM when applied to the high ionic strengths corresponding to RH below 70%.

The Pitzer, Simonson, and Clegg (PSC) method, which is used in E-AIM, overcomes the limitation of molar-based ionic strength and is applicable the over entire concentration range (Zaveri et al., 2005).

Given the above, unless a thorough analysis of how all the activity coefficients water uptake and equilibrium constants contribute to the pH differences between ISORROPIA-II and E-AIM, one cannot really state how much uncertainty in pH arises from the assumption of $\gamma_{H+} = 1$, though it appears to be bounded and much less than the difference in the predicted pH's between the two models. Perhaps it would be better to just plot the predicted particle phase fractions ("$\varepsilon(NH_4^+)$ and "$\varepsilon(NO_3^-)$ as a function of pH) by each model and compare them against the data (following the approach of Guo et al., 2017a). Then one will have a better sense of the pH uncertainty (given by the range between models) for a given value of (observed) $\varepsilon$".

We agree that the pH differences between ISORROPIA and E-AIM cannot be completely explained unless a thorough comparative analysis is made. The main goal of this manuscript is to explain the large pH discrepancy (from about 0 to 7) reported in previous North China winter haze studies. The pH difference in the forward ISORROPIA and E-AIM model calculations is much smaller compared to the other factors (i.e., forward vs. reverse, metastable vs. stable of the standard ISORROPIA) and thus a thorough analysis on this relatively small pH difference between the forward ISORROPIA and E-AIM calculations goes beyond the focus of this manuscript. Both ISORROPIA and E-AIM show a moderately acidic condition for Beijing winter haze particles. It is noted in Figure 1 and Figure S5 that ISORROPIA and E-AIM predict nearly identical $\varepsilon(NH_4^+)$ and $\varepsilon(NO_3^-)$ and both can well capture the observed gas-particle partitioning. Therefore, we think that the difference between ISORROPIA and E-AIM may provide an estimate for the uncertainty of pH.

***Specific (but important) comments***

- The authors note early in the manuscript that the discrepancy in calculated pH when assuming $\gamma_{H+} = 1$ can be multiple units. This is not supported by the supplementary figure ($\gamma_{H+}$ from AIOMFAC), where for an ionic strength range of 0-100, log $\gamma_{H+}$ (hence the contribution of assuming $\gamma_{H+} = 1$ to pH discrepancy) varies within 1 unit. This has always been, by the way, our view – so it is nice to see this confirmed by the analysis presented! Also noted throughout is that pH is overestimated when assuming $\gamma_{H+} = 1$. This is not always true as well; as noted by the supplementary figure ($\gamma_{H+}$ from AIOMFAC), pH can be underestimated or overestimated by assuming $\gamma_{H+} = 1$, but not more than half a unit.

  Please see the responses to the general comments of activity coefficients above.

- In fact, the average pH estimated by ISORROPIA-II is *actually lower* than that reported for E-AIM (4.2 vs 5.4, for ISORROPIA-II vs E-AIM respectively) inconsistent with pH trends stated above.

  Wrong numbers of pH are quoted from the manuscript. As shown in Section 3.3, the mean pH values of 5.4 (E-AIM) and 4.2 (ISORROPIA) are for the *reverse mode* calculations. The average pH values are 4.0 (E-AIM) and 4.6 (ISORROPIA) for the *forward mode* calculations.

- In ISORROPIA-II, the non-ideal interactions of H+ with all the ions in solution (especially NO3, Cl, HSO4, SO4) is explicitly considered by the Kusik-Meisner and Bromely formulations. $\gamma_{H+} = 1$ is only invoked when the singe ion activity is required. This is not sufficiently noted in the text.

We noted in Section 2.2 that "With ISORROPIA, $\gamma_{H^+}$ and $\gamma_{OH^-}$ are assumed equal to unity, whereas the activity coefficients for the other ionic pairs (e.g., $H^+$–$Cl^-$) are calculated (Fountoukis and Nenes, 2007)."

- The authors understandably treat NVC (i.e. Ca, Mg, K, Na) as equivalent sodium, because E-AIM cannot explicitly treat Ca, Mg and K. The impact of this assumption can lead to important differences in the predicted thermodynamic state, owing to the strong nonideality of divalent ions and different water uptake characteristics of sodium salts vs. their other counterparts (e.g., Fountoukis et al., 2009).

  We noted in Figure 2 of the original manuscript that $K^+$ was accounted for as equivalent $Na^+$ in ISORROPIA, and therefore, for the comparison of pH for E-AIM and ISORROPIA, the model inputs are the same.

- What constitutes a "large/important" and "small/minor" different in pH depends on the context in which the pH is used. Constraining "absolute" pH for ambient aerosol to within less than 0.5 units may prove to be extremely challenging (e.g., the difference in log $\gamma_{H+}$ between E-AIM and AIOMFAC, the effects of organics on activity and water uptake and so on); so it may most likely be necessary to use a consistent pH calculation method and thermodynamic model when comparing aerosol acidities between models and/or observations.

  Such qualitative expressions have been avoided in the revised manuscript when comparing E-AIM and ISORROPIA.

- The authors caution about the predictions of ISORROPIA-II in metastable mode (for RH below the mutual deliquescence point) owing to the large ionic strengths of the solutions. Although we agree the ionic strengths are high, literature supports that the activity coefficient models used in ISORROPIA-II are stable for ionic strengths above 30, a situation that is also not the case for the Pitzer method (Kim et al., 1993).

  The Pitzer, Simonson, and Clegg (PSC) method, which is used in E-AIM, overcomes the limitation of molar-based ionic strength and is applicable over the entire concentration range (Zaveri et al., 2005).

In closing, we very much appreciate the analysis and it demonstrates an increasing sophistication of which the community is both understanding and discussing the thermodynamics of aerosols and the important topic of aerosol acidity. We also hope that the comments provided here add insight that will considerably strengthen the paper, and provide ideas for future work on the important topic of aerosol acidity.

The authors thank Nenes et al. again for the above comments, which were helpful in improving the quality of this manuscript.

The abstract and conclusion have been revised to highlight the appropriate applications of thermodynamic modeling and the reasonable range of aerosol water pH inferred from such method.

Revisions made in the manuscript:

> *"**Abstract.** pH is an important property of aerosol particles but is difficult to measure directly. Several studies have estimated the pH values for fine particles in North China winter haze using thermodynamic models (i.e., E-AIM and ISORROPIA) and ambient measurements. The reported pH values differ widely, ranging from close to 0 (highly acidic) to as high as 7 (neutral). In order to understand the reason for this discrepancy, we calculated pH values using these models with different assumptions with regard to model inputs and particle phase states. We find that the large discrepancy is due primarily to differences in the model assumptions adopted in previous studies. Calculations using only aerosol phase composition as inputs (i.e., reverse mode) are sensitive to the measurement errors of ionic species and inferred pH values exhibit a bimodal distribution with peaks between −2 and 2 and between 7 and 10, depending on whether anions or cations are in excess. Calculations using total (gas plus aerosol phase) measurements as inputs (i.e., forward mode) are affected much less by these measurement errors. In future studies, the reverse mode should be avoided whereas the forward mode should be used. Forward mode calculations in this and previous studies collectively indicate a moderately acidic condition (pH from about 4 to about 5) for fine particles in North China winter haze, indicating further that ammonia plays an important role in determining this property. The particle phase state assumed, either stable (solid plus liquid) or metastable (only liquid), does not significantly impact pH predictions. The unrealistic pH values of about 7 in a few previous studies (using the standard ISORROPIA model and stable state assumption) resulted from coding errors in the model, which have been identified and fixed in this study."*

> *"**Conclusions.** This study suggests that the significant discrepancy of fine particle pH, ranging from about 0 (highly acidic) to about 7 (neutral), calculated in previous studies of North China winter haze is due primarily to differences in the ways in which the E-AIM and ISORROPIA thermodynamic equilibrium models have been applied. The reverse mode calculations (only using aerosol phase composition as inputs) lead to erroneous results of pH since they are strongly affected by ionic measurement errors (especially under ammonia-rich conditions), and therefore should be avoided in future winter haze studies. The forward mode calculations (using the total (gas plus aerosol phase) compositions as inputs) account for additional constraints imposed by the partitioning of semi-volatile species and are affected much less by the measurement errors, and therefore, should be used in future studies. The forward mode calculations in this and previous studies collectively indicate, during North China winter haze events, that aerosol particles are moderately acidic with pH values ranging from about 4 to about 5. The assumed particle phase state (stable or metastable) does not significantly affect the pH calculations of ISORROPIA after coding errors in its standard model being fixed. A few previous studies, in which the standard ISORROPIA model was used and the stable state was assumed, predicted unrealistic pH values of around 7, and should be re-evaluated. In agreement with previous studies, we confirm that ammonia plays an important role in determining particle pH under winter haze conditions in northern China."*

(2) The "forward stable" module was modified by the authors, although I did not read it in detail, the results seem more reasonable than previous runs. If possible, please contact GIT group to confirm it.

Our modification of the ISORROPIA source code for the "forward stable" mode has been confirmed by its developers.

(3) Some recent studies declared the aerosol pH could be close to 7 due to the high ammonia level, please use the sensitivity test to show if it is possible. The implications of aerosol pH should be very important for atmospheric reactions.

The sensitivity of pH to ammonia concentration levels has been examined in a few previous winter haze studies (e.g., Liu et al., 2017 and Guo et al., 2017). Thus, in this study we cited their results in Sect. 3.2.3 in the revised manuscript and also provided a similar sensitivity test. Our conclusions are the same as those from these previous studies.

Revisions made in the manuscript:

> *"By analyzing the sensitivity of pH to ammonia concentrations, recent studies have emphasized the important role of ammonia in determining winter haze particle pH (Guo et al., 2017b; Liu et al., 2017a). It was suggested, under ammonia-rich conditions, that a 10-fold increase in gas phase $NH_3$ concentrations roughly corresponds to one unit increase in pH (i.e., a 10-fold decrease in $H^+$ activity) (Guo et al., 2017b). This is obvious, since the equilibrium of dissolution and dissociation of ammonia in water can be expressed as: $NH_{3(g)} + H^+_{(aq)} \leftrightarrow NH^+_{4(aq)}$. These sensitivity tests have also indicated that atmospheric relevant ammonia concentrations are not high enough to achieve a fully neutralized condition (pH of around 7) for aerosol particles (Guo et al., 2017b; Liu et al., 2017a). The sensitivity tests conducted in this study are consistent with these previous studies (Fig. S9 in the Supplement)."*

Liu, M., Song, Y., Zhou, T., Xu, Z., Yan, C., Zheng, M., Wu, Z., Hu, M., Wu, Y., and Zhu, T.: Fine particle pH during severe haze episodes in northern China, Geophys. Res. Lett., 44, 5213-5221, doi:10.1002/2017GL073210, 2017a.

Guo, H., Weber, R. J., and Nenes, A.: High levels of ammonia do not raise fine particle pH sufficiently to yield nitrogen oxide-dominated sulfate production, Sci. Rep., 7, 12109, doi:10.1038/s41598-017-11704-0, 2017b.

(4) Please have some discussion on the aerosol water content effects on pH values especially when RH was high.

In the revised manuscript, we have added more discussion in Sect. 3.2.3 on the effects of ammonia concentrations and RH on the pH values predicted by ISORROPIA and E-AIM. We think it does not make much sense to directly compare aerosol water content and pH values, because the aerosol water content predicted by the thermodynamic equilibrium models depends on the amount of chemical species and RH values in the input. If the input RH and excess ammonia remain constant, increasing the amount of chemical species would increase aerosol water content but would not change predicted pH values significantly. On the other hand, if the excess ammonia and the amount of chemical species in the input remain constant, increasing RH values would affect predicted pH values, but this effect is different for E-AIM and ISORROPIA. Thus we choose to evaluate the effect of changing RH on the predicted pH values.

Below is a new figure in the supplement of the revised manuscript. It shows the sensitivity of pH to $NH_x$ concentrations and RH. The model inputs include total $H_2SO_4$ = 30 μg m$^{-3}$, total $HNO_3$ = 51 μg m$^{-3}$, temperature = 278 K, and varying RH (from 30% to 90%) and total $NH_x$ concentrations (from 25 to 100 μg m$^{-3}$). The inputs are calculated from our field measurements during haze episodes (RH > 60%) as the average temperature and average concentrations of total $H_2SO_4$, and $HNO_3$. $Na^+$ and $K^+$ are accounted for as equivalent $NH_4^+$, and $Cl^-$ as equivalent $NO_3^-$. The required $NH_x$ concentrations calculated for the input total $H_2SO_4$ and $HNO_3$ concentrations are about 24 μg m$^{-3}$. The curves in panels (c and d) show the average pH in each bin for $NH_x$ concentrations or RH.

Panel (d) of this figure shows the relationship of RH and pH. We find that the pH from ISORROPIA is insensitive to RH and the variability of pH is less than 0.3 unit. On the other hand, pH from E-AIM increases by 0.8 unit with increasing RH from 30% to 90%. The reason for this difference in pH between ISORROPIA and E-AIM remains unclear, and is discussed briefly in Sect. 3.2.1.

[Figure]

**Response to Anonymous Referee #2**

Song et al.

*Comments are in black and responses are in* *blue*.

This paper provides insight into the acidity of aerosols in Beijing. Table 1 provides a nice summary of previously published values which range from very acidic (-1) to basic (7.6). The paper uses ISORROPIA and E-AIM to estimate pH and provides discussion on how organic compounds may modify pH. The paper is well written, fairly thorough, and detailed. The Monte Carlo approach provides additional confidence in the results. A number of (mostly clarifying) comments are listed below.

Thanks for these positive comments for our manuscript. Our responses to specific comments are provided below.

One role for organic compounds in modifying pH was missing from the discussion. Specifically, on page 11, the authors list three ways in which organics can modify pH: (1) adding aerosol water, (2) participating in charge balance (e.g. dissociation of organic acids), and (3) by changing the aerosol phase state. The third area could use clarification (see detailed comments), but a fourth way (that seemed to be missing) is through modification of the chemical environment and therefore by modifying the activity coefficients of the inorganic species. This could be scoped out using the AIOMFAC model.

It is a very good comment. E-AIM assumes that the inorganic ions and organic solutes do not influence each other. Their molality-based activity coefficients are thus equal to those calculated for the systems water + inorganic ions only, and water + organic solutes only (Clegg et al., 2001). In the revised manuscript, we add the fourth way in Sect. 3.4 when discussing the effect of organics on particle pH calculation. Specifically, we modified this sentence to "*Organics may affect particle pH in several ways: (1) by increasing the absorption of aerosol water; (2) by participating in the charge balance and modifying the activity coefficients of inorganic ions in the aqueous phase; and (3) by changing the aerosol phase state (liquid-liquid phase separation)*", and added "*Note that E-AIM assumes that the organics in the aqueous solution do not affect the activity coefficients of inorganic ions (Clegg et al., 2001). Using the AIOMFAC model (web.meteo.mcgill.ca/aiomfac), Pye et al. (2018) recently showed that the interaction of inorganic ions with water-soluble organic compounds resulted in a 0.1 unit increase in pH for aerosols in the southeast United States*".

S. L. Clegg, J. H. Seinfeld, and P. Brimblecombe (2001) Thermodynamic modelling of aqueous aerosols containing electrolytes and dissolved organic compounds. J. Aerosol Sci. 32, 713-738.

Pye, H. O. T., Zuend, A., Fry, J. L., Isaacman-VanWertz, G., Capps, S. L., Appel, K. W., Foroutan, H., Xu, L., Ng, N. L., and Goldstein, A. H.: Coupling of organic and inorganic aerosol systems and the effect on gas–particle partitioning in the southeastern US, Atmos. Chem. Phys., 18, 357-370, doi:10.5194/acp-18-357-2018, 2018.

The authors should also be more forceful in their statements regarding what is a reasonable pH calculation and what is likely erroneous (see specific comment 1).

The abstract and conclusion have been revised to highlight the appropriate applications of thermodynamic modeling and the reasonable range of aerosol water pH inferred from such method. Revisions made in the manuscript:

> *"**Abstract.** pH is an important property of aerosol particles but is difficult to measure directly. Several studies have estimated the pH values for fine particles in North China winter haze using thermodynamic models (i.e., E-AIM and ISORROPIA) and ambient measurements. The reported pH values differ widely, ranging from close to 0 (highly acidic) to as high as 7 (neutral). In order to understand the reason for this discrepancy, we calculated pH values using these models with different assumptions with regard to model inputs and particle phase states. We find that the large discrepancy is due primarily to differences in the model assumptions adopted in previous studies. Calculations using only aerosol phase composition as inputs (i.e., reverse mode) are sensitive to the measurement errors of ionic species and inferred pH values exhibit a bimodal distribution with peaks between −2 and 2 and between 7 and 10, depending on whether anions or cations are in excess.*

*Calculations using total (gas plus aerosol phase) measurements as inputs (i.e., forward mode) are affected much less by these measurement errors. In future studies, the reverse mode should be avoided whereas the forward mode should be used. Forward mode calculations in this and previous studies collectively indicate a moderately acidic condition (pH from about 4 to about 5) for fine particles in North China winter haze, indicating further that ammonia plays an important role in determining this property. The particle phase state assumed, either stable (solid plus liquid) or metastable (only liquid), does not significantly impact pH predictions. The unrealistic pH values of about 7 in a few previous studies (using the standard ISORROPIA model and stable state assumption) resulted from coding errors in the model, which have been identified and fixed in this study."*

*"**Conclusions.** This study suggests that the significant discrepancy of fine particle pH, ranging from about 0 (highly acidic) to about 7 (neutral), calculated in previous studies of North China winter haze is due primarily to differences in the ways in which the E-AIM and ISORROPIA thermodynamic equilibrium models have been applied. The reverse mode calculations (only using aerosol phase composition as inputs) lead to erroneous results of pH since they are strongly affected by ionic measurement errors (especially under ammonia-rich conditions), and therefore should be avoided in future winter haze studies. The forward mode calculations (using the total (gas plus aerosol phase) compositions as inputs) account for additional constraints imposed by the partitioning of semi-volatile species and are affected much less by the measurement errors, and therefore, should be used in future studies. The forward mode calculations in this and previous studies collectively indicate, during North China winter haze events, that aerosol particles are moderately acidic with pH values ranging from about 4 to about 5. The assumed particle phase state (stable or metastable) does not significantly affect the pH calculations of ISORROPIA after coding errors in its standard model being fixed. A few previous studies, in which the standard ISORROPIA model was used and the stable state was assumed, predicted unrealistic pH values of around 7, and should be re-evaluated. In agreement with previous studies, we confirm that ammonia plays an important role in determining particle pH under winter haze conditions in northern China."*

Specific comments:

1. Page 1, line 29: The authors indicate reverse mode calculations "exhibit a bimodal distribution with peaks between -2 and 2 and between 7 and 10." This reads as if these peaks are plausible values. Consider adding "depending on whether cations or anions were in excess" to highlight that the bimodal values are artifacts.

Revised accordingly.

2. Page 1, line 34-35 "The phase state assumed, which can be either stable (solid plus liquid) or metastable (only liquid), does not significantly impact pH predictions of ISORROPIA." Presumably this is true only at high RH? Figure 4a does not provide "stable" pH estimates below 60% RH and Figure 4b indicates the metastable and stable aerosol water differs (and is nonzero) between 40 and 70%.

This statement is true for a large RH range. In Figure 4b, the predicted metastable and stable aerosol water contents are zero for 30% and 40% RH bins (the *x* and *y* axis is not intersected at zero). For the RH bin at 50% (i.e., 45%-55%), the aerosol water content is 2.8 µg m$^{-3}$ (a small value but nonzero). We did not show in Figure 4a the pH estimates for this RH range because there were many cases with no liquid phase in the stable state (the mutual deliquescence RH is around 50%). The average pH value for this RH range is 4.3 for the stable state (calculated using the available cases), which is ~0.2 unit smaller compared to that for the metastable state. Thus we think phase state does not *significantly* (compared to the reported 3 to 4 units' difference in some previous studies) impact pH predictions of ISORROPIA. We added a supplementary figure in the revised manuscript in order to clarify and emphasize this point and please see the details in the responses to Nenes et al.

3. Page 3, line 27-30. The collection efficiency of the AMS is known to be a function of the ammonium to sulfate ratio (e.g. Middlebrook et al., 2012 https://www.tandfonline.com/doi/pdf/10.1080/02786826.2011.620041). Was this factored in?

The several factors described in Middlebrook et al. (2012) which may affect the collection efficiency (CE) of the AMS have been considered when the CE of 0.5 was chosen in our AMS analysis. One sentence has been added here: "*A*

*constant collection efficiency of 0.5 was chosen because (1) particles were dried before being analyzed, (2) the mass fraction of NH₄NO₃ was smaller than 0.4, and (3) the particle acidity was not high enough to affect CE substantially*".

4. Page 4, line 12: What effects of organic compounds does E-AIM consider? Dissociation of acids? Does it treat the effects of organics on inorganic activity coefficients?

E-AIM considers the dissociation of organic acids and treats the produced organic anions by the Pitzer, Simonson and Clegg (PSC) equations. E-AIM assumes that the inorganic ions and organic solutes do not influence each other. Their molality-based activity coefficients are thus equal to those calculated for the systems water + inorganic ions only, and water + organic solutes only (Clegg et al., 2001).

S. L. Clegg, J. H. Seinfeld, and P. Brimblecombe (2001) Thermodynamic modelling of aqueous aerosols containing electrolytes and dissolved organic compounds. J. Aerosol Sci. 32, 713-738.

The above information has been added in the revised manuscript.

5. Page 5, equations: Add "charge equivalent" before "measured ion concentrations" to indicate that sulfate, Ca, Mg have been multiplied by 2.

Revised accordingly.

6. Section 3.2.1 and Figure 2: Do E-AIM and ISORROPIA predict different H+ concentrations? To what degree? How much of the difference between ISORROPIA and E-AIM is due to including gamma_H+ different than 1 in reporting pH vs the activity coefficient of H+ actually modifying the thermodynamics? In other words, if you plotted E-AIM and ISORROPIA and set the activity coefficient to 1 in both for plotting purposes only, what would the difference be?

We have provided a detailed response to this question in the response to Nenes et al. Please refer to that.

7. Figure 3: Could ISORROPIA or E-AIM predictions be overlaid on the plot? What measurement technique is the measured NH₄⁺ fraction from? Is it different than the AMS value?

In the revised manuscript, we have added the average pH values from different model calculations to this figure. It is now noted in the caption of this figure that "*the measured average aqueous fraction … is calculated with the gas phase NH₃ and PM₂.₅ NH₄⁺ concentrations.*" The AMS pH value is similar, although it is not shown.

8. Page 10, before section 3.4: Emphasize and clearly state what your best estimate of aerosol pH is.

One more sentence has been added: "*The appropriate applications of thermodynamic modeling indicate a moderately acidic condition (pH from about 4 to about 5) for fine particles in North China winter haze.*"

9. Page 11, line 6-8: See above comment about a missing organic modification to pH.

See our responses to the general comment about the impacts of organics on pH above.

10. Page 11, line 17: What fraction of the total aerosol water is due to organic compounds?

The data has been added in the revised manuscript: "*we find that the aerosol water associated with these species is only about 14 ± 3% (median ± median absolute deviation) of that associated with inorganic salts*".

11. Page 12, near line 7. What is your hypothesis regarding liquid-liquid phase separation and the effect on pH? Isn't your default configuration essentially liquid-liquid phase separation into and organic-rich and inorganic-rich phase? This ties in with the fourth possible way organics affect pH (via activity coefficients if organic compounds coexist in the inorganic phase).

The aerosol solution under liquid-liquid phase separation can be separated into two phases: organic-rich and inorganic-rich, which may have different pH values (although such effect remains unclear). The study we cited here (Dallemagne et al., 2016) showed that the organic-rich phase had a pH value higher by 0.4 unit compared to the single phase situation. This effect is not the same as the fourth possible way, which is the influence of water-soluble organics on the activity coefficients of inorganic ions in the aqueous phase (inorganic-rich).

12. In the supporting information, can you provide the exact ISORROPIA file names and line numbers and what the content was modified?

The standard ISORROPIA source code is password protected at
*http://isorropia.eas.gatech.edu/index.php?title=Code_Repository*, but there is a version of ISORROPIA-II implemented by Pye et al. (2009) in the GEOS-Chem chemical transport model and fully publicly accessible at: *http://acmg.seas.harvard.edu/geos/doc/man/.* The ISORROPIA-II code is under the directory *ISOROPIA/.* Thus we have published bug fixes for ISORROPIA-II stable mode and the exact line numbers and contents can be found at: *http://wiki.seas.harvard.edu/geos-chem/index.php/ISORROPIA_II*

Pye, H. O. T., H. Liao, S. Wu, L. J. Mickley, D. J.Jacob, D. K. Henze, and J. H. Seinfeld, Effect of changes in climate and emissions on future sulfate-nitrate-ammonium aerosol levels in the United States, J. Geophys. Res., 114, D01205, 2009.

The above information has been added in the revised manuscript and the supplement.

**Response to A. Nenes, H. Guo, A. Russell and R. Weber**

Song et al.

*Comments are in black and responses are in blue.*

We would like to commend Song et al. for their extensive analysis that goes deep into the model code and data. The importance of understanding aerosol pH is key to understanding of aerosol growth and impacts, as has been demonstrated in a growing body of literature. This literature, however, also exposes knowledge gaps. Following are some comments and thoughts about the analysis that in our opinion require attention, especially on the impact and importance of the H$^+$ activity coefficient. Addressing these points, may require considerable rewriting and refocusing of the paper, but we feel it will eventually substantially enhance the contribution.

We thank Nenes et al. for their comments, which are very useful for improving the quality of this manuscript. The comments on the H$^+$ activity coefficients are especially helpful. Our responses to the specific points raised by Nenes et al. are given below.

*Algorithm changes to ISORROPIA-II routines.*

We would like to thank the authors for their very detailed explanation of the pH calculation issue, and the resolution provided. This clearly shows the value of having open source codes so that they are continuously used and tested by the community. The alternative approach in the standard code was used in the routines identified, because loss in precision in calculating the SQRT function (at low concentrations of aerosol precursors and when solid precipitates formed, e.g., NH$_4$Cl), made partitioning calculations at times inaccurate and noisy. Although the alternative approach captured partitioning, pH was clearly not, so adopting the standard calculation approach used in the subcases with higher RH values is appropriate; however, provision still needs to be shown to avoid loss of precision (e.g., Taylor expansion approximations or renormalization instead of SQRT). We will address this in the upcoming version of ISORROPIA-II.

We thank Nenes for sharing us the source codes of ISORROPIA, which allowed us to examine and identify the coding errors in the model.

*Application of thermodynamic models when interpreting data.*

We were very pleased to see that the analysis of Beijing data carried out here fully supports our prior work on how to use observational data to constrain pH, namely: *i*) avoiding usage of molar ratios and ion balances as pH proxies (Guo et al., 2015; Hennigan et al., 2015; Guo et al., 2016; Weber et al., 2016; Guo et al., 2017b), and, *ii*) the large pH errors that can result when aerosol-only concentrations from observations are used in open-system thermodynamic calculations (i.e., "reverse mode" calculations that are not subject to a global constraint of mass balance (Pilinis et al., 2000; Hennigan et al., 2015)).

We agree with Nenes et al. on this.

It should also be noted that the secondary effect of water-soluble organics on aerosol pH is also consistent with the recent work of Pye et al. (2017). Note that the reference to Pye et al. (2017) should refer to the following publication: Coupling of organic and inorganic aerosol systems and the effect on gas–particle partitioning in the southeastern US, Atmos. Chem. Phys., 18, 357-370, https://doi.org/10.5194/acp-18-357-2018, 2018.

The effect of water-soluble organic compounds is discussed in more detail in the revised manuscript. Please see the responses to anonymous reviewer #2.

One conclusion that the authors come to is that the pH calculations are not sensitive to the assumption of metastable and stable state. As presented, this can be misinterpreted by the reader that partitioning evaluations are not valuable for constraining aerosol pH. Partitioning calculations can sufficiently constrain pH, but only when predictions of aerosol water and semivolatile partitioning (of NH$_3$-NH$_4^+$, HNO$_3$-NO$_3^-$, and HCl-Cl$^-$ if possible) are reproduced by

observations (as shown in e.g., Guo et al. (2015) and other studies) – and a sufficient fraction of the partitioned aerosol species is associated with the aqueous phase. When aerosol water measurements are lacking or too uncertain, then showing that when **aqueous phase semivolatile partitioning by itself** (i.e., provided by the metastable solution) reproduces aerosol observations, aerosol pH is sufficiently constrained. **The pH values calculated for the metastable solution, for cases where partitioning is consistent with observations, provide the most plausible estimates of acidity.** pH values for the stable solution, especially when the liquid water content becomes very small (hence aqueous-phase partitioning a secondary contribution to the total partitioning), are subject to considerably more uncertainty – even if the pH corresponding to the metastable and stable solutions agree.

We partially disagree with Nenes et al. on this.

We think partitioning evaluations are important for evaluating the rationality of pH calculations in thermodynamic modeling. A good model–observation comparison of semivolatile species partitioning is a necessary condition for a good estimate of pH, but not a sufficient condition. When using the standard source codes, although model calculations can well reproduce the partitioning of semivolatile species for both particle phase states, the predicted pH values can be significantly different, for example, on average 4.6±0.4 (metastable) and 7.0±1.3 (stable) during 2012 winter in Xi'an, China (Wang et al., 2018). On the other hand, when using the revised source codes of ISORROPIA, model calculations under both states can well reproduce the partitioning of semivolatile species and also predict very similar pH values.

It is important to note that the predicted partitioning of semivolatile species is almost identical for both particle phase states. Guo et al. (2017c) mentioned that the partitioning of aerosol inorganic concentrations (e.g., $NH_4^+$, $NO_3^-$) using the metastable mode agreed better with the observations, when compared to those using the stable mode. We believe that the model–observation comparisons in Figure S1 of Guo et al. (2017c) were conducted inappropriately for the stable mode because only aqueous phase concentrations were used. However, the total particle phase (aqueous + solid) concentrations should be used in order to be consistent with ambient observations. The same results were also given in a recent study by Wang et al. (2018). In fact, since the forward thermodynamic calculations take the measured total (gas + particle) concentrations as model inputs, good model–observation comparisons for gas phase concentration (e.g., $NH_3(g)$) definitely mean that the model can well reproduce the observed particle phase concentrations (e.g., $NH_4^+(p)$).

**We think thermodynamic model calculations with either stable or metastable state assumption can provide reasonable estimates of aerosol water pH, and the predicted pH values for the stable solution are NOT subject to "considerably more uncertainty" when the aerosol water content is small, at least for the winter haze conditions considered in our study.** In order to better describe our point of view, here we conduct some more model calculations using ISORROPIA. The inputs are the average temperature and the average concentrations of total $H_2SO_4$, $HNO_3$, and $NH_3$ from our field measurements during haze episodes, and varied RH values from low to high. Figure R1 shows the comparisons of the predicted pH, AWC, ionic strength, and partitioning of $NH_3$ for both stable and metastable solutions.

As shown in Figure R1a–b, the predicted pH values for the stable and metastable solutions are exactly the same when the RH is larger than about 80% (when the RH is larger than the deliquescence RH for all the salts and both of the solution includes an aqueous phase only). When the RH is between about 60% and about 80% (when both aqueous and solid phases are present for the stable solution), the predicted pH values for the stable solution are on average 0.02 ± 0.00 greater than those for the metastable solution. This difference in pH is small relative to the uncertainty resulting from other factors (e.g., measurements of gas and aerosol species and meteorological parameters). Note that in Figure R1c for some cases the AWC in the stable solution is more than one order of magnitude lower than that in the metastable solution, and that for the same cases in the stable solution the "aqueous-phase partitioning a secondary contribution to the total partitioning", as can be seen in Figure R1e–f.

We also would like to emphasize that there has been no observational evidence so far to suggest whether the Beijing winter haze fine particles are in a metastable or stable state. It is also unlikely to figure out particle phase states from theoretical calculations because of the very large variability of ambient RH (see Section 2.2 of our paper) and the difficulty in estimating the efflorescence RH for multicomponent salts (Seinfeld and Pandis, 2016 Chapter 10). Therefore, at the current stage, a practical approach is to predict the aerosol water content and pH for both stable and

metastable states, which can provide a way to estimate the uncertainty of these variables due to the assumption of different phase states.

[Figure]

Figure R1. Comparisons of the ISORROPIA-predicted pH (a–b), AWC (c), ionic strength (d), and partitioning of $NH_3$ (e–f) under assumptions of the metastable and stable phase states. The model inputs include total $H_2SO_4 = 30\ \mu g\ m^{-3}$, total $HNO_3 = 51\ \mu g\ m^{-3}$, total $NH_3 = 47\ \mu g\ m^{-3}$, temperature = 278 K, and varied RH. The inputs are calculated from our field measurements during haze episodes (RH > 60%) as the average temperature and the average concentrations of total $H_2SO_4$, $HNO_3$, and $NH_3$. $Na^+$ and $K^+$ are accounted for as equivalent $NH_4^+$, and $Cl^-$ as equivalent $NO_3^-$.

*Activity coefficient discussion*

The authors extensively comment on the usage of $\gamma_{H+} = 1$ in some of the calculations behind ISORROPIA-II. In fact, the assumption that $\gamma_{H+} = 1$ is thought to be a major source of pH discrepancy between ISORROPIA and E-AIM (it's even stated in the abstract). The data presented does not really support this for the following reasons:

1. $\gamma_{H+}$ varies by ±0.2 units over the ionic strengths considered (0-20), Figure 2b, while pH differences between the models are typically larger than 0.2 units.

2. The correlation of pH discrepancy between ISORROPIA-II (as calculated with the formula of Guo et al. (2015)) and E-AIM with $\gamma_{H+}$ *does not* indicate a causal relationship.

3. If $\gamma_{H+} = 1$ was indeed the reason for the discrepancy, then at an ionic strength of ~20, when $\gamma_{H+}$ ~1, the pH discrepancy between ISORROPIA and E-AIM should be zero (Figure 2b). This is not the case at all.

Considering points 1, 2, 3 together, one can actually conclude that about 0.2 pH units discrepancy between ISORROPIA-II and E-AIM may arise from the assumption of $\gamma_{H+} = 1$ for the RH (ionic strength) range considered, while the rest of the discrepancy may be related to the predicted concentration of $H^+$. This may even suggest that $\gamma_{H+} = 1$ is not a leading cause of discrepancy. In support of this, we find it very interesting that when one compares the $\gamma_{H+}$ values from E-AIM (Figure 2c) and from AIOMFAC (Figure S6), $\log(\gamma_{H+})$ differs by about 0.6 units at an ionic strength of 20 M (E-AIM gives 0.1 and AIOMFAC gives -0.5; note the -0.6 difference in $\log(\gamma_{H+})$ means +0.6 pH compared to E-AIM), which seems to be consistent with the 0.6 higher pH comparing ISORROPIA to E-AIM at the same ionic strength. Could it just be then that the calculation of $\gamma_{H+}$ by E-AIM is more uncertain than implied? The Beijing haze polluted period has an ionic strength close to 40 M, which brings $\gamma_{H+}$ close to 1 according to Figure S6.

Assuming $\gamma_{H+} = 1$ to diagnose pH from ISORROPIA (single point) translates to an uncertainty of less than 0.5 pH units over a large range of RH or ionic strength (Figure S6).

**We agree with Nenes et al. that our data analysis does not support the statement in the original manuscript that the assumption of $\gamma_{H+} = 1$ is the major source of pH discrepancy between E-AIM and ISORROPIA. The Section 3.2.1 has been extensively revised and this statement has been removed from the abstract and conclusion.** In the revised manuscript, we conduct additional model calculations using E-AIM version II (for $H^+$ - $NH_4^+$ - $SO_4^{2-}$ - $NO_3^-$ - $H_2O$ aerosol), because this version can be used assuming a metastable state and thus predict pH at low RH values. It was not insightful to introduce a third model, AIOMFAC, to help explain the differences in predicted pH between ISORROPIA and E-AIM. Figure R2 below the difference in pH predicted by ISORROPIA and E-AIM as well as several other parameters. What we find from this additional data analysis is that the difference in predicted pH is: (1) systematic and related to RH, (2) related to both $H^+$ concentrations and activity coefficients, and (3) smaller than one unit for the cases tested. We note in the revised manuscript that "*The exact factors contributing to $\Delta pH$ remain unclear, since these two thermodynamic models differ in many ways (e.g., their methods in calculating the activity coefficients for ionic species other than $H^+$)*", and that "*the above analysis is based on the data sets collected in Beijing winter and may not apply to other conditions*".

[Figure]

Figure R2. Comparison of predicted pH values and several other parameters from ISORROPIA and E-AIM (version II) under typical Beijing winter haze conditions ($NH_x$-rich). The curve in each panel (c–f) shows the average value for each bin of RH. The model inputs include total $H_2SO_4 = 30$ µg m$^{-3}$, total $HNO_3 = 51$ µg m$^{-3}$, temperature = 278 K, and varied RH (from 30% to 90%) and total $NH_x$ concentrations (from 25 to 100 µg m$^{-3}$). The inputs are calculated from our field measurements during haze episodes (RH > 60%) as the average temperature and average concentrations of total $H_2SO_4$ and $HNO_3$. $Na^+$ and $K^+$ are accounted for as equivalent $NH_4^+$, and $Cl^-$ as equivalent $NO_3^-$.

There is a lack of discussion on the effects of the other ionic species as sources of discrepancy between E-AIM and ISORROPIA-II, which is surprising, given that E-AIM uses the single ion activity approach, while ISORROPIA uses mean activity coefficients of ion pairs. Predictions from the two types of activity coefficient models do show important differences (e.g., Kim et al., 1993).

We agree that the different methods in calculating activity coefficients for ionic species may be a source of difference in predicted pH between E-AIM and ISORROPIA (Kim et al., 1993; Zhang et al., 2000).

The configuration used in ISORROPIA-II (Kusik-Meisner binary activity coefficients with the Bromley mixing rule for multicomponent aerosol) has been shown to provide stable solutions for ionic strengths that far exceed 30, the limit where Pitzer coefficients have been shown to work well (Kim et al., 1993). The latter point of course is quite relevant for the discussion raised by the authors concerning the applicability of the activity coefficient models used by ISORROPIA-II and E-AIM when applied to the high ionic strengths corresponding to RH below 70%.

The Pitzer, Simonson, and Clegg (PSC) method, which is used in E-AIM, overcomes the limitation of molar-based ionic strength and is applicable the over entire concentration range (Zaveri et al., 2005).

Given the above, unless a thorough analysis of how all the activity coefficients water uptake and equilibrium constants contribute to the pH differences between ISORROPIA-II and E-AIM, one cannot really state how much uncertainty in pH arises from the assumption of $\gamma_{H+} = 1$, though it appears to be bounded and much less than the difference in the predicted pH's between the two models. Perhaps it would be better to just plot the predicted particle phase fractions ("$\varepsilon(NH_4^+)$ and "$\varepsilon(NO_3^-)$ as a function of pH) by each model and compare them against the data (following the approach of Guo et al., 2017a). Then one will have a better sense of the pH uncertainty (given by the range between models) for a given value of (observed) $\varepsilon$".

We agree that the pH differences between ISORROPIA and E-AIM cannot be completely explained unless a thorough comparative analysis is made. The main goal of this manuscript is to explain the large pH discrepancy (from about 0 to 7) reported in previous North China winter haze studies. The pH difference in the forward ISORROPIA and E-AIM model calculations is much smaller compared to the other factors (i.e., forward vs. reverse, metastable vs. stable of the standard ISORROPIA) and thus a thorough analysis on this relatively small pH difference between the forward ISORROPIA and E-AIM calculations goes beyond the focus of this manuscript. Both ISORROPIA and E-AIM show a moderately acidic condition for Beijing winter haze particles. It is noted in Figure 1 and Figure S5 that ISORROPIA and E-AIM predict nearly identical $\varepsilon(NH_4^+)$ and $\varepsilon(NO_3^-)$ and both can well capture the observed gas-particle partitioning. Therefore, we think that the difference between ISORROPIA and E-AIM may provide an estimate for the uncertainty of pH.

*Specific (but important) comments*

- The authors note early in the manuscript that the discrepancy in calculated pH when assuming $\gamma_{H+} = 1$ can be multiple units. This is not supported by the supplementary figure ($\gamma_{H+}$ from AIOMFAC), where for an ionic strength range of 0-100, log $\gamma_{H+}$ (hence the contribution of assuming $\gamma_{H+} = 1$ to pH discrepancy) varies within 1 unit. This has always been, by the way, our view – so it is nice to see this confirmed by the analysis presented! Also noted throughout is that pH is overestimated when assuming $\gamma_{H+} = 1$. This is not always true as well; as noted by the supplementary figure ($\gamma_{H+}$ from AIOMFAC), pH can be underestimated or overestimated by assuming $\gamma_{H+} = 1$, but not more than half a unit.

  Please see the responses to the general comments of activity coefficients above.

- In fact, the average pH estimated by ISORROPIA-II is *actually lower* than that reported for E-AIM (4.2 vs 5.4, for ISORROPIA-II vs E-AIM respectively) inconsistent with pH trends stated above.

  Wrong numbers of pH are quoted from the manuscript. As shown in Section 3.3, the mean pH values of 5.4 (E-AIM) and 4.2 (ISORROPIA) are for the *reverse mode* calculations. The average pH values are 4.0 (E-AIM) and 4.6 (ISORROPIA) for the *forward mode* calculations.

- In ISORROPIA-II, the non-ideal interactions of H+ with all the ions in solution (especially NO3, Cl, HSO4, SO4) is explicitly considered by the Kusik-Meisner and Bromely formulations. $\gamma_{H+} = 1$ is only invoked when the singe ion activity is required. This is not sufficiently noted in the text.

We noted in Section 2.2 that "With ISORROPIA, $\gamma_{H^+}$ and $\gamma_{OH^-}$ are assumed equal to unity, whereas the activity coefficients for the other ionic pairs (e.g., $H^+$–$Cl^-$) are calculated (Fountoukis and Nenes, 2007)."

- The authors understandably treat NVC (i.e. Ca, Mg, K, Na) as equivalent sodium, because E-AIM cannot explicitly treat Ca, Mg and K. The impact of this assumption can lead to important differences in the predicted thermodynamic state, owing to the strong nonideality of divalent ions and different water uptake characteristics of sodium salts vs. their other counterparts (e.g., Fountoukis et al., 2009).

  We noted in Figure 2 of the original manuscript that $K^+$ was accounted for as equivalent $Na^+$ in ISORROPIA, and therefore, for the comparison of pH for E-AIM and ISORROPIA, the model inputs are the same.

- What constitutes a "large/important" and "small/minor" different in pH depends on the context in which the pH is used. Constraining "absolute" pH for ambient aerosol to within less than 0.5 units may prove to be extremely challenging (e.g., the difference in log $\gamma_{H+}$ between E-AIM and AIOMFAC, the effects of organics on activity and water uptake and so on); so it may most likely be necessary to use a consistent pH calculation method and thermodynamic model when comparing aerosol acidities between models and/or observations.

  Such qualitative expressions have been avoided in the revised manuscript when comparing E-AIM and ISORROPIA.

- The authors caution about the predictions of ISORROPIA-II in metastable mode (for RH below the mutual deliquescence point) owing to the large ionic strengths of the solutions. Although we agree the ionic strengths are high, literature supports that the activity coefficient models used in ISORROPIA-II are stable for ionic strengths above 30, a situation that is also not the case for the Pitzer method (Kim et al., 1993).

  The Pitzer, Simonson, and Clegg (PSC) method, which is used in E-AIM, overcomes the limitation of molar-based ionic strength and is applicable over the entire concentration range (Zaveri et al., 2005).

In closing, we very much appreciate the analysis and it demonstrates an increasing sophistication of which the community is both understanding and discussing the thermodynamics of aerosols and the important topic of aerosol acidity. We also hope that the comments provided here add insight that will considerably strengthen the paper, and provide ideas for future work on the important topic of aerosol acidity.

The authors thank Nenes et al. again for the above comments, which were helpful in improving the quality of this manuscript.

[revised manuscript text omitted]

**Supplement**

**Contents:**

**Section S1. Revised ISORROPIA-II Model and Influence on pH Prediction**

**Section S2. Uncertainties of the AMS Measurements**

**Section S3. S-curves for gas-particle partitioning of NH₃, HNO₃, and HCl**

**Figures S1–S13**

**Tables S1–S8**

**Section S1. Revised ISORROPIA-II Model and Influence on pH Prediction**

The revised ISORROPIA-II model in this study has fixed some coding errors in the standard ISORROPIA-II model (*http://isorropia.eas.gatech.edu/*, last accessed: 2017/12/17). These errors are found to be closely related to aerosol water pH calculations under North China winter haze conditions. Note that only the forward stable state pH predictions are affected. Details are given in this section. The standard ISORROPIA-II model source code is password protected, but there is a version of ISORROPIA-II source code, implemented by Pye et al. (2009) into the GEOS-Chem chemical transport model and publicly accessible at *http://acmg.seas.harvard.edu/geos/doc/man/*. The code revision is available at *http://wiki.seas.harvard.edu/geos-chem/index.php/ISORROPIA_II#Bug_fixes_for_ISORROPIA_II_stable_mode* (last accessed: 2018/04/02). Details are given in this section.

**S1.1 General Solution Procedure of ISORROPIA-II**

As shown in the reference manual (*http://nenes.eas.gatech.edu/ISORROPIA/Version2_1/ISORROPIA21Manual.pdf*, last accessed: 2017/12/17), the ISORROPIA-II model consists of eight submodels according to the type of problem defined (forward or reverse) and the input chemical species (Table S1). For example, the submodel ISRP3F solves the forward problem for the $NH_3$–$Na$–$H_2SO_4$–$HNO_3$–$HCl$–$H_2O$ aerosol system.

Under each submodel, there are several subregimes determined by the molar ratios of basic chemical species ($NH_3$, $Na$, $K$, $Ca$, and $Mg$) to sulfuric acid (Table S2). These molar ratios are referred as "sulfate ratios". Table S3 presents the subregimes under the submodels ISRP3F and ISRP4F. Different major and minor species potentially present in the solution are assumed by different subregimes, which reduces the number of thermodynamic reactions required. For example, gas phase $NH_3$ is considered as a minor species for "sulfate rich" and "sulfate super-rich" aerosols, whereas bisulfate ion $HSO_4^-_{(l)}$ is a minor species for "sulfate poor" aerosols.

**Table S1.** Eight submodels in ISORROPIA-II

| Input Chemical Species | Submodel |
| :---: | :---: |
| $NH_3$, $H_2SO_4$ | ISRP1F (forward) or ISRP1R (reverse) |
| $NH_3$, $H_2SO_4$, $HNO_3$ | ISRP2F (forward) or ISRP2R (reverse) |
| $NH_3$, $H_2SO_4$, $HNO_3$, $Na$, $HCl$ | ISRP3F (forward) or ISRP3R (reverse) |
| $NH_3$, $H_2SO_4$, $HNO_3$, $Na$, $HCl$, $K$, $Ca$, $Mg$ | ISRP4F (forward) or ISRP4R (reverse) |

~~Under each submodel, there are several subregimes determined by the molar ratios of basic chemical species (NH₃, Na, K, Ca, and Mg) to sulfuric acid (Table S2). These molar ratios are referred as "sulfate ratios". Table S3 presents the subregimes under the submodels ISRP3F and ISRP4F. Different major and minor species potentially present in the solution are assumed by different subregimes, which reduces the number of thermodynamic reactions required. For example, gas phase NH₃ is considered as a minor species for "sulfate rich" and "sulfate super rich" aerosols, whereas bisulfate ion HSO₄⁻ (H) is a minor species for "sulfate poor" aerosols.~~

**Table S2.** Definition of different sulfate ratios

| Sulfate Ratio | Equation |
|---|---|
| Total sulfate molar ratio | $R_{Total} = \dfrac{\left[NH_3^{gas+aerosol} + Na^{gas+aerosol} + Ca^{gas+aerosol} + K^{gas+aerosol} + Mg^{gas+aerosol}\right]}{\left[H_2SO_4^{gas+aerosol}\right]}$ |
| Ammonia & Sodium molar ratio | $R_{NH_3+Na} = \dfrac{\left[NH_3^{gas+aerosol} + Na^{gas+aerosol}\right]}{\left[H_2SO_4^{gas+aerosol}\right]}$ |
| Crustal & Sodium molar ratio | $R_{Crustal+Na} = \dfrac{\left[Na^{gas+aerosol} + Ca^{gas+aerosol} + K^{gas+aerosol} + Mg^{gas+aerosol}\right]}{\left[H_2SO_4^{gas+aerosol}\right]}$ |
| Crustal molar ratio | $R_{Crustal} = \dfrac{\left[Ca^{gas+aerosol} + K^{gas+aerosol} + Mg^{gas+aerosol}\right]}{\left[H_2SO_4^{gas+aerosol}\right]}$ |
| Sodium molar ratio | $R_{Na} = \dfrac{\left[Na^{gas+aerosol}\right]}{\left[H_2SO_4^{gas+aerosol}\right]}$ |

**Table S3.** Subregimes under the submodels ISRP3F and ISRP4F

| Aerosol Type | Sulfate Ratio | Subregime | Subcase |
|---|---|---|---|
| *ISRP3F (NH₃–Na–H₂SO₄–HNO₃–HCl–H₂O aerosol)* | | | |
| Sulfate Poor, Sodium Rich | $R_{Na} \geq 2$ | H | H1–H6 |
| Sulfate Poor, Sodium Poor | $R_{NH_3+Na} \geq 2, R_{Na} < 2$ | G | G1–G5 |
| Sulfate Rich | $1 \leq R_{NH_3+Na} < 2$ | I | I1–I6 |
| Sulfate Super-Rich | $R_{NH_3+Na} < 1$ | J | J1–J3 |
| *ISRP4F (K–Ca–Mg–NH₃–Na–H₂SO₄–HNO₃–HCl–H₂O aerosol)* | | | |
| Sulfate Poor, Crustal & Sodium Rich, Crustal Rich | $R_{Crustal} > 2$ | P | P1–P13 |
| Sulfate Poor, Crustal & Sodium Rich, Crustal Poor | $R_{Crustal+Na} \geq 2, R_{Crustal} \leq 2$ | M | M1–M8 |
| Sulfate Poor, Crustal & Sodium Poor | $R_{Total} \geq 2, R_{Crustal+Na} < 2$ | O | O1–O7 |
| Sulfate Rich | $1 \leq R_{Total} < 2$ | L | L1–L9 |
| Sulfate Super-Rich | $R_{Total} < 1$ | K | K1–K4 |

Further, each subregime includes several subcases which depend on the input relative humidity (RH). This is because the possible solid salts have different associated deliquescence relative humidities (DRH). The RH ranges and possible solid and aqueous phases are shown in Table S4 (for subcases G1–G5) and Table S5 (for subcases O1–O7). For the stable state solution, RH increases gradually from G1 to G5 and from O1 to O7, and the solid salts are dissolved one

by one (depending on their DRH). When the input RH is larger than the DRH for all possible salts, an aqueous phase always exists (G5 and O7). G5 and O7 are used thus also for the metastable state solution (no precipitate is formed).

| Subcase | RH Subdomain | Notes |
|---|---|---|
| G1 | RH < DRNH4NO3 | Solids: $(NH_4)_2SO_4$, $NH_4NO_3$, $NH_4Cl$, $Na_2SO_4$;
Aqueous phase: Present when RH ≥ MDRH. |
| G2 | DRNH4NO3 ≤ RH < DRNH4CL | Solids: $(NH_4)_2SO_4$, $NH_4Cl$, $Na_2SO_4$;
Aqueous phase: Present when there is $NH_4NO_3$ (which deliquesces) or when RH ≥ MDRH. |
| G3 | DRNH4CL ≤ RH < DRNH42S4 | Solids: $(NH_4)_2SO_4$, $Na_2SO_4$;
Aqueous phase: Present when there is $NH_4NO_3$ or $NH_4Cl$ (which deliquesces) or when RH ≥ MDRH. |
| G4 | DRNH42S4 ≤ RH < DRNA2SO4 | Solids: $Na_2SO_4$;
Aqueous phase: Present. |
| G5 | RH ≥ DRNA2SO4 | Solids: None;
Aqueous phase: Present;
*This subroutine is used for the metastable mode calculation.* |

DRNH4NO3, DRNH4CL, DRNH42S4 and DRNA2SO4 represent the deliquescence relative humidity (DRH) of $NH_4NO_{3(s)}$, $NH_4Cl_{(s)}$, $(NH_4)_2SO_{4(s)}$, and $Na_2SO_{4(s)}$, respectively. The MDRH (mutual deliquescence relative humidity) for each subdomain represents the deliquescence point of the corresponding salt mixture and thus varies from case to case.

| Subcase | RH Subdomain | Notes |
|---|---|---|
| O1 | RH < DRNH4NO3 | Solids: $CaSO_4$, $(NH_4)_2SO_4$, $NH_4NO_3$, $NH_4Cl$, $MgSO_4$, $Na_2SO_4$, $K_2SO_4$;
Aqueous phase: Present when RH ≥ MDRH. |
| O2 | DRNH4NO3 ≤ RH < DRNH4CL | Solids: $CaSO_4$, $(NH_4)_2SO_4$, $NH_4Cl$, $MgSO_4$, $Na_2SO_4$, $K_2SO_4$;
Aqueous phase: Present when there is $NH_4NO_3$ (which deliquesces) or when RH ≥ MDRH. |
| O3 | DRNH4CL ≤ RH < DRNH42S4 | Solids: $CaSO_4$, $(NH_4)_2SO_4$, $MgSO_4$, $Na_2SO_4$, $K_2SO_4$;
Aqueous phase: Present when there is $NH_4NO_3$ or $NH_4Cl$ (which deliquesces) or when RH ≥ MDRH. |
| O4 | DRNH42S4 ≤ RH < DRMGSO4 | Solids: $CaSO_4$, $MgSO_4$, $Na_2SO_4$, $K_2SO_4$;
Aqueous phase: Present. |
| O5 | DRMGSO4 ≤ RH < DRNA2SO4 | Solids: $CaSO_4$, $Na_2SO_4$, $K_2SO_4$;
Aqueous phase: Present. |
| O6 | DRNA2SO4 ≤ RH < DRK2SO4 | Solids: $CaSO_4$, $K_2SO_4$;
Aqueous phase: Present. |
| O7 | RH ≥ DRK2SO4 | Solids: $CaSO_4$;
Aqueous phase: Present;
*This subroutine is used for the metastable mode calculation.* |

DRNH4NO3, DRNH4CL, DRNH42S4, DRMGSO4, DRNA2SO4 and DRK2SO4 represent the deliquescence relative humidity (DRH) of $NH_4NO_{3(s)}$, $NH_4Cl_{(s)}$, $(NH_4)_2SO_{4(s)}$, $MgSO_{4(s)}$, $Na_2SO_{4(s)}$, and $K_2SO_{4(s)}$, respectively. The MDRH (mutual deliquescence relative humidity) for each subdomain represents the deliquescence point of the corresponding salt mixture and thus varies from case to case. $CaSO_4$ is assumed completely insoluble (Fountoukis and Nenes, 2007).

**S1.2 Coding Errors within Several Subcases**

For the subcase G2 (an $NH_3$–Na–$H_2SO_4$–$HNO_3$–HCl–$H_2O$ aerosol, $R_{NH_3+Na} \geq 2$, $R_{Na} < 2$, DRNH4NO3 $\leq$ RH < DRNH4CL), an aqueous phase exists if $NH_4NO_3$ is present (which deliquesces). The problem is solved iteratively in ISORROPIA-II. For each iteration, it calculates the levels of solids, gases ($NH_3$, $HNO_3$, HCl), and aqueous ions. The major ions include $Na^+$, $NH_4^+$, $H^+$, $SO_4^{2-}$, $NO_3^-$, and $Cl^-$ ($HSO_4^-$ and $OH^-$ are considered minor species under such conditions) (Fountoukis and Nenes, 2007). The objective function is the departure of $Cl^-_{(l)}$, $NH_4^+_{(l)}$, $HCl_{(g)}$, and $NH_{3(g)}$ from the equilibrium reaction $NH_{3(g)} + HCl_{(g)} \leftrightarrow NH_4^+_{(l)} + Cl^-_{(l)}$. The aerosol water pH is calculated based on ion balance:

$$IB = \left[Na^+_{(l)}\right] + \left[NH_4^+_{(l)}\right] - \left[Cl^-_{(l)}\right] - \left[NO_3^-_{(l)}\right] - 2\times\left[SO_4^{2-}_{(l)}\right] \tag{S1}$$

Here, $\left[Na^+_{(l)}\right]$ is assumed to be zero as the RH is lower than the DRH of $Na_2SO_{4(s)}$ and its dissolution does not affect pH. Eq. (S1) indicates that $\left[NH_4^+_{(l)}\right]$, $\left[Cl^-_{(l)}\right]$, $\left[NO_3^-_{(l)}\right]$, and $\left[SO_4^{2-}_{(l)}\right]$ should be known in order to calculate pH.

The solution procedure begins by assuming that a very small amount of $Cl^-_{(l)}$ exists. $\left[NO_3^-_{(l)}\right]$ is computed taking advantage of the equilibrium reactions $HNO_{3(g)} \leftrightarrow H^+_{(l)} + NO_3^-_{(l)}$ and $HCl_{(g)} \leftrightarrow H^+_{(l)} + Cl^-_{(l)}$:

$$\left[NO_3^-_{(l)}\right] = \frac{\left[HNO_{3(T)}\right]}{1 + \frac{K_2}{K_1} \times \frac{\gamma^2_{HNO_3}}{\gamma^2_{HCl}} \times \frac{\left[HCl_{(T)}\right] - \left[Cl^-_{(l)}\right]}{\left[Cl^-_{(l)}\right]}} \tag{S2}$$

where $K_1$ and $K_2$ are the equilibrium constants for $HNO_{3(g)} \leftrightarrow H^+_{(l)} + NO_3^-_{(l)}$ and $HCl_{(g)} \leftrightarrow H^+_{(l)} + Cl^-_{(l)}$, respectively. The symbol $\gamma$ represents the activity coefficient. The subscript $_{(T)}$ defines the total input.

Then, $\left[NH_4^+_{(l)}\right]$ is calculated, which consists of two parts, $\left[NH_4^+_{(l),NC}\right]$ (associated with $NO_3^-_{(l)}$ and $Cl^-_{(l)}$) and $\left[NH_4^+_{(l),S}\right]$ (associated with $SO_4^{2-}_{(l)}$). Thus, $\left[NH_4^+_{(l)}\right] = \left[NH_4^+_{(l),NC}\right] + \left[NH_4^+_{(l),S}\right]$. $\left[SO_4^{2-}_{(l)}\right]$ and $\left[NH_4^+_{(l),S}\right]$ are computed from the equilibrium reaction $(NH_4)_2SO_{4(s)} \leftrightarrow 2NH_4^+_{(l)} + SO_4^{2-}_{(l)}$ solving a cubic equation. Note that $\left[NH_4^+_{(l),S}\right] = 2\times\left[SO_4^{2-}_{(l)}\right]$. Accordingly, Eq. (S1) becomes:

$$IB = \left[NH_4^+_{(l),NC}\right] - \left[Cl^-_{(l)}\right] - \left[NO_3^-_{(l)}\right] \tag{S3}$$

Eq. (S3) indicates that the estimation of $\left[NH_4^+_{(l),NC}\right]$ is important for pH calculation. However, we find, in the subcase G2 of the standard ISORROPIA-II model, that $\left[NH_4^+_{(l),NC}\right]$ is wrongly calculated by Eq. (S4):

$$\left[NH_4^+_{(l),NC}\right] = MIN\left(\left[Cl^-_{(l)}\right] + \left[NO_3^-_{(l)}\right], C_1\right) \tag{S4}$$

where $C_1 = \left[NH_{3(T)}\right] + \left[Na_{(T)}\right] - 2\times\left[H_2SO_{4(T)}\right]$. As the iteration begins with a very small $\left[Cl^-_{(l)}\right]$ (and thus a very small $\left[NO_3^-{}_{(l)}\right]$), Eq. (S4) is usually reduced to Eq. (S5):

$$\left[NH_4^+{}_{(l),NC}\right] = \left[Cl^-{}_{(l)}\right] + \left[NO_3^-{}_{(l)}\right] \tag{S5}$$

Consequently, the ion balance IB obtained from Eq. (S3) becomes zero in the subcase G2 and the pH is very often around 7 (i.e., neutral). On the other hand, the subcases G3, G4, and G5 in the ISORROPIA-II subregime G correctly calculate $\left[NH_4^+{}_{(l),NC}\right]$ based on the equilibrium reaction $NH_{3(g)} + H^+{}_{(l)} \leftrightarrow NH_4^+{}_{(l)}$ (with an equilibrium constant $K_3$) and the ion balance equation, Eq. (S1). The following equations are derived:

$$\frac{C_3\left(\left[NH_4^+{}_{(l),NC}\right]+C_2\right)}{\left(C_1-\left[NH_4^+{}_{(l),NC}\right]\right)} + \left[NH_4^+{}_{(l),NC}\right] - \left[Cl^-{}_{(l)}\right] - \left[NO_3^-{}_{(l)}\right] = 0 \tag{S6}$$

$$\left[NH_4^+{}_{(l),NC}\right]^2 - \left(C_1 + C_3 + \left[Cl^-{}_{(l)}\right] + \left[NO_3^-{}_{(l)}\right]\right)\left[NH_4^+{}_{(l),NC}\right] + C_1\left(\left[Cl^-{}_{(l)}\right] + \left[NO_3^-{}_{(l)}\right]\right) - C_2C_3 = 0 \tag{S7}$$

where $C_2 = 2\times\left[H_2SO_{4(T)}\right] - \left[Na_{(T)}\right]$, $C_3 = \frac{1}{K_3RT}\times\frac{\gamma_{NH_4NO_3}^2}{\gamma_{HNO_3}^2}$, $R$ is the gas constant, and T is the temperature. Eq. (S7) is a quadratic equation in which $\left[NH_4^+{}_{(l),NC}\right]$ is the only unknown.

The difference in calculating $\left[NH_4^+{}_{(l),NC}\right]$ between G2 (using Eq. (S5)) and G3–G5 (using Eq. (S7)) is that Eq. (S7) accounts for $NH_3$ evaporation. Note that if $K_3 \to \infty$ (i.e., $NH_3$ does not evaporate), then $C_3 \to 0$, and Eq. (S7) is reduced to Eq. (S8), which is essentially the same as Eq. (S4).

$$\left(\left[NH_4^+{}_{(l),NC}\right] - \left[Cl^-{}_{(l)}\right] - \left[NO_3^-{}_{(l)}\right]\right)\left(\left[NH_4^+{}_{(l),NC}\right] - C_1\right) = 0 \tag{S8}$$

The coding errors in the subcase G2 also affect the pH calculation for the subcase G1 ($R_{NH_3+Na} \geq 2$, $R_{Na} < 2$, RH < DRNH4NO3). An aqueous phase is present only for G1 when the RH is larger than the mutual deliquescence relative humidity (MDRH) of the salt mixture (($NH_4)_2SO_4$, $NH_4NO_3$, $NH_4Cl$, $Na_2SO_4$) (Table S4). In this situation, the ISORROPIA model calculates a "dry" solution of chemical composition (no aqueous phase) and a "wet" solution (assuming the deliquescence of $NH_4NO_3$) using results from the subcase G2. The actual gas/liquid/solid composition is then a weighted average of the "dry" and "wet" solutions (Fountoukis and Nenes, 2007). The molar concentrations of chemical species in the aqueous phase are the same as the results from G2, and thus the aerosol water pH in G1 is the same as that in G2.

Similar coding errors are found also for the subcases O1 and O2 (K–Ca–Mg–NH₃–Na–H₂SO₄–HNO₃–HCl–H₂O aerosol, $R_{Total} \geq 2$, $R_{Crustal+Na} < 2$; see Tables S3 and S5). Because the standard ISORROPIA-II model fails to account for $NH_3$ evaporation, the calculated aerosol water pH is very often ~ 7 for O1 and O2.

Overall, we have identified coding errors in the standard ISORROPIA-II model which are related to the calculation of aerosol water pH for the four subcases (G1, G2, O1, and O2). It is important to note that only the forward stable mode calculations are affected by these errors. The forward metastable mode solutions remain the same since other subcases (G5 and O7) are used. It is also important to note that these errors have little effect on the predicted gas phase $NH_3$ levels. In ISORROPIA-II, the gas phase $NH_3$ is computed from the difference between the total $NH_3$ and aqueous phase $NH_4^+$. The difference caused by these coding errors is equal to $\left[H^+_{(l)}\right]$, much smaller than $\left[NH^+_{4\,(l)}\right]$. In addition, the same coding issues also exist in previous ISORROPIA versions 1.5 and 1.7.

In this study, the ISORROPIA-II model with these coding errors fixed is denoted as the revised ISORROPIA-II model, which is used to predict aerosol water and pH in the stable state.

**S1.3 Sensitivity Tests**

In order to explore the effect of our model revisions on the aerosol water pH calculations in ISORROPIA-II, we have carried out two sets of sensitivity tests. The first is for an $NH_3$–Na–$H_2SO_4$–$HNO_3$–HCl–$H_2O$ aerosol system (Fig. S1). The forward metastable mode and forward stable mode simulations are performed for the standard ISORROPIA-II model; for the revised ISORROPIA-II model, only the forward stable mode simulations are made. The input data of Na, $HNO_3$, HCl, RH, and temperature are fixed, which represent the average $PM_1$ (particles with size smaller than 1 µm) observations of Beijing winter haze pollution episodes reported by Wang et al. (2016), and are summarized in Table S6. The levels of $H_2SO_4$ and $NH_3$ are varied over large ranges. As shown in Fig. S1d, three subcases (G1, I3, and J3) are included in these sensitivity tests. Our model revisions have no effect on I3 and J3. For G1, the standard forward stable mode simulations almost always predict pH around 7 (Fig. S1b), whereas the standard forward metastable mode simulations and the revised forward stable mode simulations predict similar values for pH < 7 (Fig. S1a–c). It is also seen from Fig. S1 that Beijing winter haze conditions fall within the subcase G1. Thus, our model revisions have a significant impact on estimating Beijing winter haze aerosol pH.

The second set of sensitivity tests is for a K–Ca–Mg–$NH_3$–Na–$H_2SO_4$–$HNO_3$–HCl–$H_2O$ aerosol. The ISORROPIA-II model simulations are analogous to those in the first set. The levels of $H_2SO_4$ and $NH_3$ are varied whereas the other inputs which represent the average $PM_{2.5}$ observations of Xi'an winter haze pollution episodes reported by Wang et al. (2016) are fixed. As shown in Fig. S2, our model revisions change the pH output (from ~ 7 to < 7) in the subcase O1 (most of the Xi'an winter haze conditions fall within O1), but do not affect the other subcases (P5, M1, L3, and K3). In addition, some non-monotonic features (i.e., noises) of the pH output are observed in Figs. S1 and S2 for all of the ISORROPIA-II simulations, when the total molar concentrations of basic species ($[K_{(T)}] + 2\times[Ca_{(T)}] + 2\times[Mg_{(T)}] + [Na_{(T)}] + [NH_{3(T)}]$) are smaller than those of acidic species ($2\times[H_2SO_{4(T)}] + [HNO_{3(T)}] + [HCl_{(T)}]$). Such noises are due likely to instability of the numerical solver used in ISORROPIA-II. This issue is currently being investigated by Dr. Sebastian D. Eastham (*wiki.seas.harvard.edu/geos-chem/index.php/ISORROPIA_II*, last accessed: 2017/12/01). Fortunately, this issue does not strongly affect the pH calculation results under North China winter haze conditions.

**Table S6.** Summary of gases and aerosol measurements in Beijing and Xi'an reported by Wang et al. (2016)

| | Beijing Polluted | | Xi'an Polluted | |
|---|---|---|---|---|
| Year | 2015 | | 2012 | |
| PM size | PM$_1$ | | PM$_{2.5}$ | |
| | Mean | Range | Mean | Range |
| SO$_4^{2-}$, µg m$^{-3}$ | 26 | 20–38 | 38 | 20–83 |
| NO$_3^-$, µg m$^{-3}$ | 26 | 4.5–48 | 33 | 12–55 |
| Cl$^-$, µg m$^{-3}$ | 1.7 | 0.0–4.5 | 14 | 2.6–34 |
| NH$_4^+$, µg m$^{-3}$ | 20 | 9.1–30 | 25 | 3.2–44 |
| Na$^+$, µg m$^{-3}$ | NA | NA | 4.2 | 0.5–17 |
| K$^+$, µg m$^{-3}$ | NA | NA | 4.6 | 1.8–8.3 |
| Ca$^{2+}$, µg m$^{-3}$ | NA | NA | 2.3 | 0.2–5.9 |
| Mg$^{2+}$, µg m$^{-3}$ | NA | NA | 0.3 | 0.0–0.8 |
| NH$_3$, ppb | 17 | 10–32 | 23 | 9.3–61 |
| T, °C | 0.9 | −1.7–8.2 | 4.1 | −3.1–14 |
| RH, % | 56 | 22–72 | 68 | 41–93 |

NA = Not Available. The polluted condition is defined by the concentration of SO$_4^{2-}$ > 20 µg m$^{-3}$.

[Figure]

**Figure S1.** Sensitivity of pH to the total (gas + aerosol) NH$_3$ and H$_2$SO$_4$ concentrations. The results reflect thermodynamic equilibrium predictions with different ISORROPIA-II model assumptions: (a) forward metastable mode, (b) standard forward stable mode, and (c) revised forward stable mode. The subregimes of the ISORROPIA-II forward stable mode are shown in panel (d). The solid red curves are used to distinguish different subregimes. The chemical and meteorological input data (total Na = 0 µg m$^{-3}$, total HNO$_3$ = 26 µg m$^{-3}$, total HCl = 1.7 µg m$^{-3}$, RH = 56%, T = 274.1 K) for the NH$_3$–Na–H$_2$SO$_4$–HNO$_3$–HCl–H$_2$O aerosol

system reflect average PM$_1$ measurements for Beijing winter haze pollution episodes reported by Wang et al. (2016). The dashed red curves indicate the situation in which the total molar concentrations of acidic and basic species are equal ([Na$_{(T)}$] + [NH$_{3(T)}$] = 2×[H$_2$SO$_{4(T)}$] + [HNO$_{3(T)}$] + [HCl$_{(T)}$]). Boxes define observed concentration ranges for the Beijing winter haze pollution episodes and diamonds represent the average Beijing haze conditions (total NH$_3$ = 32 µg m$^{-3}$, total H$_2$SO$_4$ = 26 µg m$^{-3}$).

[Figure]

**Figure S2.** Sensitivity of pH to the total (gas + aerosol) NH$_3$ and H$_2$SO$_4$ concentrations. The results reflect thermodynamic equilibrium predictions with different ISORROPIA-II model assumptions: (a) forward metastable mode, (b) standard forward stable mode, and (c) revised forward stable mode. The subregimes of the ISORROPIA-II forward stable mode are shown in panel (d). The solid red curves are used to distinguish different subregimes. The chemical and meteorological input data (total Na = 4.2 µg m$^{-3}$, total K = 4.6 µg m$^{-3}$, total Ca = 2.3 µg m$^{-3}$, total Mg = 0.3 µg m$^{-3}$, total HNO$_3$ = 34 µg m$^{-3}$, total HCl = 14 µg m$^{-3}$, RH = 68%, T = 277.3 K) for the K–Ca–Mg–NH$_3$–Na–H$_2$SO$_4$–HNO$_3$–HCl–H$_2$O aerosol system reflect average PM$_{2.5}$ measurements for Xi'an winter haze pollution episodes reported by Wang et al. (2016). The dashed red curves indicate the situation in which the total molar concentrations of acidic and basic species are equal ([K$_{(T)}$] + 2×[Ca$_{(T)}$] + 2×[Mg$_{(T)}$] + [Na$_{(T)}$] + [NH$_{3(T)}$] = 2×[H$_2$SO$_{4(T)}$] + [HNO$_{3(T)}$] + [HCl$_{(T)}$]). Boxes define observed concentration ranges for Xi'an winter haze pollution episodes and diamonds represent the average Xi'an haze conditions (total NH$_3$ = 41 µg m$^{-3}$, total H$_2$SO$_4$ = 39 µg m$^{-3}$).

We also calculate particle pH using our observational data collected during 2014 winter in Beijing and the standard and revised ISORROPIA-II models (Fig. S3). As expected, predicted pH values are different for the subcases G1, G2, O1, and O2. The predicted $NH_{3(g)}$ from the standard and revised calculations are similar and thus it is impossible to differentiate them by comparing the $NH_{3(g)}$ concentrations. Similarly, predicted particle $NH_4^+$ concentrations from the standard and revised model calculations should also be similar (because in the forward-mode calculations the total (gas + aerosol) quantity is fixed). Therefore, we believe that the measurement–model comparisons of $NH_3$ gas-particle partitioning for the standard ISORROPIA-II forward stable mode calculations cannot be used to evaluate the success or failure of pH predictions, in contrast to previous studies (Wang et al., 2016; Guo et al., 2017). The subtle difference $\Delta NH_3$ ($< 1\times10^{-3}$ ppb) shown in Figs. S3c and f suggests that incorporating the partitioning of $NH_3$ in the revised calculations pushes a little more ammonia to the gas phase, and thus more $H^+$ is needed in the aqueous phase and the solution is more acidic.

[Figure]

**Figure S3.** Comparisons of the predicted pH and gas phase $NH_3$ concentrations between the standard and revised ISORROPIA-II models with the stable state assumptions. (a–c) show the results using the AMS PM$_1$ measurements (an $NH_3$–$H_2SO_4$–$HNO_3$–$HCl$–$H_2O$ aerosol), and (d–f) show the results using the GAC-IC PM$_{2.5}$ measurements (a K–$NH_3$–Na–$H_2SO_4$–$HNO_3$–$HCl$–$H_2O$ aerosol).

**Section S2. Uncertainties of the AMS Measurements**

The AMS measurement uncertainty arises from inaccuracies in the ionization efficiency of nitrate ($IE_{NO_3}$), the relative ionization efficiency of a species $X$ relative to nitrate ($RIE_X$), the collection efficiency (CE), flow rate (Q), and the transmission efficiency (TE):

$$\frac{\Delta X}{X} = \sqrt{\left(\frac{\Delta IE_{NO_3}}{IE_{NO_3}}\right)^2 + \left(\frac{\Delta RIE_X}{RIE_X}\right)^2 + \left(\frac{\Delta CE}{CE}\right)^2 + \left(\frac{\Delta Q}{Q}\right)^2 + \left(\frac{\Delta TE}{TE}\right)^2} \tag{S9}$$

where $\frac{\Delta IE_{NO_3}}{IE_{NO_3}}$, $\frac{\Delta CE}{CE}$, $\frac{\Delta Q}{Q}$, and $\frac{\Delta TE}{TE}$ are estimated to be 10%, 30%, <0.5%, and 10%, respectively; and $\frac{\Delta RIE_X}{RIE_X}$ depends on the species $X$ (10% for ammonium, 15% for sulfate and 20% for organics) (Bahreini et al., 2009). Using the above equation, we estimate that the overall relative uncertainties of the AMS measurements are 33% (nitrate), 35% (ammonium), 36% (sulfate), and 39% (organics). The relative uncertainties for chloride and black carbon have not been quantified and are assumed to be 40% in this study.

**Section S3. S-curves for gas-particle partitioning of NH₃, HNO₃, and HCl**

Note that we assume water activity and all of the activity coefficients equal to unity (i.e., an ideal aqueous solution).

**S3.1 NH₃**

The ammonia–water equilibrium is (Seinfeld and Pandis, 2016)

$$NH_3\ (g) + H_2O \leftrightarrow NH_3 \cdot H_2O\ (l) \tag{S10}$$

$$NH_3 \cdot H_2O\ (l) \leftrightarrow NH_4^+ + OH^- \tag{S11}$$

Their equilibrium constants can be expressed as $H_{NH_3} = \frac{[NH_3 \cdot H_2O_{(l)}]}{p_{NH_3}}$ and $K_a = \frac{[NH_4^+_{(l)}][OH^-]}{[NH_3 \cdot H_2O_{(l)}]} = \frac{[NH_4^+_{(l)}][H^+]}{K_w[NH_3 \cdot H_2O_{(l)}]}$, where $H_{NH_3}$ (M atm⁻¹) is the Henry's law constant for NH₃, $p_{NH_3}$ (atm) is the partial pressure for NH₃, $K_a$ (M) is the dissociation equilibrium constant for NH₃·H₂O, $K_w$ (M²) is the dissociation equilibrium constant for water, and [X] represents aqueous concentrations of the species X (M). Thus, the total ammonia concentration in the liquid phase is

$$\left[NH_{4\,(Tl)}^+\right] = \left[NH_3 \cdot H_2O_{(l)}\right] + \left[NH_{4\,(l)}^+\right] = H_{NH_3}p_{NH_3}\left(1 + \frac{K_a}{[OH^-]}\right) = H_{NH_3}p_{NH_3}\left(1 + \frac{K_a[H^+]}{K_w}\right) \tag{S12}$$

Under neutral or acidic conditions, $\frac{K_a[H^+]}{K_w} \gg 1$, and thus $\left[NH_{4\,(Tl)}^+\right] \cong \frac{H_{NH_3}K_a}{K_w}[H^+]p_{NH_3}$. The aqueous fraction of total (gas + particle) ammonia, $\varepsilon\left(NH_4^+\right)$, is calculated as

$$\varepsilon\left(NH_4^+\right) = \frac{\frac{H_{NH_3}K_a}{K_w}[H^+]p_{NH_3}W}{\frac{H_{NH_3}K_a}{K_w}[H^+]p_{NH_3}W + \frac{p_{NH_3}}{RT}} = \frac{H_{NH_3}^* WRT}{1 + H_{NH_3}^* WRT} \tag{S13}$$

where W is the aerosol water content, R is the ideal gas constant, T is the ambient temperature, and $H_{NH_3}^* = \frac{H_{NH_3}K_a}{K_w}[H^+]$ is known as the effective Henry's law coefficient for NH₃.

**S3.2 HNO₃**

The nitric acid-water equilibrium is (Seinfeld and Pandis, 2016)

$$HNO_3\ (g) \leftrightarrow HNO_3\ (l) \tag{S14}$$

$$HNO_3\ (l) \leftrightarrow NO_3^- + H^+ \tag{S15}$$

The equilibrium constants for these two equations are $H_{HNO_3} = \frac{[HNO_{3(l)}]}{p_{HNO_3}}$ and $K_{n1} = \frac{[NO_3^-_{(l)}][H^+]}{[HNO_{3(l)}]}$, where $H_{HNO_3}$ (M atm⁻¹) is the Henry's law constant of HNO₃, $p_{HNO_3}$ (atm) is the partial pressure of HNO₃, and $K_{n1}$ is the dissociation equilibrium constant. The total nitrate in the liquid phase can be expressed as

$$\left[NO_3^-{}_{(l)}\right] = \frac{H_{HNO_3}K_{n1}}{[H^+]}p_{HNO_3} \tag{S16}$$

The aqueous fraction of total (gas + particle) nitric acid, $\varepsilon(NO_3^-)$, is calculated as

$$\varepsilon(NO_3^-) = \frac{\frac{H_{HNO_3}K_{n1}}{[H^+]}p_{HNO_3}W}{\frac{H_{HNO_3}K_{n1}}{[H^+]}p_{HNO_3}W + \frac{p_{HNO_3}}{RT}} = \frac{H_{HNO_3}^*WRT}{1 + H_{HNO_3}^*WRT} \tag{S17}$$

where W is the aerosol water content, R is the ideal gas constant, T is the ambient temperature, and $H_{HNO_3}^* = \frac{H_{HNO_3}K_{n1}}{[H^+]}$ is the effective Henry's law coefficient.

**S3.3 HCl**

Similar to the nitric acid-water equilibrium, the hydrochloric acid-water equilibrium is (Seinfeld and Pandis, 2016)

$$HCl\ (g) \leftrightarrow HCl\ (l) \tag{S18}$$

$$HCl\ (l) \leftrightarrow Cl^- + H^+ \tag{S19}$$

The equilibrium constants for these two equations are $H_{HCl} = \frac{[HCl_{(l)}]}{p_{HCl}}$ and $K_{n2} = \frac{[Cl^-][H^+]}{[HCl_{(l)}]}$, where $H_{HCl}$ (M atm$^{-1}$) is the Henry's law constant of HCl, $p_{HCl}$ (atm) is the partial pressure of HCl, $K_{n2}$ is the dissociation equilibrium constant of $HCl_{(l)}$. The total $[Cl^-]$ in the liquid phase can be expressed as

$$\left[Cl^-{}_{(l)}\right] = \frac{H_{HCl}K_{n2}}{[H^+]}p_{HCl} \tag{S20}$$

The aqueous fraction of total (gas + particle) hydrochloric acid, $\varepsilon(Cl^-)$, is calculated as

$$\varepsilon(Cl^-) = \frac{\frac{H_{HCl}K_{n2}}{[H^+]}p_{HCl}W}{\frac{H_{HCl}K_{n2}}{[H^+]}p_{HCl}W + \frac{p_{HCl}}{RT}} = \frac{H_{HCl}^*WRT}{1 + H_{HCl}^*WRT} \tag{S21}$$

where W is the aerosol water content, R is the ideal gas constant, T is the ambient temperature, and $H_{HCl}^* = \frac{H_{HCl}K_{n2}}{[H^+]}$ is known as the effective Henry's law coefficient for hydrochloric acid.

[Figure]

**Figure S4.** Time series of measured RH (a) and AMS PM$_1$ concentrations (b). The shaded area indicates a time period of ~ 6 days which were very dry (with RH from 7% to 34%) and relatively clean, and thus were not included in the thermodynamic analysis.

[Figure]

| △ | ISORROPIA Forward Metastable | ▽ | E-AIM Forward |
| × | ISORROPIA Reverse Metastable | + | E-AIM Reverse |

**Figure S5.** Relationship between ion balance and gas phase HNO$_3$ (a) and HCl (b) mixing ratios predicted using forward and reverse mode calculations. It is seen that the reverse mode calculations predict either very high or very low levels of HNO$_3$ and HCl depending on the sign (negative or positive) of the ion balance, whereas forward mode predictions are insensitive to ion balance. Because the measured mixing ratios of HNO$_3$ and HCl are very low and sometimes below detection limits, we do not present a quantitative comparison but show the 95% percentile of the HNO$_3$ and HCl data in our measurement period. As shown, the very high levels of HNO$_3$ and HCl in the reverse mode calculations (corresponding to negative ion balance, cations < anions, and low pH values, see Fig. 1 in the main text) are unlikely to be detected in the atmosphere.

[Figure]

**Figure S6.** Comparison of predicted pH and several other parameters by ISORROPIA and E-AIM (version II) under representative Beijing winter haze conditions (NH$_x$-rich). These variables are shown as a function of RH and total NH$_x$ concentrations. pH from ISORROPIA (a), pH from E-AIM (b), ΔpH (ISORROPIA − E-AIM) (c), $-\Delta \log_{10} m_{H^+}$ (d), $\log_{10} \gamma_{H^+}$ (e), and the ratio of AWC between E-AIM and ISORROPIA (f). The curve in each panel (c–f) shows the average value for each bin of RH. E-AIM (version II) and ISORROPIA are run in the forward metastable mode. The model inputs are calculated as the average values during haze episodes (RH > 60%) from our field measurements in Beijing, which include total (gas + particle) H$_2$SO$_4$ = 30 µg m$^{-3}$, total HNO$_3$ = 51 µg m$^{-3}$, and temperature = 278 K. The total NH$_x$ concentrations and RH vary from 25 to 100 µg m$^{-3}$ and from 30% to 90%, respectively. Na$^+$ and K$^+$ are accounted for as equivalent NH$_4^+$, and Cl$^-$ as equivalent NO$_3^-$. The average total NH$_x$ concentration in our measurements is 47 µg m$^{-3}$. Note that the $y$ axis is in log scale.

[Figure]

**Figure S7.** Comparisons of the ISORROPIA predicted pH (a–b), AWC (c), ionic strength (d), and partitioning of $NH_3$ (e–f) under assumptions of the metastable and stable phase states. The model inputs include total $H_2SO_4 = 30 \ \mu g \ m^{-3}$, total $HNO_3 = 51 \ \mu g \ m^{-3}$, total $NH_x = 47 \ \mu g \ m^{-3}$, temperature = 278 K, and varied RH values. The inputs are calculated from our field measurements during haze episodes (RH > 60%) as the average temperature and the average concentrations of total $H_2SO_4$, $HNO_3$, and $NH_x$. $Na^+$ and $K^+$ are accounted for as equivalent $NH_4^+$, and $Cl^-$ as equivalent $NO_3^-$. When the RH is between about 60% and about 80% (when both aqueous and solid phases are present for the stable solution), the predicted pH values for the stable solution are on average 0.02 ± 0.00 greater than those for the metastable solution. This difference in pH is small relative to the uncertainty resulting from other factors (e.g., measurements of gas and aerosol species and meteorological parameters).

~~Relationship between the ionic strength and hydrogen ion activity coefficient predicted by the AIOMFAC model under representative winter haze conditions (the relative abundance of $NH_4^+$, $SO_4^{2-}$, $NO_3^-$, $Cl^-$, $Na^+$, $K^+$, and $H^+$ in the aerosol solution is obtained from our field measurements). The AIOMFAC calculations are made at a temperature of 288 K (the applicable range of AIOMFAC is 298 ± 10 K), higher than the measured average ambient temperature (278 K) in this study, but activity coefficients are rather weak functions of temperature (web.meteo.mcgill.ca/aiomfac/).~~

[Figure]

**Figure S87.** pH (a) and AWC (b) predicted by the AMS $PM_1$ measurements and forward-mode ISORROPIA calculations using both stable and metastable state assumptions. Data are grouped in RH bins (10% increment). The shaded regions indicate the 25th and 75th percentiles. Note that the revised ISORROPIA model is used for the stable state. The uncertainties of ionic and gas measurements are considered using a Monte Carlo approach.

[Figure]

**Figure S9.** Sensitivity of particle pH to excess NH$_x$ and RH. The model simulations conducted are the same as in Fig. S6. The required NH$_x$ concentrations calculated for the input total H$_2$SO$_4$ and HNO$_3$ concentrations are 24 µg m$^{-3}$. The curves in panels (c and d) show the average pH in each bin of NH$_x$ concentrations or RH. Note that the *y* axis in panel (a–b) and the *x* axis in panel (c) are in log scale.

[Figure]

**Figure S10.** Sensitivity of Ca and Mg on particle pH evaluated using ISORROPIA forward metastable calculations. Based on the measured mass concentrations of Na$^+$, K$^+$, Ca$^{2+}$, and Mg$^{2+}$ in previous studies (summarized in Table S7), the concentration of Ca$^{2+}$ is rarely higher than K$^+$, and Mg$^{2+}$ is rarely higher than 20% of K$^+$ concentration. Thus we make a sensitivity test by assuming that Ca$^{2+}$ equals to K$^+$ and that the concentration of Mg$^{2+}$ is 20% of K$^+$. Results show, during winter haze events (RH > 60%), that including Ca$^{2+}$ and Mg$^{2+}$ in the calculations increases the predicted particle pH by 0.12 ± 0.05 unit.

**Table S7.** Mass concentrations of crustal species in PM$_{2.5}$ measured in Beijing during winter haze events (µg m$^{-3}$)

| Studies/Species | Na$^+$ | K$^+$ | Ca$^{2+}$ | Mg$^{2+}$ |
|---|---|---|---|---|
| Jiang et al. (2016) | 2.0±0.9 | 2.9±0.4 | 2.9±1.8 | 0.4±0.2 |
| Yang et al. (2015) | 0.9±0.3 | 1.4±0.9 | 0.5±0.3 | 0.1±0.1 |
| Huang et al. (2014) | 1.0±0.5 | 4.2± 2.1 | 0.4± 0.3 | 0.2±0 |
| Liu et al. (2017) case 1 | 0.9±0.2 | 1.8±0.5 | 0.8±0.2 | 0.1±0.0 |
| Liu et al. (2017) case 2 | 0.5±0.1 | 0.4±0.2 | 0.1±0.1 | 0.1±0.0 |
| Liu et al. (2017) case 3 | 0.8±0.2 | 0.6±0.3 | 0.04 | 0.04 |

[Figure]

**Figure S11.** The potential impact of aerosol water associated with organic compounds and black carbon on the predicted fine particle pH. The pH values are obtained from the ISORROPIA forward mode metastable calculations. The uncertainties of ionic and gas measurements are considered using a Monte Carlo approach. The solid and dashed curves use $\kappa_{org}$ of 0.06 and 0.20, respectively, and both use a $\kappa$ of 0.04 for black carbon. The vertical lines indicate the average ∆pH values of 0.05 and 0.13, respectively.

**Table S8.** Mass concentrations of major organic acid salts in PM$_{2.5}$ measured in urban Beijing during winter (ng m$^{-3}$)

| Reference | Wang et al. (2007) | Huang et al. (2005) | Du et al. (2014) | Jiang et al. (2016) | Wang et al. (2017) |
|---|---|---|---|---|---|
| Year | 2002 | 2003 | 2010 | 2014 | 2014 |
| Oxalic | 477±304 | 107±35 | 195±137 | 441±429 | 166±157 |
| Malonic | | 28±12 | | | 16±11 |
| Succinic | | 24±7 | | | 36±26 |
| Glutaric | | 10±4 | | | 5±4 |
| Formic | 178±81 | | | | |
| Acetic | 3±3 | | | | |
| Glyoxylic | | 18±5 | | | 20±23 |
| Pyruvic | | 31±14 | | | 15±9 |

Blank means not measured.

[Figure]

**Figure S12.** The potential impact of oxalic acid on particle pH evaluated using the E-AIM forward-mode calculations. The *x* axis defines the pH values when only inorganic species ($Na^+$, $NH_4^+$, $SO_4^{2-}$, $NO_3^-$, and $Cl^-$) are included in the aerosol system, and the *y* axis indicates the pH values when oxalate is also included. The dashed line indicates a 1:1 relationship.

[Figure]

**Figure S11.** Submicron particle organic aerosol atomic O/C ratios as a function of RH. Data are measured by the AMS and grouped in RH bins (10% increment). The shaded region indicates the 25[th] and 75[th] percentiles.